# scCBGM: Single-Cell Editing via Concept Bottlenecks

## Abstract

How would a cell behave under different conditions? Counterfactual editing of single cells is essential for understanding biology and designing targeted therapies, yet current scRNA-seq generative methods fall short: disentanglement models rarely support interventions, and most intervention-based approaches perform conditional generation that synthesizes new cells rather than editing existing ones. We introduce Single-Cell Concept Bottleneck Generative Models (scCBGMs), unifying counterfactual reasoning and generative modeling. scCBGM incorporates decoder skip connections and a cross-covariance penalty to decouple annotated concepts from unannotated sources of variation, enabling robust counterfactuals even under noisy concept annotations. Using an abduction–action–prediction procedure, we edit cells at the concept level with per-cell precision and generalize zero-shot to unseen concept combinations. Conditioning modern generators (e.g., flow matching) on scCBGM embeddings preserves state-of-the-art fidelity while providing precise controllability. Across three datasets (up to 21 cell types), scCBGM improves counterfactual accuracy by up to 4×. It also supports mechanism-of-action analyses by jointly editing perturbation and pathway-activity concepts in real scRNA-seq data. Together, scCBGM establishes a principled framework for high-fidelity in silico cellular experimentation and hypothesis testing in single-cell biology.

## 1 Introduction

Understanding cellular phenotypes and how they mediate response to exposures, e.g., drugs, cytokines, chemokines, is critical to disease biology and translational clinical research. Single-cell RNA sequencing (scRNA-seq) enables the characterization of cellular phenotypes and measuring responses at cellular resolution, revealing cell states, trajectories, and disease mechanisms (Bergen et al., 2020; Aevermann et al., 2018; Kang et al., 2017; Wu et al., 2021). Yet the combinatorial space of cellular populations and conditions (treatments, exposures, doses) makes exhaustive experimental mapping infeasible (Kana et al., 2023). Computational models that can help fill this map by predicting cellular responses under unseen conditions are therefore essential.

A critical capability required of these models is the ability to perform precise **cellular editing**: starting with an observed cell and systematically modifying specific biological properties while preserving others. Editing differs from conditional generation in that the latter considers *any* cell under given conditions, while the former considers a *specific* cell under altered conditions. For example, a researcher might ask *"What would this T-cell look like after receiving anti-PD-1 or anti-TNF treatment, compared to its observed untreated state? What if its NF-kB pathway is turned off?"* or *"What would this KRAS-mutant tumor cell look like if PI3K were turned off instead of on?"*. Such editing capabilities enable cell-level counterfactual reasoning, which is critical for causal discovery, therapeutic design, and precision medicine.

A cell-editing model must satisfy two requirements. First, it should generate cell-specific counterfactuals, predicting how an observed cell would respond to a specified intervention, rather than only capturing population-level averages. Second, it should provide interpretable control, allowing interventions on biologically meaningful concepts such as gene programs or cell types, rather than on opaque latent variables. Earlier methods modeled conditional distributions of cell states across different contexts, capturing population-level effects but not counterfactuals for individual cells (Lotfollahi et al., 2019; Kana et al., 2023; Adduri et al., 2025). More recent work has moved

toward cell-level counterfactual prediction (Zhang et al., 2024; Piran et al., 2024), but explicit, interpretable control remains an open and important area for further development. Meanwhile, existing interpretability methods are descriptive but not actionable: they explain correlations in the data but cannot be used to simulate or edit cellular responses (Zhao et al., 2021; Chen et al., 2024). Enabling precise counterfactual editing at the level of individual cells with biologically interpretable control remains an open challenge.

In this work, we introduce single-cell Concept Bottleneck Generative Models (scCBGM), a generative framework that enables interpretable and controllable cellular editing. Our approach builds on Concept Bottleneck Generative Models (CBGMs) (Ismail et al., 2023), adapting them to the unique challenges of single-cell data: high heterogeneity, complex biological processes, substantial technical noise, and unreliable concept annotations. Our main contributions are:

- We introduce architectural modifications to CBGMs: a computationally efficient cross-covariance penalty that promotes decoupled embeddings without imposing dimensionality constraints on model components, and decoder concept skip connections. Together, these enable controllable generation under noisy biological annotations.

- We extend scCBGM beyond VAEs to flow matching models, enabling concept-guided editing in both decoding-only and encoding–decoding regimes. This demonstrates the flexibility of our framework across generative architectures.

- We develop a synthetic data generation process that separates exogenous noise from conditions, providing access to true counterfactuals. This enables systematic evaluation of concept-based editing under realistic single-cell settings (noise, missing labels, heterogeneous populations) and creates a valuable benchmark for future methods.

- We demonstrate superior editing accuracy and zero-shot generalization across three real-world datasets and our synthetic benchmark, outperforming several state-of-the-art methods. We show a use case where in-silico interventions with scCBGM on both cellular phenotype, i.e., pathway activity *and* treatment elucidate mechanistic hypotheses of treatment response.

Our framework provides computational biologists with practical tools for cellular editing that guide experimental design and accelerate biological discovery.

## 2    scCBGM Single-Cell Concept Bottleneck Generative Model

This section formalizes the problem of single-cell counterfactual editing, describes the architecture and training objectives of scCBGM, and details a procedure for augmenting flow matching generative models with scCBGM.

### 2.1    Problem setup

We assume $N$ i.i.d. single-cell RNA-seq profiles $\mathbf{x} \in \mathbb{R}^d$, each with $d$ genes, and an associated vector of $K$ biological concepts $\mathbf{c} \in \mathbb{R}^K$ (e.g., cell type labels or treatment). Our goal is to generate *counterfactual gene expression*: given a factual cell $(\mathbf{x}, \mathbf{c})$, predict what the *same* cell would look like if its concepts had been set to $\mathbf{c}'$ instead.

**Notation.**   We use capital letters for random variables $(U, U_C, C, X)$, lowercase bold for observed data $(\mathbf{x}, \mathbf{c}, \mathbf{c}')$, and lowercase with hats for model predictions $(\hat{\mathbf{c}}, \hat{\mathbf{u}})$.

**Data-generating process.**   We adopt Pearl's structural causal model (SCM) formalism (Pearl, 2009). A gene expression $X$ is influenced by two sources of variation as illustrated in Figure 1: the observed concepts $C$ and unobserved residual factors $U$. Concepts themselves are driven by their own exogenous variables $U_C$:

$$U \sim P(U), \quad U_C \sim P(U_C), \qquad C \leftarrow f_C(U_C), \qquad X \leftarrow f_X(C, U). \tag{1}$$

where $P(U)$ and $P(U_C)$ are probability distributions over the unobserved factors, and $f_C$ and $f_X$ are functions that determine how these factors generate concepts and gene expression, respectively. We assume $U \perp U_C$; therefore $U \perp C$ under $C \leftarrow f_C(U_C)$.

**Counterfactuals.**   The key insight is to ask *"what if"* questions about cells. Given observed gene expression $\mathbf{x}$ and associated concepts $\mathbf{c}$, we ask: what would the *same* cell look like if its concepts were instead $\mathbf{c}'$, everything else being equal? We refer to the observed gene expression as the

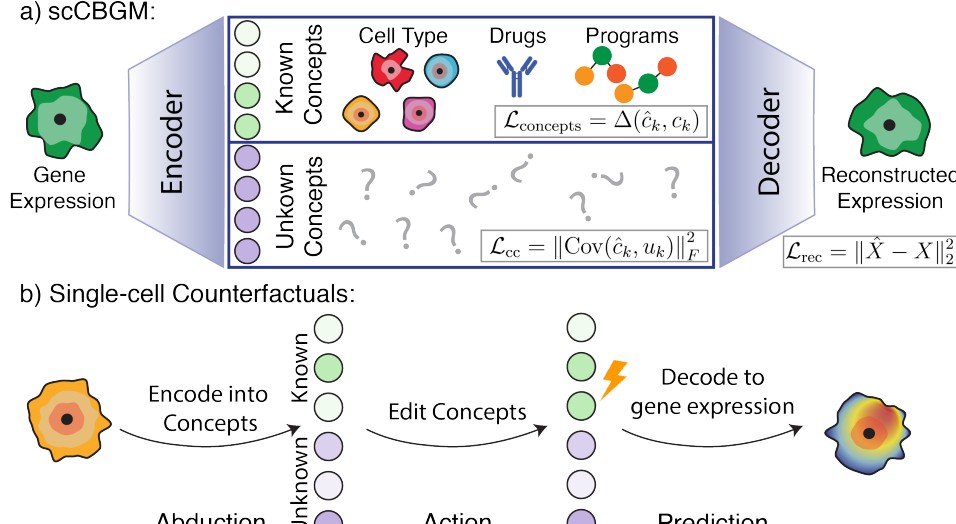

Figure 2: Single-cell Concept Bottleneck Generative Model Overview. **Top:** A concept bottleneck VAE encodes gene expression into known concepts and unknown concepts, which are subsequently decoded to reconstruct the expression profile. The model is trained to match the encoded concepts ($\hat{c}_k$) to the ground truth ($c_k$) while keeping the unknown concepts ($u_k$) independent via the cross-covariance loss. **Bottom:** Counter-factuals are constructed by: **1.** encoding each cells into its known and unknown concept. **2.** editing the desired concepts while keeping the rest stationary and **3.** decoding with the edited embedding.

*factual* outcome $X$, and the hypothetical alternative as the *counterfactual* outcome $X'$. Formally, for a given unobserved residual factor $\mathbf{u}$, if the concepts were $\mathbf{c}'$ rather than $\mathbf{c}$, the counterfactual gene expression would be:

$$X'(\mathbf{u}) = f_X(\mathbf{c}', \mathbf{u}), \tag{2}$$

where $\mathbf{u}$ represents the same unobserved factors that generated the original cell. Given a factual observation $\mathbf{x}$, we define the *counterfactual edit* of that cell to concepts $\mathbf{c}'$ as the expected counterfactual outcome:

$$\mu_{\mathbf{x},\mathbf{c}'} := \mathbb{E}_U[f_X(\mathbf{c}', U) \,|\, X = \mathbf{x}]. \tag{3}$$

This represents what we expect that specific cell would look like on average if it had concepts $\mathbf{c}'$ instead, where the expectation is over the posterior of $U$ given $X = \mathbf{x}$ (abduction).

**Cell editing as counterfactual prediction.** We model single-cell editing as learning a predictor of counterfactual edit (Equation 3). Given an observed cell $\mathbf{x}$ with concepts $\mathbf{c}$, the goal is to learn a function $f_\theta$ that generates the cell under alternative concepts $\mathbf{c}'$:

$$\hat{\mathbf{x}}' = f_\theta(\mathbf{x}, \mathbf{c}') \approx \mu_{\mathbf{x},\mathbf{c}'}. \tag{4}$$

In other words, $f_\theta$ estimates how the same cell $\mathbf{x}$ would appear if its concepts were $\mathbf{c}'$ rather than $\mathbf{c}$; this is equivalent to the cell editing task.

### 2.2 ARCHITECTURE OF SCCBGM

The overall architecture of our model is shown in Figure 2. It follows an encoder–decoder design with a concept

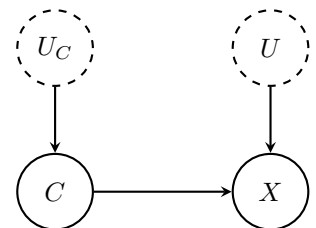

Figure 1: Directed Acyclic Graph (DAG) of the data-generating process. Two unobserved variables noises $U$ and $U_C$ drive $X$ and $C$, respectively, with $C$ also influencing $X$, consistent with $C \leftarrow f_C(U_C)$, $X \leftarrow f_X(C, U)$.

bottleneck. The encoder $E(\cdot)$ maps single-cell expression $\mathbf{x}$ to a latent $\mathbf{z}$, which is decomposed into known factors $\hat{\mathbf{c}}$ and unknown factors $\mathbf{u}$ reflecting the assumed data-generating process. These are concatenated and passed to the decoder $D(\cdot)$ to reconstruct $\mathbf{x}$. Details of the encoder, decoder, and bottleneck are given below, with losses and training described in Section 2.3.

**Encoder** The encoder is a neural network $E(\cdot)$ with two heads that maps the input vector $\mathbf{x} \in \mathbb{R}^d$ to the means and standard deviation of a multivariate Gaussian: $E(\mathbf{x}) = \big(E_\mu(\mathbf{x}), E_\Sigma(\mathbf{x})\big) = (\mu_z, \Sigma_z)$, that defines a posterior distribution $q(\mathbf{z}|\mathbf{x})$. A latent vector $\mathbf{z}$ is then sampled from $q(\mathbf{z}|\mathbf{x}) = \mathcal{N}(\mu_z, \Sigma_z)$.

**Concept bottleneck module** The concept bottleneck module is composed of (1) a concept network Koh et al. (2020) that parametrizes a function $f_c$ mapping the latent representation $\mathbf{z}$ into concept predictions $\hat{\mathbf{c}} = f_c(\mathbf{z}) \in \mathbb{R}^K$, and (2) a network $f_u$ that produces representation $\mathbf{u} = f_u(\mathbf{z}) \in \mathbb{R}^{d_u}$ capturing variation not explained by $\hat{\mathbf{c}}$. We refer to $\hat{\mathbf{c}}$ and $\mathbf{u}$ as the known and unknown factors, respectively.

**Decoder with concept skip connections** The decoder is a network $D(\cdot)$ that takes the concatenation of the known and unknown factors as input ($\mathbf{h}_0 = [\mathbf{u}, \hat{\mathbf{c}}]$). At each layer $\ell > 1$, we feed the previous hidden state concatenated with the known factors:

$$\mathbf{h}_1 = D_1(\mathbf{h}_0), \quad \mathbf{h}_\ell = D_\ell\big([\mathbf{h}_{\ell-1}, \hat{\mathbf{c}}]\big), \quad \ell = 2, \dots, L, \qquad \hat{\mathbf{x}} = D_{\text{final}}(\mathbf{h}_L), \tag{5}$$

Including $\hat{\mathbf{c}}$ at every layer enforces a stronger and more systematic conditioning on the concepts. This leads to better performance than only adding $\hat{\mathbf{c}}$ in the input $\mathbf{h}_0$, especially when dealing with noisy concept annotations, as demonstrated empirically in Section 3.

## 2.3 LOSS FUNCTIONS AND TRAINING

We train the model by minimizing a loss derived from the $\beta$-VAE evidence lower bound (ELBO), *i.e.,* a reconstruction term, a concept supervision term, and a $\beta$-scaled KL regularization:

$$\mathcal{L}_{\text{VAE}} = -\mathbb{E}_{q(\mathbf{z}|\mathbf{x})}[\log p(\mathbf{x}|\mathbf{z})] + \lambda_c\,\mathcal{L}_{\text{concept}} + \beta\,\text{KL}(q(\mathbf{z} \mid \mathbf{x}) \,\|\, p(\mathbf{z}))\,. \tag{6}$$

where $\mathcal{L}_{\text{concept}}$ is a surrogate for $-\mathbb{E}_{q(\mathbf{z}|\mathbf{x})}[\log p(\mathbf{c} \mid \mathbf{z})]$, as described next.

**Concept loss $\mathcal{L}_{\text{concept}}$** Biological data often has missing concept annotations. To accommodate this, we define a binary mask $\mathbf{m} \in \{0,1\}^K$ indicating which of the $K$ total concepts are present for a given sample. We partition the concepts into binary and continuous sets, indexed by $K_{\text{bin}}$ and $K_{\text{cont}}$ respectively. The total concept loss is the normalized sum of the **Binary Cross Entropy (BCE)** loss for binary concepts and the **Mean Squared Error (MSE)** loss for continuous concepts:

$$\mathcal{L}_{\text{concept}} = \frac{1}{\sum_k m_k} \left( \sum_{k \in K_{\text{bin}}} m_k\, \mathcal{L}_{\text{BCE}}(\hat{c}_k, c_k) + \sum_{k \in K_{\text{cont}}} m_k\, \mathcal{L}_{\text{MSE}}(\hat{c}_k, c_k) \right), \tag{7}$$

where $\hat{c}_k$ is the predicted value for the ground-truth concept $c_k$.

**Cross-covariance penalty** To enforce independence between latent factors $U$ and concepts $C$, we penalize their cross-covariance. Unlike cosine-similarity losses, this approach accommodates arbitrary embedding dimensions ($K \neq d_u$). For a minibatch of size $B$, the penalty is the squared Frobenius norm of the empirical cross-covariance between predicted concepts $\hat{C} \in \mathbb{R}^{B \times K}$ and unknown factors $U \in \mathbb{R}^{B \times d_u}$:

$$\mathcal{L}_{\text{cc}} = \left\| \frac{1}{B-1}(\hat{C} - \mathbf{1}\boldsymbol{\mu}_{\hat{c}}^\top)^\top (U - \mathbf{1}\boldsymbol{\mu}_u^\top) \right\|_F^2 \tag{8}$$

where $\boldsymbol{\mu}_{\hat{c}}$ and $\boldsymbol{\mu}_u$ are the column mean vectors of $\hat{C}$ and $U$, respectively. Scaling is enforced implicitly for $\hat{C}$ via the **Concept Loss**, and for $U$ via a ReLU non-linearity. This prevents degenerate solutions where the unknown factors take on near-zero values. This pushes the model to encode non-concept specific sources of variation (e.g., batch, cell type) in $\mathbf{u}$, as theoretically analyzed in Appendix B.1 and empirically demonstrated in Appendix D.7.

**Final loss** The complete training objective is obtained by combining $\mathcal{L}_{\text{VAE}}$ and $\mathcal{L}_{\text{cc}}$:

$$\mathcal{L} = \mathcal{L}_{\text{VAE}} + \lambda_{\text{cc}}\mathcal{L}_{\text{cc}} \tag{9}$$

We provide proofs for the consistency of the counterfactual estimator learnt by minimizing the above loss in Appendix B.

**Differences with standard concept bottleneck generative models.** Our approach differs from standard CBGMs (Ismail et al., 2023) in three important ways: (i) we use a standard concept bottleneck model (Koh et al., 2020), where the bottleneck maps inputs to scalar concept predictions that directly correspond to interpretable concepts, rather than a concept embedding model (Espinosa Zarlenga et al., 2022) where the bottleneck maps each concept to high-dimensional representations; (ii) we add skip connections to the decoder to maintain persistent concept conditioning; and (iii) we introduce a cross-covariance loss instead of the cosine similarity loss for orthogonality. Changes (i) and (ii) improve robustness to noisy concept annotations, while change (iii) removes dimensional constraints on embeddings that are enforced by CBGMs. We show empirical performance gains compared to existing CBGMs in Section 3.

## 2.4 COUNTERFACTUAL PREDICTION

Our model enables counterfactual predictions by implementing the standard abduction-action-prediction framework (Pearl, 2009) with our generative architecture. (**1-Abduction Step**) Given an observed cell $\mathbf{x}$, we first encode it with our encoder to obtain $\mathbf{z} = E_\mu(\mathbf{x})$, and decompose it into concepts $\hat{\mathbf{c}}$ and unknown factors $\mathbf{u}$. (**2-Action Step**) We edit the desired concept values, to obtain a modified concept vector $\hat{\mathbf{c}}'$. We edit the vector $\hat{\mathbf{c}}$ by assigning the dimensions $k$ for which the $\hat{\mathbf{c}}$ and $\mathbf{c}'$ differ, and leave the other dimensions untouched. That is, $\hat{\mathbf{c}}' \equiv \hat{\mathbf{c}}_k \leftarrow \mathbf{c}'_k, \forall k : \mathbf{c}'_k \neq \mathbf{c}_k$. (**3-Prediction Step**) Lastly, we decode the modified representation $[\mathbf{u}, \hat{\mathbf{c}}']$ to produce the counterfactual prediction $\hat{\mathbf{x}}' = D([\mathbf{u}, \hat{\mathbf{c}}'])$. Figure 2 shows an illustration of the abduction-action-prediction process.

## 2.5 ENABLING FINE-GRAINED CONTROL OF GENERATIVE MODELS WITH scCBGM CONDITIONING

While scCBGM is a standalone generative model, it can also enhance other generative models' architectures to enable more fine-grained control. By conditioning the generative process on the scCBGM embeddings, we can combine the generation quality of state-of-the-art generative models with the interpretability and controllability of our method. In this section, we describe the procedure for flow matching (FM) models (Lipman et al., 2023; Liu et al., 2023); the extension to diffusion models is analogous. For further details on flow matching, we refer the reader to the excellent tutorial in Lipman et al. (2024).

**scCBGM-guided FM** Given a trained scCBGM-VAE, we train a flow matching model by directly conditioning on the scCBGM embeddings. For a cell $\mathbf{x}$, we extract its concept embeddings $\hat{\mathbf{c}} = f_c(\mathbf{z})$ and unknown representations $\mathbf{u} = f_u(\mathbf{z})$ where $\mathbf{z} = E_\mu(\mathbf{x})$. We use these vectors to learn a conditional vector field $v_\theta(\mathbf{x}_t, t; [\mathbf{u}, \hat{\mathbf{c}}])$ by minimizing the conditional flow matching loss:

$$\mathcal{L}_{\text{FM}} = \mathbb{E}_{t, \zeta \sim q, \mathbf{x}_t \sim p_t(\cdot | \mathbf{z})} \left[ \| v_\theta(\mathbf{x}_t, t; [\mathbf{u}, \hat{\mathbf{c}}]) - v^*(\mathbf{x}_t, t \mid \zeta) \|_2^2 \right] \tag{10}$$

Here, $v^*$ is the target conditional velocity field that generates the conditional probability paths $p_t(\cdot \mid \zeta)$ for $t \in [0, 1]$, with $p_0(\cdot|\zeta)$ and $p_1(\cdot|\zeta)$ are boundary conditional probabilities whose marginal match with a noise distribution and the data distribution at $t = 0$ and $t = 1$ respectively (Lipman et al., 2023). The learnt conditional vector field $v_\theta(\mathbf{x}_t, t; [\mathbf{u}, \hat{\mathbf{c}}])$ generates a flow $\varphi_t(\mathbf{x}, [\mathbf{u}, \hat{\mathbf{c}}])$ that is defined by $\frac{d}{dt}\varphi_t(\mathbf{x}, [\mathbf{u}, \hat{\mathbf{c}}]) = v_\theta(\varphi_t(\mathbf{x}_t, [\mathbf{u}, \hat{\mathbf{c}}]), t; [\mathbf{u}, \hat{\mathbf{c}}])$ and $\varphi_0(\mathbf{x}, [\mathbf{u}, \hat{\mathbf{c}}]) = \mathbf{x}$. Using the flow, one can generate counterfactuals in two ways: decoding-only and encoding-decoding.

**Decoding-only counterfactual prediction** Given $\mathbf{x}$, we obtain $\hat{\mathbf{c}}'$ via the abduction and action steps on scCBGM. We then predict the counterfactuals by sampling $\mathbf{x_0} \sim \mathcal{N}(\mathbf{0}, \mathbf{I})$ and passing it to the conditional flow $\hat{\mathbf{x}}' = \varphi_1(\mathbf{x_0}, [\mathbf{u}, \hat{\mathbf{c}}'])$. This approach is useful for generating diverse examples of cells with a specific profile. We refer to this approach as scCBGM-FM (decode) .

**Encoding-decoding counterfactual prediction** A more accurate option for counterfactual prediction relies on mapping $\mathbf{x}$ back to the noise distribution (encoding step) and then using the conditional flow from this initial condition to generate the modified cell (decoding step). Importantly, the encoding step uses a conditioning on $\hat{\mathbf{c}}$ while, the decoding uses the edited concepts $\hat{\mathbf{c}}'$. Given a starting cell $\mathbf{x}$, we *encode* it using $\mathbf{x}_0 = \varphi_1^{-1}(\mathbf{x}, [\mathbf{u}, \hat{\mathbf{c}}])$, then *decode* it using $\hat{\mathbf{x}}' = \varphi_1(\mathbf{x}_0, [\mathbf{u}, \hat{\mathbf{c}}'])$. This implements Pearl's abduction-action-prediction framework (Pearl, 2009; Xia et al., 2025; Rout et al., 2024): mapping $\mathbf{x}$ to noise $\mathbf{x}_0$ constitutes the *abduction* of exogenous factors (cell identity), while

the forward pass under intervention $\hat{c}'$ performs the *action* and *prediction* (Sanchez & Tsaftaris, 2022; Wang et al., 2024b). We refer to this approach as scCBGM-FM (edit) .

Before evaluating scCBGM on real datasets, we validate our architectural choices on synthetic data where ground-truth counterfactuals are available, enabling direct measurement of counterfactual accuracy.

# 3 WHY DO WE NEED A SINGLE-CELL SPECIFIC CBGM?

While concept bottleneck models have been explored in prior work, existing architectures do not address the unique challenges of scRNA-seq data, a modality that presents fundamentally different problems than well-curated domains such as images. In scRNA-seq, some concepts are well-defined by the experimental design (e.g., drug treatments, dosages), whereas others are inherently noisier (e.g., cell state, pathway activation). Noisy labels may arise from challenging annotation tasks, limited signal, incomplete coverage, or redundancy across concepts. To demonstrate that our architectural modifications address these challenges, we systematically compare scCBGM against standard CBGMs using synthetic data where ground-truth counterfactuals are available for direct evaluation.

**Synthetic Data for Evaluation**   To compare model performance, we evaluate (1) cell-level counterfactual generation and (2) robustness to noisy concepts. Because real data do not permit controlled evaluation, we propose a synthetic scRNA-seq generation process based on a hierarchical overdispersed Poisson model Pan et al. (2023); Xiao et al. (2021); Subedi & Dang (2025). In this process, gene expression is decomposed into contributions from batches, tissues, cell types, and concepts. By isolating exogenous noise, we can generate true counterfactuals for evaluating model predictions. The process also supports the injection of four types of noisy concept annotations: incorrectly annotated, irrelevant, missing, and duplicated concepts. Appendix C.1-C.3 describes the data generating process , and Appendix C.4 details the noisy concept types.

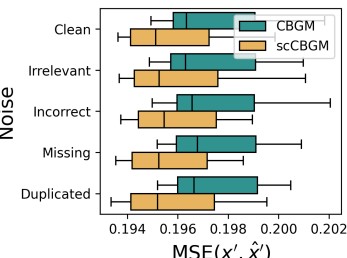

Figure 3: Standard CBGMs vs. scCBGM under different concept annotations noise. MSE between true and predicted counterfactuals across datasets (3), interventions (5), noise levels (3), and seeds (2).

**Results**   We compare standard CBGMs with scCBGM to demonstrate the importance of our architectural modifications for single-cell data. Experiments use three synthetic datasets (20,000 cells, 5,000 genes) varying in technical noise and concept effect size (Appendix C.5). For each dataset, we hold out five concept–cell type pairs, meaning the model never observes these combinations during training, thus emulating a compositional generalization task (Appendix C.6). We also evaluate robustness under four types of concept noise at two corruption levels with four random seeds. Intervention performance is measured by MSE between predicted and true counterfactuals. To ensure fairness, we conducted a hyperparameter sweep (432 configurations) on one noiseless dataset and selected the best model by intervention performance (Appendix C.7). Results are presented in Figure 3. scCBGM outperforms standard CBGM across all scenarios, both noisy and clean. Notably, scCBGM demonstrates greater robustness to noise, with only minimal performance degradation under different noise conditions. Furthermore, as shown in Appendix D.6, scCBGM remains stable as the number of concepts increases, suggesting it can accommodate the complexity of larger biological systems.

**Ablation Study**   To evaluate the impact of our main architectural components, we conducted an ablation study varying three factors: the decoder type (skip vs. direct), the concept head (concept bottleneck vs. concept embedding), and the orthogonality loss (cross-covariance vs. cosine). Experiments were performed on three synthetic datasets, each with five intervention types, two random seeds, and multiple noise settings. The results, summarized in Table S8, indicate that the concept bottleneck (CBM) variant is preferable to the concept embedding (CEM) counterpart on this class of data. Within the CBM model family, incorporating the cross-covariance loss further improves performance. Notably, the skip-connection decoder exhibits a conditional effect – it enhances performance only when paired with both the cross-covariance loss and the concept-bottleneck architecture. This behavior is supported by ablations on real datasets, see Appendix D.8.2.

## 4 RELATED WORKS

scCBGM delivers accurate control over single-cell editing by uniquely integrating decoupled and interpretable representations with counterfactual generation. Existing methods typically address these aspects in isolation: most approaches that learn disentangled representations of single cells do not directly support interventions, while most intervention-based models capture conditional probability distributions without enabling fine-grained editing or counterfactual generation. Below, we review representative methods from both domains.

**Disentangled and interpretable representations for single cell data.** Many methods have recently proposed interpretable latent decompositions for single cell data (Lopez et al., 2018; Svensson et al., 2020; Gut et al., 2021; Seninge et al., 2021; Choi et al., 2023; Lopez et al., 2023; Pouyabahar et al., 2025). Among these, a few combine interpretability with intervention prediction. For example, Celcomen (Megas et al., 2025) leverages disentangled representations to model gene–gene interactions in spatial data and MichiGAN (Yu & Welch, 2021) conditions a generative model on the latent space of a $\beta-$VAE which provides limited biological interpretability. Closer to our work, scDis-InFac (Zhang et al., 2024) learns a disentangled representation of biological conditions and batch effects but assumes mutually independent factors and additive treatment mechanism in the latent space, which are unrealistic in our intended applications. biolord (Piran et al., 2024) incorporates interventions and use a noise injection mechanism for disentanglement. CinemaOT (Dong et al., 2023) first factors the data into causal and spurious factors before predicting treatment effects, but is limited to mapping cells to observations in the training data, restricting its generalizability.

**Single-cell intervention prediction.** Because scRNA-seq measurements cannot be repeated on the same cell, models typically predict intervention effects at the population level. Early approaches modeled temporal dynamics between cell populations (Yeo et al., 2021; Schiebinger et al., 2019), later extended to perturbation-response prediction (Rohbeck et al., 2025; Bunne et al., 2023; Klein et al., 2025). Conditional generative models without explicit dynamics have also been adapted (Palma et al., 2025; Luo et al., 2024; Marouf et al., 2020; Huang et al., 2024). However, these models primarily learn *conditional distributions* and do not support counterfactual inference or cell-level editing. Moreover, many focus on specific perturbation classes such as gene knock-outs (Roohani et al., 2023; Wang et al., 2024a; Littman et al., 2025) or chemical treatments (Qi et al., 2024), whereas our approach supports diverse biological interventions. Methods that attempt cell-level editing, such as scVIDR and scGen (Kana et al., 2023; Lotfollahi et al., 2019), depend on strong assumptions about perturbation mechanisms (e.g., additive latent effects) and do not provide interpretable representations.

## 5 EXPERIMENTS

### 5.1 EXPERIMENT SETUP AND METRICS

We evaluate scCBGM on its ability to perform precise, fine-grained, and identity-preserving edits across three diverse real-world datasets (Kang et al., 2017; Cui et al., 2024; Nault et al., 2023). Since exogenous noise in real-world data cannot be controlled, exact counterfactual evaluation is infeasible. To approximate this, we treat broad cell-type categories as observable concepts (*e.g.*, CD4 T cells) and consider granular subtypes (*e.g.*, *activated* CD4 T cells) as ground-truth counterfactual distributions.

Let $\mathbf{s}$ denote a subtype. We define $\hat{p}_{\mathbf{s},\mathbf{c}}$ as the empirical distribution of cells with subtype $\mathbf{s}$ under concept $\mathbf{c}$. The corresponding ground-truth counterfactual distribution is $\hat{p}_{\mathbf{s},\mathbf{c}'}$, i.e., the distribution of cells with the same subtype but different observable concept values. Since the subtype information is hidden from the model, it can be considered part of the exogenous noise $U$, making this construction a principled proxy for counterfactual evaluation.

Given a counterfactual predictor $f(\mathbf{x}, \mathbf{c}')$, we require the distribution induced by applying $f$ to samples from $\hat{p}_{\mathbf{s},\mathbf{c}}$ to align with $\hat{p}_{\mathbf{s},\mathbf{c}'}$. We assess this alignment using the Maximum Mean Discrepancy ratio (rMMD):

$$\text{rMMD} = \frac{\text{MMD}((f_{\mathbf{c}'})_{\#}\hat{p}_{\mathbf{s},\mathbf{c}}, \hat{p}_{\mathbf{s},\mathbf{c}'})}{\min_{\mathbf{c}} \text{MMD}(\hat{p}_{\mathbf{c}}, \hat{p}_{s,\mathbf{c}'})}. \tag{11}$$

Here, $(f_{\mathbf{c}'})\#\hat{p}_{\mathbf{s},\mathbf{c}}$ denotes the distribution induced by transforming $\hat{p}_{\mathbf{s},\mathbf{c}}$ with $f$, while $\hat{p}_{\mathbf{c}}$ is the empirical distribution of cells under concept $\mathbf{c}$. The denominator minimizes MMD over all observed concepts in the training data, normalizing for baseline misspecification common in single-cell per-

turbation modeling (Wenteler et al., 2024). Intuitively, the numerator measures the discrepancy between predicted and observed counterfactuals at the subtype level, while the denominator reflects the similarity between the target population and its closest match in the training data. An rMMD below 1 indicates success, i.e., the model outperforms the trivial baseline of mapping to the most similar existing population. Analogously, we also evaluate the Frechet Inception Distance ratio and the Sinkhorn Divergence ratio, which are discussed and presented in Appendices D.2 and D.9.

**Baselines.** scCBGM is compared against CBGM, a standard concept bottleneck model (Ismail et al., 2023), as well as single-cell editing baselines: CINEMA-OT (Dong et al., 2023), biolord (Piran et al., 2024), scGen (Lotfollahi et al., 2019), and a standard Conditional Flow Matching model, representing state-of-the-art generative modeling in single-cell analysis (akin to the framework used in CellFlow Klein et al. (2025)). We refer to this baseline as *Vanilla-FM*. A standard Conditional VAE (CVAE) is included as an additional baseline. This architecture represents approaches common in batch correction tasks which achieve disentanglement by implicitly regressing out conditioning variables via the latent. Benchmarks are further extended to scCBGM combined with flow matching models (scCBGM-FM). To verify that scCBGM-FM's performance stems from the structured scCBGM latent space rather than solely the generative capabilities of Flow Matching, scCVAE-FM and biolord-FM are also evaluated; these variants utilize scCVAE and biolord latents, respectively, as conditioning for the Flow Matching model. Finally, for Vanilla-FM, cVAE and scCBGM-FM, two counterfactual strategies—*edit* and *decode*—are evaluated as detailed in Section 2.5.

Complete data processing and model implementation details are provided in Appendix D.

## 5.2 BENCHMARKING SINGLE-CELL EDITING

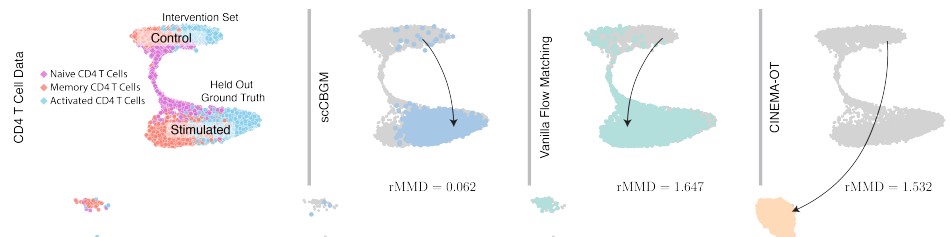

Figure 4: Counterfactual modeling predicts cellular response to perturbation. **Left:** UMAP of CD4 T-cell data from the Kang et al. data, showing Control and Stimulated cells, colored by subtype. Stimulated Naive CD4 T cells were held out during training. **Right:** Control Naive CD4 T cells are edited in silico to predict their stimulated state. scCBGM (second panel) accurately predicts the held-out stimulated Naive CD4 T cells, demonstrating superior zero-shot generalization compared to Vanilla-FM and CINEMA OT.

**Counter-factual modeling predicts cellular response to perturbation** We first benchmarked scCBGM on predicting treatment responses across 14 immune cell subtypes using the Kang et al. (2017) dataset, which contains Peripheral Blood Mononuclear Cells (PBMCs) with and without IFN-$\beta$ stimulation. For each subtype, the stimulation response was treated as an independent benchmark across all models. The model's concepts included 7 high-level cell types and a binary indicator for IFN-$\beta$ stimulation. Table 1 shows that scCBGM-based edits accurately predict the stimulation response while preserving subtype identity, significantly outperforming existing conditional generation methods in counterfactual accuracy on 5 out of the 7 experiments. Additional metrics can be found in Appendix D.9. While Table 1 reports average rMMD across all CD4 T cell subtypes, Figure 4 illustrates this zero-shot generalization capability for a specific subtype. Here, we held out stimulated Naive CD4 T cells during training and evaluated our model's ability to predict their response from control Naive CD4 T cells. We find that scCBGM accurately predicts the held-out stimulated Naive CD4 T cells, demonstrating superior zero-shot generalization compared to baseline methods. A discussion on the challenges of evaluating on real scRNA-seq data, where cell type annotations and stimulation responses may be noisy or partially confounded, can be found in Appendix D.3.

The Cui et al. (2024) immune dictionary dataset measures the response of 17 immune cell subtypes across an expansive panel of 86 cytokine-based stimulations. This setting presents a more challenging scenario, requiring the model to learn distinct responses for each perturbation applied to different cell types. Concepts here included both broad cell type and stimulation identity. The counterfactual task was to predict the response of control cells to a particular stimulation, which the model had not encountered in combination with that specific cell subtype during training. We restricted our

| Model | B cells | T cells (CD4) | T cells (CD8) | Monocytes (FCGR3A) | Monocytes (CD14) | Dendritic | NK cells |
|---|---|---|---|---|---|---|---|
| scCBGM | **0.112 ± 0.028** | **0.169 ± 0.086** | **0.171 ± 0.012** | **1.845 ± 1.776** | 1.309 ± 0.231 | **0.375 ± 0.041** | 1.167 ± 0.271 |
| scCBGM-FM (decode) | 0.106 ± 0.032 | 0.162 ± 0.073 | 0.138 ± 0.040 | 1.141 ± 1.078 | 1.334 ± 0.296 | 0.288 ± 0.061 | **0.093 ± 0.016** |
| scCBGM-FM (edit) | 0.093 ± 0.031 | 0.156 ± 0.066 | 0.119 ± 0.019 | 1.206 ± 1.167 | **1.171 ± 0.390** | 0.231 ± 0.042 | 0.084 ± 0.003 |
| CBGM | 0.902 ± 0.058 | 2.228 ± 0.079 | 1.914 ± 0.049 | 9.514 ± 0.290 | 7.206 ± 0.196 | 1.503 ± 0.039 | 1.270 ± 0.140 |
| Vanilla-FM (decode) | 0.926 ± 0.067 | 1.037 ± 0.197 | 0.912 ± 0.079 | 4.389 ± 4.029 | 0.549 ± 0.240 | 2.250 ± 1.019 | 0.099 ± 0.015 |
| Vanilla-FM (edit) | 0.492 ± 0.174 | 0.487 ± 0.098 | 0.364 ± 0.078 | 2.188 ± 1.848 | 0.394 ± 0.095 | 1.307 ± 0.145 | 0.082 ± 0.021 |
| biolord | 2.622 ± 0.128 | 5.514 ± 1.856 | 4.829 ± 0.680 | 12.123 ± 11.118 | 2.350 ± 0.385 | 3.904 ± 1.480 | 2.355 ± 0.006 |
| biolord-FM | 1.163 ± 0.003 | 2.309 ± 0.003 | 1.762 ± 0.008 | 11.749 ± 0.054 | 10.091 ± 0.071 | 2.191 ± 0.010 | 1.051 ± 0.000 |
| CINEMA-OT | 2.259 ± 0.276 | 7.042 ± 2.520 | 5.362 ± 0.562 | 11.193 ± 9.807 | 10.185 ± 5.194 | 1.367 ± 0.333 | 3.707 ± 0.008 |
| scGen | 1.830 ± 0.397 | 5.117 ± 2.398 | 4.748 ± 0.072 | 7.227 ± 6.673 | 5.748 ± 0.349 | 1.133 ± 0.587 | 2.436 ± 0.072 |
| CVAE | 0.620 ± 0.008 | 1.374 ± 0.008 | 1.043 ± 0.027 | 9.055 ± 0.040 | 6.446 ± 0.067 | 1.134 ± 0.021 | 0.706 ± 0.010 |
| CVAE-FM (decode) | 0.525 ± 0.005 | 1.115 ± 0.010 | 0.901 ± 0.007 | 8.379 ± 0.045 | 6.051 ± 0.076 | 0.982 ± 0.012 | 0.604 ± 0.008 |
| CVAE-FM (edit) | 0.512 ± 0.003 | 1.080 ± 0.007 | 0.847 ± 0.005 | 8.442 ± 0.066 | 5.930 ± 0.067 | 0.987 ± 0.013 | 0.567 ± 0.004 |

Table 1: rMMD per cell group for different models (best, second-best, and **third-best** bolded) in the Kang et al. (2017) dataset.

evaluation to seven cell-type–cytokine pairs previously identified by (Cui et al., 2024) to induce a significant transcriptional shifts relative to controls. As reported in Table 2, scCBGM successfully predicts accurate cellular responses for these unseen combinations on 4 out of the 7 experiments; additional metrics can be found in Appendix D.9.

| Model | T cells (Gamma-delta) | T cells (CD4) | T cells (CD8) | Dendritic (cDC2) | Dendritic (Langerhans) | Myeloid (Macrophages) | Lymphoid (NK cells) |
|---|---|---|---|---|---|---|---|
| scCBGM | 0.304 ± 0.024 | 0.062 ± 0.014 | 0.295 ± 0.038 | 1.877 ± 0.012 | **0.110 ± 0.007** | 2.016 ± 0.318 | 4.695 ± 0.061 |
| scCBGM-FM (decode) | 0.238 ± 0.032 | 0.046 ± 0.011 | 0.259 ± 0.042 | 1.846 ± 0.026 | 0.104 ± 0.005 | 1.845 ± 0.036 | 4.819 ± 0.057 |
| scCBGM-FM (edit) | 0.278 ± 0.017 | 0.034 ± 0.012 | 0.254 ± 0.022 | 1.798 ± 0.039 | 0.097 ± 0.003 | 1.871 ± 0.116 | 4.659 ± 0.132 |
| CBGM | 0.824 ± 0.151 | 0.401 ± 0.113 | 0.389 ± 0.054 | 1.753 ± 0.210 | 0.139 ± 0.009 | 4.518 ± 1.483 | 5.754 ± 0.192 |
| Vanilla-FM (decode) | 1.822 ± 0.264 | 1.647 ± 0.252 | 1.721 ± 0.232 | 1.505 ± 0.102 | 1.609 ± 0.110 | 37.735 ± 6.707 | 2.388 ± 0.818 |
| Vanilla-FM (edit) | 0.540 ± 0.116 | 0.164 ± 0.037 | 0.210 ± 0.062 | 0.679 ± 0.080 | 0.368 ± 0.058 | 30.872 ± 3.428 | 1.171 ± 0.265 |
| biolord | 2.848 ± 0.005 | 2.308 ± 0.006 | 1.934 ± 0.004 | 3.419 ± 0.005 | 2.751 ± 0.003 | 51.931 ± 2.116 | 7.370 ± 0.082 |
| biolord-FM | 1.363 ± 0.004 | 0.894 ± 0.007 | 0.907 ± 0.013 | 1.954 ± 0.005 | 1.020 ± 0.012 | 5.751 ± 0.138 | 5.674 ± 0.060 |
| CINEMA-OT | 2.033 ± 0.004 | 1.532 ± 0.003 | 1.244 ± 0.004 | **1.583 ± 0.002** | 1.504 ± 0.002 | 26.429 ± 0.303 | 11.179 ± 0.091 |
| scGen | 2.069 ± 0.074 | 2.849 ± 0.134 | 1.506 ± 0.083 | 1.989 ± 0.016 | 0.388 ± 0.013 | 8.877 ± 0.544 | 8.435 ± 0.120 |
| CVAE | 0.300 ± 0.007 | 0.049 ± 0.006 | 0.238 ± 0.012 | 1.975 ± 0.007 | 0.127 ± 0.003 | 2.005 ± 0.019 | 4.995 ± 0.043 |
| CVAE-FM (decode) | 0.284 ± 0.006 | **0.043 ± 0.006** | 0.224 ± 0.006 | 1.938 ± 0.004 | 0.121 ± 0.003 | 1.955 ± 0.066 | 4.945 ± 0.039 |
| CVAE-FM (edit) | **0.292 ± 0.003** | 0.036 ± 0.010 | **0.232 ± 0.011** | 1.930 ± 0.001 | 0.117 ± 0.001 | **1.907 ± 0.020** | 4.889 ± 0.034 |

Table 2: rMMD per cell group for different models (best, second-best, and **third-best** bolded) in the Cui et al. (2024) dataset

**Modeling continuous dose responses and hypothesis generation** To assess whether scCBGM can model continuous responses, we used the Nault et al. (2023) dataset, which measures liver responses to varying dosages of TCDD, a toxic dioxin compound. In this setting, concepts consisted of broad cell type and treatment dosage. The counterfactual task was to predict gene expression profiles for intermediate dosages, given that high and low dosages of the test subtypes were withheld during training. As shown in Table 3, scCBGM generally outperforms other methods, thereby helping to fill experimental gaps while supporting mechanistic hypothesis generation. Results on all available cell types are given in Appendix D.9.3.

**Benchmarking on synthetic data** We also benchmarked all methods on the three synthetic datasets introduced in Section 3, where access to true counterfactuals enables fine-grained cell-level rather than population-level evaluation. Flow matching models showed a clear advantage, with scCBGM-FM outperforming all others across datasets (Appendix, Table S5).

**scCBGM boosts performance of flow matching models.** As discussed in Section 2.5, scCBGM can

| Model | Hepatocytes (Centrilobular) | Stellate cell | Hepatocytes (Periportal) |
|---|---|---|---|
| scCBGM | 0.627 ± 0.011 | 0.895 ± 0.548 | **0.752 ± 0.046** |
| scCBGM-FM (decode) | 0.608 ± 0.029 | **0.861 ± 0.440** | 0.719 ± 0.056 |
| scCBGM-FM (edit) | **0.617 ± 0.001** | 0.844 ± 0.451 | 0.708 ± 0.056 |
| Vanilla-FM (decode) | 1.458 ± 0.690 | 10.481 ± 7.058 | 1.185 ± 0.484 |
| Vanilla-FM (edit) | 0.442 ± 0.088 | 7.235 ± 3.239 | 1.030 ± 0.317 |
| biolord | / | 44.340 ± 32.531 | 4.707 ± 2.031 |
| CINEMA-OT | 4.667 ± 1.598 | 45.570 ± 33.289 | 5.295 ± 1.363 |
| scGen | 2.214 ± 0.642 | 12.067 ± 9.021 | 2.389 ± 0.798 |
| CVAE | 1.842 ± 0.809 | 1.349 ± 0.281 | 1.554 ± 0.433 |
| CVAE-FM (decode) | 1.333 ± 0.585 | 0.882 ± 0.117 | 1.143 ± 0.263 |
| CVAE-FM (edit) | 1.337 ± 0.569 | 0.853 ± 0.171 | 1.131 ± 0.253 |

Table 3: rMMD per cell group for different models (best, second-best, and **third-best** bolded) in the Nault et al. (2023) dataset.

be combined with generative models to harness their established generative capabilities. Tables 1, 2, and 3 compare scCBGM-FM directly with Vanilla-FM (*i.e* our implementation of CellFlow).

Whereas Vanilla-FM conditions the vector field only on observed concepts—allowing conditional but not fully counterfactual distributions—scCBGM-FM conditions on the scCBGM embedding, enabling richer counterfactual inference. In our experiments, scCBGM-FM improved over or matches vanilla-FM in 13/17 of the cell types across all datasets. We conjecture that the superior performance of scCBGM-FM on the remaining cases is related to the overlap between the subtype populations. When the treated distributions of the subtypes of the same cell type overlap significantly, learning a conditional distribution of the treated population becomes optimal. While scCBGM should still theoretically match the performance of Vanilla-FM in this case, the limited number of data samples prevents it in practice. We also note that the *edit* procedure is generally much more effective than the *decode* procedure.

### 5.3 CASE STUDY: CONTROLLED SINGLE-CELL EDITING FOR ENHANCED DRUG RESPONSE

Beyond cell-editing, we show that scCBGM offers a framework for guiding cells toward a desired biological state in an interpretable manner. In the original Nault et al. (2023) dataset, stellate cells exhibited only a moderate response to TCDD. With scCBGM, we can identify biological concepts that appear to be crucial for mediating the treatment effect in this specific cell type. To this end, we trained the model with gene regulation pathways extracted from a reference signaling pathway database (Badia-i Mompel et al., 2022; Schubert et al., 2018), represented as soft concepts (*i.e.*, continuous values). By comparing control and treated stellate cells, we uncovered key pathways that were differentially regulated. We then applied scCBGM to edit control cells by jointly activating these pathways while simulating TCDD treatment. As illustrated in Figures 5, S8 & S9, the resulting cell population after editing appeared similar to the populations which responded to the treatment, in both rMMD and gene-expression changes. That is despite the model not being trained on any

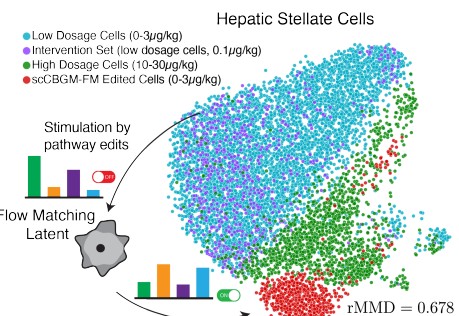

Figure 5: scCBGM enables interpretable control of cells to enhance response to stimulation. Stellate cells with low dosage of TCDD showed limited treatment response, while cells with high dosage showed a clear response. By editing control cells' pathway activity profile, cells become more sensitive to TCDD, showing a treatment response similar to cells with higher TCDD dosage.

treatment data, suggesting a potential mechanistic route for mediating treatment response. These results highlight the utility of scCBGM for testing mechanistic hypotheses and exploring strategies to overcome cellular resistance.

## 6 DISCUSSION AND CONCLUSION

In this work, we introduced scCBGM, a framework for interpretable and precise single-cell counterfactual editing. Our experiments demonstrate improved accuracy, flexibility, and robustness over existing methods, including under noisy annotations.

**Limitations.** A key limitation in our evaluation approach involves assessing counterfactual methods on real single-cell data: true cell-specific counterfactuals cannot be observed, as this would require measuring the same cell under different conditions simultaneously. For real data, we therefore rely on population level metrics as the best available approximation for validating cell level edits. While this approach is standard in the field, it introduces potential pitfall population level accuracy does not guarantee that individual cell predictions preserve biological realism or cell identity at the single-cell level. Our synthetic data evaluation addresses this limitation by providing access to ground-truth counterfactuals under controlled conditions.

**Future Directions.** While the framework relies on concept annotations, which may be incomplete or subjective in practice, future work could explore automated or semi-supervised concept discovery as a mitigation strategy. In addition, next steps could explore combining the VAE and Flow Matching frameworks into a unified model, with end-to-end concept bottleneck encoding and FM-based generation. More broadly, scCBGM moves toward computational representations of cells that support fine-grained, *in silico* experimentation, laying groundwork for predictive models that integrate with and guide biological discovery.

## 7 ETHICS STATEMENT

To the best of our knowledge, this work adheres to the ICLR Code of Ethics. It does not involve human subjects, sensitive data, or dual-use risks. Large language models (LLMs) were used only in a limited way for grammar correction and text checking; they were *not* used to generate results or write any major parts of the text.

## 8 REPRODUCIBILITY STATEMENT

We provide code and notebooks to fully reproduce all results, figures, and tables in a public anonymous GitHub repository: `https://anonymous.4open.science/r/sccbgm/`. All data used in this work are publicly available; for convenience, we also share the processed files through an anonymous OSF repository: `https://osf.io/kfqj8/?view_only=02cfaddc86da47d5b8fca0577628ddf7`.

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

APPENDIX CONTENTS

# Appendix

## A    ADDITIONAL DETAILS ON SINGE-CELL CONCEPT BOTTLENECK GENERATIVE MODELS

### A.1    A VERY SHORT PRIMER ON COUNTERFACTUALS

We posit a structural causal model (SCM) $M = (G, F, P(U))$ with a directed acyclic graph $G$, endogenous variables $V$, exogenous noise $U$, and assignments $F = \{v_i \leftarrow f_i(\text{pa}_i, u_i)\}$.

Intuitively, a counterfactual query asks: *What would an observed variable $X$ have been, had we set a parent variable to a different value, while keeping everything else (including latent background $U$) fixed?*

Formally, counterfactuals are generated by applying the Pearlian do-operator within an SCM, and their values depend on both the structural assignments and the realization of $U$.

**Setup in this paper.**    Let $X$ denote a cell's RNA-seq counts and $C$ a concept variable we wish to intervene on. Let $Z$ collect other endogenous covariates (e.g. other concepts). Then, we can abstract the data generating process as:

$$Z \leftarrow f_Z(\text{pa}_Z, U_Z), \tag{12}$$
$$C \leftarrow f_C(\text{pa}_C, U_C), \tag{13}$$
$$X \leftarrow f_X(Z, C, U_X). \tag{14}$$

**Counterfactual of $X$ under an action on $C$.**    For an observed cell with concept $C = c$, $X_{C=c} = x$, we write the counterfactual value of $X$ if $C$ had been set to $c'$ as

$$X^{\text{cf}} = X_{\text{do}(C:=c')}(U), \tag{15}$$

defined by the standard abduction–action–prediction steps:

1. **Abduction:** infer the posterior $p(u, z \mid X_C = x)$ over latent variables;
2. **Action:** replace $C \leftarrow f_C(\cdot)$ by $C := c'$;
3. **Prediction:** compute $X^{\text{cf}} = f_X(z, c', u_X)$ with $(z, u)$ drawn from the abducted posterior.

Given $X_{C=c} = x$, the probability distribution of the counterfactual is computed as follows:

$$p(x^{\text{cf}} \mid X_{C=c} = x, \text{do}(C := c')) = \int \mathbb{I}[f_X(z, c', u) = x^{\text{cf}}], p(z, u \mid X_{C=c} = x) \, dz \, du. \tag{16}$$

**Ground-truth counterfactuals obtained from our synthetic data generating process**    When the data-generating process is fully known and simulatable, one can hold the exogenous noise $U$ fixed and generate two outcomes with different concept values $C = c$ and $C = c'$. This yields

$$x = f_X(z, c, u_X), \tag{17}$$
$$x^{\text{cf}} = f_X(z, c', u_X), \tag{18}$$

for the same realization of $(z, u_X)$. Such pairs $(x, x^{\text{cf}})$ constitute *true counterfactual examples* and can serve as ground truth for evaluating predictive methods.

## B    CONSISTENCY OF OUR COUNTERFACTUAL PREDICTOR

In this Section, we prove the consistency of our counterfactual estimator. That is, in the infinite data regime, our estimator converges to the expected counterfactual outcome defined in Equation 3, as the following results shows.

**Proposition B.1** (Consistency of scCBGM counterfactuals). *Let $\hat{\theta}_n \in \arg\min_\theta \widehat{L}_n(\theta)$ be any empirical minimizer of the training objective in Equation 9 on $n$ i.i.d. samples $(\mathbf{x}, \mathbf{c})$. For $\mathbf{x}$ fixed and any counterfactual edit $\mathbf{c}'$ in the model's support, we define the learned counterfactual-mean predictor as*

$$g_\theta(\mathbf{x}, \mathbf{c}') := \mathbb{E}_{\mathbf{z} \sim q_\theta(\mathbf{z}|\mathbf{x})} \left[ D_\theta([f_{u,\theta}(\mathbf{z}), \mathbf{c}']) \right]. \tag{19}$$

*Assuming:*

*(A1) (Data-generating SCM & exogeneity) $U \perp C$ with $X = f_X(C, U)$ and the target $\mu_{x,c'} := \mathbb{E}[f_X(\mathbf{c}', U) \mid X = \mathbf{x}]$ well-defined (exists and is finite).*

*(A2) (Realizability) There exists $\theta^\star$ such that the pushforward $\left(D_{\theta^\star}([f_{u,\theta^\star}(\mathbf{z}), \mathbf{c}])\right)_\# q_{\theta^\star}(\mathbf{z} \mid \mathbf{x}) \equiv f_X(\mathbf{c}, U)_\# P(U|X = \mathbf{x})$, for all $\mathbf{c} \sim P(C)$.*

*(A3) (Regularity) The parameter space is compact, The neural networks are bounded and Lipschitz in parameters and inputs. $\widehat{L}_n$ admits near-global minimizers.*

*(A4) (Uniform convergence) $\sup_\theta |\widehat{L}_n(\theta) - L(\theta)| \xrightarrow{p} 0$, where $L(\theta) = \mathbb{E}_{X,C}[\widehat{L}_n(\theta)]$*

*(A5) (Support/positivity) $\mathbf{c}'$ lies in the support of the training distribution of concepts (and in the support of the model).*

*We have:*

$$g_{\hat{\theta}_n}(\mathbf{x}, \mathbf{c}') \xrightarrow{p} \mu_{x,c'}. \tag{20}$$

*That is, the learned counterfactual* mean *converges in probability to the Pearlian counterfactual mean for the same cell $\mathbf{x}$ under the edit $\mathbf{c}'$.*

*Proof Sketch.* The proof follows an empirical minimization argument combined with the abduction-action-prediction formalism of counterfactual inference.

(S1) From the consistency of M estimators (Van der Vaart, 2000), and assumption (A4), any sequence of empirical minimizers $\hat{\theta}_n$ converges in probability to $\theta^\star := \arg\min_\theta L(\theta)$ (or to a set whose $g_\theta$-image equals $g_{\theta^\star}$).

(S2) At $\theta^\star$, (A2) leads $q_{\theta^\star}(\mathbf{z} \mid \mathbf{x}) = p(\mathbf{z} \mid \mathbf{x})$ in the model's latent space and ensures $D_{\theta^\star}([f_{u,\theta^\star}(\mathbf{z}), \mathbf{c}])$ implements the true structural map $f_X(\mathbf{c}, U)$ when $\mathbf{z}$ is drawn from that posterior.

(S3) Let $T(\theta) := g_\theta(\mathbf{x}, \mathbf{c}')$. By (A3), we have that $T$ is continuous at $\theta^\star$, so $T(\hat{\theta}_n) \xrightarrow{p} T(\theta^\star)$.

(S4) By (A2), the pushforward the pushforward $\left(D_{\theta^\star}([f_{u,\theta^\star}(\mathbf{z}), \mathbf{c}])\right)_\# q_{\theta^\star}(\mathbf{z} \mid \mathbf{x})$ matches $f_X(\mathbf{c}, U)_\# P(U|X = \mathbf{x})$. Hence,

$$T(\theta^\star) = \int D_{\theta^\star}([f_{u,\theta^\star}(\mathbf{z}), \mathbf{c}']) \, q_{\theta^\star}(\mathbf{z} \mid \mathbf{x}) \, dz = \int f_X(\mathbf{c}', \mathbf{u}) \, p(\mathbf{u} \mid \mathbf{x}) \, du = \mu_{x,c'}. \tag{21}$$

Combining (S3) and (S4), we have

$$g_{\hat{\theta}_n}(\mathbf{x}, \mathbf{c}') \xrightarrow{p} \mu_{x,c'}. \tag{22}$$

$\square$

The realizability assumption is the crux of the above result. The following proposition shows that minimizing our loss in Equation 9 leads to the correct pushforward.

**Proposition B.2** (Realizability of the pushforward). *Assume:*

*(A1)* **SCM and exogeneity.** *The data follow $X = f_X(C, U)$ with $U \perp C$, so that $f_X(\mathbf{c}, U)_{\#} P(U|X = \mathbf{x})$ is well-defined for all $\mathbf{c}'$.*

*(A2)* **Proper reconstruction and rich classes.** *The reconstruction term in Equation 9 is a proper conditional risk for the law of $X$ given the decoder input (e.g., a correctly specified exponential-family likelihood), and the encoder/decoder classes are rich enough to realize the data distribution. During training, the decoder is conditioned on the* true *c*.

*(A3)* **Concept supervision.** *The concept head is consistent for predicting $\mathbf{c}$ from $\mathbf{x}$ under the chosen loss, so that at the population optimum we predict $\mathbf{c}$ correctly.*

*Let $L(\theta)$ be the population objective corresponding to Equation 9, and let $\theta^{\star} \in \arg\min_{\theta} L(\theta)$. Then for almost every $\mathbf{x}$ and every $\mathbf{c}'$,*

$$\left( D_{\theta^{\star}}([f_{u,\theta^{\star}}(Z), \mathbf{c}']) \right) \# q_{\theta^{\star}}(Z \mid X = \mathbf{x}) = f_X(\mathbf{c}', U)_{\#} P(U|X = \mathbf{x}), \tag{23}$$

*i.e., the learned pushforward (decoder applied to the residual code with $\mathbf{c}'$) matches the true counterfactual kernel in law. Equivalently, for any bounded measurable $\varphi$,*

$$\mathbb{E}\left[ \varphi\left( D_{\theta^{\star}}([f_{u,\theta^{\star}}(Z), \mathbf{c}']) \right) | X = x \right] = \mathbb{E}\left[ \varphi\left( f_X(\mathbf{c}', U) \right) | X = \mathbf{x} \right]. \tag{24}$$

*Proof sketch.* (S1) *(Optimal conditional reconstruction)* From A2, because $-\log p_{\theta}(x \mid \mathbf{z}, \mathbf{c})$ is a strictly proper conditional scoring rule and the model class is rich, the population minimizer must realize the true conditional law of $X$ given $(Z, C)$:

$$p_{\theta^{\star}}(\mathbf{x} \mid Z, C) = p(\mathbf{x} \mid Z, C) \quad \text{a.s.} \tag{25}$$

Equivalently, there exists a measurable decoder $D_{\theta^{\star}}$ such that $X = D_{\theta^{\star}}([f_{u,\theta^{\star}}(Z), C])$ almost surely under the joint law induced by $(X, C)$ and $Z \sim q_{\theta^{\star}}(\mathbf{z} \mid X)$.

(S2) *(Coupling with the structural SCM)* Combine with Step 1: for a.s. $(X, C)$ we can couple $(Z, U)$ given $X = x$ so that

$$D_{\theta^{\star}}([f_{u,\theta^{\star}}(Z), C]) = f_X(C, U) \quad \text{a.s.}$$

This is just matching two a.s. representations of the same $X$ under an appropriate coupling.

(S3) *(Kernel equality in law)* Condition on $X = \mathbf{x}$ and replace $C$ by any counterfactual value $\mathbf{c}'$. By the equality in Step 2 and the coupling of $(Z, U) \mid X = \mathbf{x}$, pushing $q_{\theta^{\star}}(Z \mid X = \mathbf{x})$ through $\mathbf{u} \mapsto D_{\theta^{\star}}([\mathbf{u}, \mathbf{c}'])$ yields the same law as pushing $P(U \mid X = \mathbf{x})$ through $u \mapsto f_X(\mathbf{c}', \mathbf{u})$:

$$\left( D_{\theta^{\star}}([f_{u,\theta^{\star}}(Z), \mathbf{c}']) \right) \# q_{\theta^{\star}}(Z \mid X = \mathbf{x}) = f_X(\mathbf{c}', U)_{\#} P(U|X = \mathbf{x}), \quad \text{a.s.} \tag{26}$$

This is precisely the equality of the learned pushforward with the target counterfactual kernel.

$\square$

Note that Step 3 matches *measures on $X$* (the object we use for counterfactual prediction); it is invariant to internal reparameterizations of the residual code. The cross-cov penalty and independent training are used to *select* residuals $\mathbf{u}$ that do not leak concept information, improving identifiability and stability.

## B.1 DECOUPLING OF KNOWN AND UNKNOWN CONCEPTS THROUGH CROSS-COVARIANCE PENALTY

In Section 2.3 we introduced the cross-covariance loss $\mathcal{L}_{cc}$. Here we provide a theoretical proof for how the (predicted) known concepts $\hat{\mathbf{c}}$ and unknown factors $\mathbf{u}$ become decoupled through the proposed loss. Let:

$$\mathbf{u} = \phi(A\mathbf{z} + \mathbf{a}^0), \qquad \hat{\mathbf{c}} = \psi(B\mathbf{z} + \mathbf{b}^0) \tag{27}$$

where

$$A \in \mathbb{R}^{d_u \times d_z}, \qquad B \in \mathbb{R}^{K \times d_z} \tag{28}$$

are weight matrices, and $\phi$, $\psi$ are element-wise, piecewise $C^1$ monotonically increasing activation functions (e.g., ReLU or sigmoid). The vectors $\mathbf{a}^0$, $\mathbf{b}^0$ are bias terms. We aim to show that using a cross-covariance loss

$$\mathcal{L}_{cc} = \|\mathrm{Cov}(\mathbf{u}, \hat{\mathbf{c}})\|_F^2 \tag{29}$$

encourages decoupling between the representations $\mathbf{u}$ and $\hat{\mathbf{c}}$. Define the Jacobians:

$$J_{\mathbf{u}}(\mathbf{z}) = \underbrace{\mathrm{diag}(\phi'(A\mathbf{z} + \mathbf{a}^0))}_{D_{\mathbf{u}}} A, \qquad J_{\hat{\mathbf{c}}}(\mathbf{z}) = \underbrace{\mathrm{diag}(\psi'(B\mathbf{z} + \mathbf{b}^0))}_{D_{\hat{\mathbf{c}}}} B \tag{30}$$

We define the centered variables $\tilde{\mathbf{u}}$ and $\tilde{\hat{\mathbf{c}}}$ as:

$$\begin{aligned} \tilde{\mathbf{u}}(\mathbf{z}) &= \mathbf{u}(\mathbf{z}) - \mathbb{E}[\mathbf{u}(\mathbf{z})] \\ \tilde{\hat{\mathbf{c}}}(\mathbf{z}) &= \hat{\mathbf{c}}(\mathbf{z}) - \mathbb{E}[\hat{\mathbf{c}}(\mathbf{z})] \end{aligned} \tag{31}$$

If we assume that

$$\mathbb{E}[\mathbf{u}(\mathbf{z})] \approx \mathbf{u}(\mathbb{E}[\mathbf{z}]) \quad , \quad \mathbb{E}[\hat{\mathbf{c}}(\mathbf{z})] \approx \hat{\mathbf{c}}(\mathbb{E}[\mathbf{z}]) \tag{32}$$

*Note: This assumption is true for all regions of the ReLU activation function where leakage is possible (positive domain).*

Next, using a first order Taylor approximation we can linearize around $\boldsymbol{\mu}_{\mathbf{z}}$. Here, we define $\boldsymbol{\mu}_{\mathbf{x}} := \mathbb{E}[\mathbf{x}]$.

$$\tilde{\mathbf{u}}(\mathbf{z}) \approx \mathbf{u}(\mathbf{z}) - \underbrace{\mathbf{u}(\boldsymbol{\mu}_{\mathbf{z}})}_{\approx \boldsymbol{\mu}_{\mathbf{u}}} \approx J_{\mathbf{u}}(\boldsymbol{\mu}_{\mathbf{z}})(\mathbf{z} - \boldsymbol{\mu}_{\mathbf{z}}) \tag{33}$$

$$\tilde{\hat{\mathbf{c}}}(\mathbf{z}) \approx \hat{\mathbf{c}}(\mathbf{z}) - \underbrace{\hat{\mathbf{c}}(\boldsymbol{\mu}_{\mathbf{z}})}_{\approx \boldsymbol{\mu}_{\hat{\mathbf{c}}}} \approx J_{\hat{\mathbf{c}}}(\boldsymbol{\mu}_{\mathbf{z}})(\mathbf{z} - \boldsymbol{\mu}_{\mathbf{z}}) \tag{34}$$

With this we can express the covariance $\mathrm{Cov}(\mathbf{u}, \hat{\mathbf{c}})$ as:

$$\mathrm{Cov}(\mathbf{u}, \hat{\mathbf{c}}) = \mathbb{E}[\tilde{\mathbf{u}}\tilde{\hat{\mathbf{c}}}^T] \approx \mathbb{E}[J_{\mathbf{u}}(\boldsymbol{\mu}_{\mathbf{z}})(\mathbf{z} - \boldsymbol{\mu}_{\mathbf{z}})(\mathbf{z} - \boldsymbol{\mu}_{\mathbf{z}})^T J_{\hat{\mathbf{c}}}(\boldsymbol{\mu}_{\mathbf{z}})^T] \tag{35}$$

Which can be written as:

$$\mathrm{Cov}(\mathbf{u}, \hat{\mathbf{c}}) \approx J_{\mathbf{u}}(\boldsymbol{\mu}_{\mathbf{z}})\Sigma_{\mathbf{z}} J_{\hat{\mathbf{c}}}(\boldsymbol{\mu}_{\mathbf{z}})^T, \quad \Sigma_{\mathbf{z}} := \mathrm{Cov}(\mathbf{z}) \tag{36}$$

And thus minimizing the covariance means that:

$$\mathrm{Cov}(\mathbf{u}, \hat{\mathbf{c}}) \to 0 \quad \Leftrightarrow \quad J_{\mathbf{u}}(\boldsymbol{\mu}_{\mathbf{z}})\Sigma_{\mathbf{z}} J_{\hat{\mathbf{c}}}(\boldsymbol{\mu}_{\mathbf{z}})^T \to 0 \tag{37}$$

Let us now look at what happens when we add some infinitesimal change $\delta\mathbf{z} \sim N(\mathbf{0}, \Sigma_{\mathbf{z}})$ around $\boldsymbol{\mu}_{\mathbf{z}}$:

$$\begin{aligned} \delta\mathbf{u} &= J_{\mathbf{u}}(\boldsymbol{\mu}_{\mathbf{z}})\delta\mathbf{z} \\ \delta\hat{\mathbf{c}} &= J_{\hat{\mathbf{c}}}(\boldsymbol{\mu}_{\mathbf{z}})\delta\mathbf{z} \end{aligned} \tag{38}$$

Then:

$$\mathrm{Cov}(\delta\mathbf{u}, \delta\hat{\mathbf{c}}) = \mathbb{E}[\delta\mathbf{u}\delta\hat{\mathbf{c}}^T] = \mathbb{E}[J_{\mathbf{u}}(\boldsymbol{\mu}_{\mathbf{z}})\delta\mathbf{z}\delta\mathbf{z}^T J_{\hat{\mathbf{c}}}(\boldsymbol{\mu}_{\mathbf{z}})] \tag{39}$$

Which becomes:

$$\mathbb{E}[J_{\mathbf{u}}(\boldsymbol{\mu}_{\mathbf{z}})\delta\mathbf{z}\delta\mathbf{z}^T J_{\hat{\mathbf{c}}}(\boldsymbol{\mu}_{\mathbf{z}})] = J_{\mathbf{u}}(\boldsymbol{\mu}_{\mathbf{z}})\Sigma_{\mathbf{z}} J_{\hat{\mathbf{c}}}(\boldsymbol{\mu}_{\mathbf{z}}) = 0 \tag{40}$$

Hence, there's no first order covariance in the shift of $\mathbf{u}$ and $\hat{\mathbf{c}}$ upon changes to $\mathbf{z}$ along the direction of variance in the data. This concludes the proof that minimizing $\mathcal{L}_{cc}$ enforces local decoupling between $\mathbf{u}$ and $\hat{\mathbf{c}}$.

## C  SYNTHETIC DATA

### C.1  GENERATIVE PROCESS

Our synthetic data generation proccess builds on an overdispersed hierarchical Poisson distribution. For each cell $i$ and gene $j$, the logarithm of the mean expression is modeled as the sum of several structured components. Because these factors are combined in log-space, their effects translate into multiplicative interactions on the original scale. Specifically, the log mean expression ($\lambda_{ij}$) is defined as:

$$\log(\lambda_{ij}) = w_j + l_j + \gamma_{b_ij} + \omega_{u_ij} + \tau_{t_ij} + \sum_{k=1}^{K} v_{ij} c_{kj} + \varepsilon_{ij} \tag{41}$$

In this formulation, gene expression is decomposed into baseline levels ($w_j$), cell-specific scaling ($l_i$), batch effects ($\gamma_{b_ij}$), cell-type variation ($\omega_{u_ij}$), and tissue effects ($\tau_{t_ij}$). In addition, latent concept influences are captured by $\sum_{k=1}^{K} v_{ik} c_{kj}$, where $v_{ik}$ denotes the activation of concept $k$ in cell $i$, and $c_{kj}$ its effect on gene $j$. Residual variation is represented by $\varepsilon_{ij}$.

Gene expression counts are then sampled via the inverse transform method:

$$x_{ij} = Q(\delta_{ij}; \lambda_{ij}) = \inf\{k \in \mathbb{N} : F_{\text{Poi}}(k; \lambda_{ij}) > \delta_{ij}\}, \quad \delta_{ij} \sim \mathcal{U}(0, 1) \tag{42}$$

which is equivalent to drawing directly from the Poisson distribution:

$$x_{ij} \sim \text{Poi}(\lambda_{ij}) \tag{43}$$

Here, $F_{\text{Poi}}$ denotes the cumulative distribution function (CDF) of the Poisson distribution, and $Q$ its percent point function (PPF).

By explicitly modeling the exogenous noise ($\delta_{ij}$), the framework enables the generation of true cell-level counterfactuals following concept-level interventions. To produce the counterfactual $x'_{ij}$, where we change concepts from $c$ to $c'$, we simply do:

$$\log(\lambda'_{ij}) = w_j + l_i + \gamma_{b_ij} + \omega_{u_ij} + \tau_{t_ij} + \sum_{k=1}^{K} v_{ik} \boxed{c'_{kj}} + \varepsilon_{ij}$$
$$x'_{ij} = Q(\delta_{ij}; \lambda'_{ij}) \tag{44}$$

The factors in the model are distributed as follows:

$$
\begin{aligned}
l_i &\sim Norm(0, \sigma_{b_il}) & \text{(library size)} \\
\sigma_b &\sim Unif(r_l, s_l) & \text{(library size std)} \\
w_j &\sim Unif(r_w, s_w) & \text{(baseline expression)} \\
\varepsilon_{ij} &\sim Norm(0, \sigma_\varepsilon) & \text{(noise)} \\
\gamma_{bj} &\sim Norm(0, \sigma_\mathbf{b}) & \text{(batch effect)} \\
\tau_{tj} &\sim Norm(0, \sigma_\mathbf{t}) & \text{(tissue effect)} \\
\omega_{uj} &\sim Norm(0, \sigma_\mathbf{u}) & \text{(cell type effect)} \\
b_i &\sim Cat(\mathbf{p}_i) & \text{(batch id)} \\
p_i &\sim Dir(\alpha_b) & \text{(batch proportions)} \\
t_i &\sim Cat(\mathbf{p_q}) & \text{(tissue id)} \\
p_q &\sim Dir(\alpha_\mathbf{t}) & \text{(tissue proportions)} \\
u_i &\sim Cat(\mathbf{p}_{t_i}) & \text{(cell type id)} \\
p_t &\sim Dir(\alpha_u) & \text{(cell type proportions, in batch)}
\end{aligned}
$$

And the concept-related factors being:

$$
\begin{aligned}
c_{kj} &\sim ZINorm(\pi_k, 0, \sigma_c) &&\text{(concept coefficient)}\\
\pi_k &\sim Beta(\alpha, \beta) &&\text{(zero inflation)}\\
p_u &\sim Dir(\alpha_c) &&\text{(concept probability, in cell type } u\text{)}\\
v_{ik} &\sim Ber(p_{u_i k}) &&\text{(concept indicator)}
\end{aligned}
$$

Here we briefly describe the intuition behind the different terms:

- **library size** $(l_i)$ – regulates the general expression level of cell $i$. This captures how some cells are more transcriptionally active than others.

- **baseline expression** $(w_j)$ – the general expression level of gene $j$. This captures how some genes (e.g., housekeeping genes) tend to have similar expression across multiple cells.

- **batch id** $(b_i)$ – the batch that cell $i$ belongs to.

- **batch effect** $(\gamma_{bj})$ – how gene $j$ is impacted in cells that belong to batch $b$. Batches can be, for example, different assays or collection sites.

- **tissue id** $(t_i)$ – the tissue that cell $i$ belongs to. The tissue distribution is dependent on the batch.

- **tissue effect** $(\tau_{tj})$ – determines how gene $j$ is impacted in cells that belong to tissue $t$. Tissue represents the anatomical region from which the sample was collected.

- **cell type id** $(u_i)$ – the cell type that cell $i$ belongs to. The tissue distribution is dependent on the batch. The cell type distribution is dependent on the tissue.

- **cell type effect** $(\omega_{uj})$ – determines how gene $j$ is impacted in cells that belong to tissue $u$. Cell type represents the phenotypic state of the cell.

- **concept activation** $(v_{ik})$ – the activation of concept $k$ in cell $i$. In the synthetic data, the concepts are binary, hence $v_{ik} \in \{0, 1\}$. Concept activations are dependent on cell type, to mimic how the likelihood of a concept (e.g., biological pathways) being activated differs between cell types.

- **concept coefficients** $(c_{kj})$ – determines how concept $k$ influences gene $j$. We model the concept coefficients with zero inflation to create a sparse concept coefficient matrix. This is similar to how most genes are not affected by a biological pathway being activated.

## C.2 MOTIVATION

scRNA-seq data is derived from a sequencing protocol in which cells are dissociated from tissue, their transcripts are barcoded, and the number of transcripts from each gene is counted. This process is inherently noisy: only a fraction of transcripts in each cell is captured and sequenced. As a result, modeling scRNA-seq data requires accounting for both measurement noise and biological variation.

Let $x_{ij}$ denote the observed number of transcripts (i.e., gene count) for gene $j$ in cell $i$, and let $y_{ij}$ represent the true number of transcripts of gene $j$ present in the cell prior to sequencing.

This process can be viewed as a random sampling step: a fixed number of transcripts $n_i$ are captured from the full pool of $N_i = \sum_j y_{ij}$ transcripts present in cell $i$. Each captured transcript is barcoded and sequenced, resulting in observed counts $x_{ij}$.

The sampling is performed without replacement and can be modeled by a multivariate hypergeometric (MHG) distribution:

$$
\mathbf{x}_i \sim \text{MHG}(\mathbf{t}_i, n_i) \tag{45}
$$

where $\mathbf{t}_i = [t_{i1}, \ldots, t_{iG}]$ denotes the true transcript counts per gene in cell $i$, and $n_i$ is the total number of transcripts captured from that cell. The resulting vector $\mathbf{x}_i$ describes how many transcripts of each gene were observed among the $n_i$ sampled molecules.

Since only a small fraction of all transcripts are captured, we have:

$$\frac{\sum_j x_{ij}}{\sum_j y_{ij}} = \frac{n_i}{N_i} \to 0 \tag{46}$$

where $N_i = \sum_j y_{ij}$ is the true total number of transcripts. Under this low-sampling regime, the MHG distribution can be approximated by a multinomial distribution:

$$\mathbf{x}_i \sim \text{Multinomial}(n_i, \boldsymbol{\pi}_i), \quad \pi_{ij} = \frac{y_{ij}}{N_i} \tag{47}$$

Assuming that the total number of captured transcripts $n_i$ follows a Poisson distribution with cell-specific rate $\eta_i$, we obtain:

$$P(\mathbf{x}_i = \mathbf{k}) = \text{Mult}(\mathbf{k} \mid n_i, \boldsymbol{\pi}_i)\, \text{Poi}(n_i \mid \eta_i) = \prod_j \text{Poi}(k_j \mid \eta_i \pi_{ij}) \tag{48}$$

Hence, the observed count for each gene $j$ in cell $i$ follows a Poisson distribution:

$$x_{ij} \sim \text{Poisson}(\eta_i \pi_{ij}) \tag{49}$$

This model captures measurement noise but assumes fixed proportions $\pi_{ij}$. In reality, gene expression varies across cells due to biological factors such as transcriptional bursting, regulation, or cell state heterogeneity. This leads to overdispersion in the observed counts:

$$\text{Var}(x_{ij}) > \mathbb{E}[x_{ij}]$$

To capture this variability, we treat the Poisson rate $\lambda_{ij} := \eta_i \pi_{ij}$ as a random variable, and model its logarithm as a sum of structured and stochastic components:

$$\begin{aligned} \log(\lambda_{ij}) &= w_j + l_i + \gamma_{b_i j} + \omega_{u_i j} + \tau_{t_i j} + \sum_{k=1}^{|C|} v_{ik} c_{kj} + \varepsilon_{ij} \\ x_{ij} &\sim \text{Poisson}(\lambda_{ij}) \end{aligned} \tag{50}$$

This formulation corresponds to a more general overdispersed Poisson model (e.g., a Poisson GLMM or latent variable model), which flexibly accounts for both technical and biological variability in scRNA-seq data.

### C.3 EXAMPLE OF GENERATED SYNTHETIC DATA

Figure S1 and Figure S2 show an example of synthetic data generated by our model, comprising 20,000 cells and 5,000 genes, with 2 batches, 2 tissue types, 4 cell types and 6 concepts. We let the effects from batch, tissue, cell type, and concepts follow the following impact order: batch > tissue > concept > cell type.

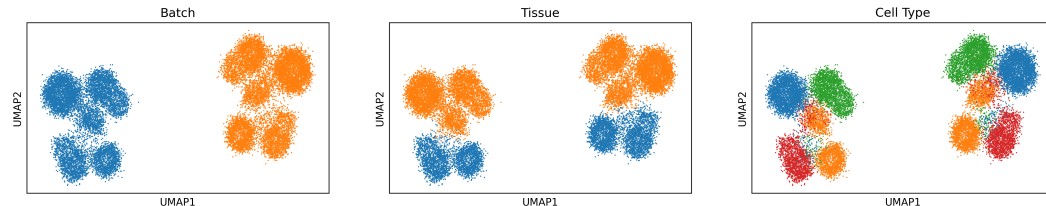

Figure S1: Example of synthetic data with highlighted covariates (batch, tissue, cell type). Each dot is a cell, coloring indicate is cells belonging to the same class (e.g, same batch or cell type). Class colors are subplot-specific and not comparable across panels.

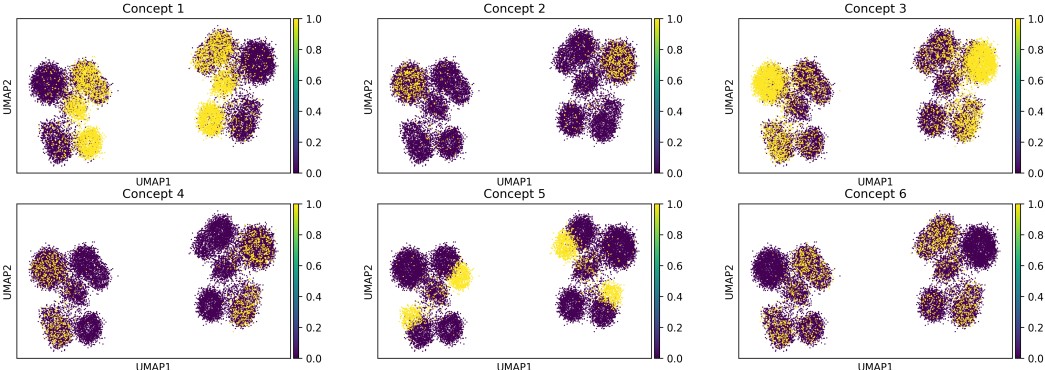

Figure S2: Same synthetic data as in Figure S1 but with highlighted concept activations. Each subplot correspond to one concept.

### C.4  NOISY CONCEPT ANNOTATIONS

To evaluate robustness, we introduce controlled perturbations into the concept annotations, which we refer to as *noisy concepts*. These corruptions mimic realistic imperfections in annotation processes and take the following forms:

- **Missing:** Randomly removes a specified number or fraction of concept columns, simulating important concepts that are not annotated.

- **Irrelevant:** Adds concepts with random on/off values, emulating annotations that capture spurious or uninformative signals.

- **Incorrect:** Selects a subset of existing concepts and independently flips each entry ($0 \leftrightarrow 1$) with probability $p_{\text{noise}}$, mimicking systematic mislabeling at the concept level.

- **Duplicate:** Copies selected concept columns one or more times and appends them, increasing redundancy and inducing stronger—or potentially spurious—correlations between concepts and genes.

In all experiments, we evaluated two noise levels: corruption of 1 or 3 concepts. For the *Missing case*, this meant dropping 1 or 3 concepts; for *Irrelevant*, adding 1 or 3 artificial concepts; for *Incorrect*, randomly flipping a certain percentage of the values in 1 or 3 concepts; and for *Duplicate*, replicating 1 or 3 concepts. The intervened-upon concept was always left unaffected.

### C.5  SYNTHETIC DATASETS

We generated three synthetic datasets with controlled variations in noise and concept effect strength. **Synthetic 1** serves as the baseline with default parameters. **Synthetic 2** increases "technical noise" (std_noise = 0.08), introducing more variability and reducing signal clarity. **Synthetic 3** reduces concept effect variance (std_concept = 0.02), producing sparser and weaker concept-driven influences on gene expression.

| Dataset | std_noise | std_concept | Key difference vs. baseline |
|---|---|---|---|
| **Synthetic 1** | 0.01 | 0.08 | Baseline |
| **Synthetic 2** | 0.08 | 0.08 | More technical noise |
| **Synthetic 3** | 0.01 | 0.02 | Weaker concept effects |

Table S1: Overview of synthetic datasets.

**Shared parameters.** All datasets were generated with **20,000 cells** and **5,000 genes**, across **4 cell types**, **3 tissues**, and **2 batches**, using **5 concepts**. Baseline expression values were sampled uniformly between 1 and 5, with batch, tissue, and cell type effects drawn from Normal distributions with standard deviations 0.06, 0.05, and 0.04, respectively. Library size variation was set between 0.01–0.03, residual noise at 0.01 (except where modified), and concept coefficients from a zero-inflated Normal with standard deviation 0.08 (except where modified). We let $\alpha = 1$, $\beta = 0.8$ in the Beta distribution used to sample $\pi_k$, impacting the sparsity of the genes that a concept influences.

### C.6 INTERVENTION SPLITS FOR EVALUATION

For each dataset, we define 5 intervention tasks by holding out specific concept–cell type pairs, i.e., cases where a given concept is active in a particular cell type. The model is then asked to predict what cells of type $u$ with the concept inactive would look like if it were switched on. This setup emulates a *compositional generalization* scenario. Concretely, let $(u, k)$ denote a held-out pair, where $u$ is the cell type and $k$ the concept of interest:

- Remove all cells of type $u$ with $\mathbf{c}_k = 1$;
- Split all cells of type $u$ with $\mathbf{c}_k = 0$ into two equally sized groups: `train` and `intervene`
- Generate ground-truth counterfactuals for the `intervene` group using Eq. 44, yielding the `test` set
- Assign all remaining cells to the `train` group.

In this way, cells of type $u$ are present in the training data, and cells with $c_k = 1$ are also present, but no instance of type $u$ with $c_k = 1$ is ever observed during training. The model is trained on the `train` group, interventions are applied to the held-out `intervene` group, and predictions are evaluated against the corresponding counterfactuals in the `test` set. Because true counterfactuals are available for each intervened cell, evaluation can be performed at the cell level. The five intervention tasks are created by repeating this procedure across randomly chosen $(u, k)$ pairs to ensure robust assessment.

### C.7 HYPERPARAMETER SWEEP

For each model (scCBGM and CBGM) we performed a grid search over key hyperparameters. We use the dataset "Synthetic 1", with noiseless (clean) concept annotations for the sweep. Specifically, we varied the random seed $\{13, 69\}$, the latent dimension $\{64, 128\}$, hidden dimension $\{128, 256\}$, $\beta \in \{10^{-4}, 5 \times 10^{-5}, 10^{-5}\}$, learning rate $\{5 \times 10^{-4}, 10^{-5}\}$, orthogonality regularization $\{0.05, 0.2, 0.5\}$, and concept regularization $\{0.005, 0.05, 0.1\}$. Other parameters (e.g., bottleneck size = 128, 2 layers) were held fixed. We picked the set of hyperparameters that performed best across all five (synthetic interventions). This resulted in a total of 2160 different runs and a total of 432 hyperparameter sets that were evaluated. The full set of results across all sweeps are displayed in Figure S3.

## D EXPERIMENTAL DETAILS

### D.1 DATA PROCESSING

We benchmarked scCBGM and it's Flow Matching variants against contemporary approaches for single-cell perturbations prediction on 3 datasets with annotated cell-types across varying perturba-

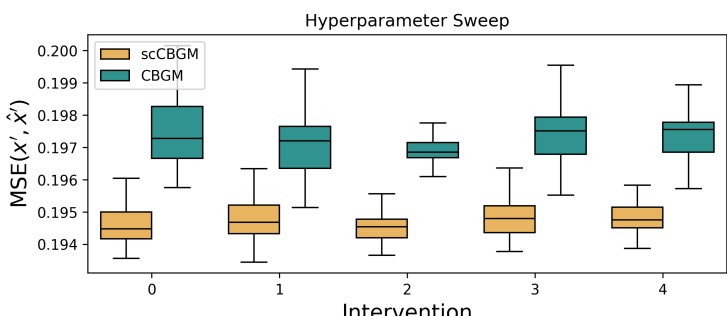

Figure S3: Results from hyperparameter sweep for the scCBGM and CBGM model. Error are cell-level MSEs, comparing the predicted counterfactual and the true counterfactual observation.

tion conditions. Each dataset was minimally processed to fit within our counter-factual framework and to allow for systematic comparisons.

### D.1.1 KANG ET AL. PBMC IFN-$\beta$ STIMULATION DATA

The Kang et al. (2017) dataset comprises 24,264 cells across 8 broad-cell types, observed under two conditions: with and without IFN-$\beta$ stimulation. We excluded megakaryocytes due to their low cell count (210 cells). Data was preprocessed using *scanpy* Wolf et al. (2018), involving median library size normalization, log-transformation of all counts, and filtering to the top 3000 most highly expressed genes.

To benchmark our model on high-fidelity edits that preserve cell phenotype while changing experimental conditions, we focused on identifying granular phenotypes consistent across stimulated and unstimulated cells. We first integrated the two conditions into a unified latent space using Harmony Korsunsky et al. (2019). Within this unified latent space, we applied Leiden clustering Traag et al. (2019) independently to each broad-cell type to discover direct subtypes. This process yielded a total of 14 subtypes, each approximately evenly distributed across the two experimental conditions, see Table S3. For the Kang et al. experiments, concepts were defined using these original broad-cell types, while stimulation predictions and held-out validations were conducted at the more granular subtype level.

### D.1.2 CUI ET AL. IMMUNE DICTIONARY DATA

We further evaluated scCBGM using the Immune Dictionary dataset by Cui et al. (2024). This dataset comprises 110,378 cells, encompassing over 17 distinct immune cell types stimulated with 86 different cytokines in vivo, providing a rich map of cellular responses to immune perturbations. For preprocessing, similar to the Kang et al. dataset, we applied median library size normalization, log-transformation of counts, and filtered to the top 3000 most highly expressed genes.

Unlike the Kang et al. data where we derived subtypes, the Cui et al. dataset already provides 17 granular immune cell type labels. For this dataset, we created broader concept groupings by pooling these original labels into seven aggregate cell types: Stromal cells, Granulocytes, Monocyte-Macrophages (referred to as myeloid in the tables), Dendritic cells, Innate lymphoid cells (referred to as lymphoid in the tables), B cells, and T cells. These broader categories served as our primary cell type concepts. Additionally, concepts were created for each cytokine stimulation. Consistent with our approach for the Kang et al. data, stimulation predictions and held-out validations were performed at the original, granular 17-subtype level.

Given the vast number of possible cytokine-cell type combinations (86 cytokines across 17 cell types), we focused our evaluation on a select set of seven cell-type - cytokine pairs. These pairs were chosen based on the Immune Dictionary paper's findings, representing combinations shown to induce a significant transcriptional shift compared to control conditions. The specific intervention pairs evaluated were (cell type - cytokine):

Table S2: Overview of cell types and subtypes with instance counts per condition. Note the sparsity in CD14+ Monocytes 1 (ctrl).

| Broad | Subtype | Condition | Instances |
|---|---|---|---|
| B cells | B cells 0 | ctrl | 1105 |
| | B cells 0 | stim | 1044 |
| | B cells 1 | ctrl | 201 |
| | B cells 1 | stim | 187 |
| CD14+ Monocytes | CD14+ Monocytes 0 | ctrl | 2764 |
| | CD14+ Monocytes 0 | stim | 2023 |
| | CD14+ Monocytes 1 | ctrl | 9 |
| | CD14+ Monocytes 1 | stim | 497 |
| CD4 T cells | CD4 T cells 0 | ctrl | 2743 |
| | CD4 T cells 0 | stim | 2743 |
| | CD4 T cells 1 | ctrl | 1873 |
| | CD4 T cells 1 | stim | 1830 |
| | CD4 T cells 2 | ctrl | 538 |
| | CD4 T cells 2 | stim | 518 |
| CD8 T cells | CD8 T cells 0 | ctrl | 650 |
| | CD8 T cells 0 | stim | 607 |
| | CD8 T cells 1 | ctrl | 432 |
| | CD8 T cells 1 | stim | 333 |
| Dendritic cells | Dendritic cells 0 | ctrl | 159 |
| | Dendritic cells 0 | stim | 126 |
| | Dendritic cells 1 | ctrl | 44 |
| | Dendritic cells 1 | stim | 98 |
| FCGR3A+ Monocytes | FCGR3A+ Monocytes 0 | ctrl | 431 |
| | FCGR3A+ Monocytes 0 | stim | 556 |
| | FCGR3A+ Monocytes 1 | ctrl | 305 |
| | FCGR3A+ Monocytes 1 | stim | 276 |
| NK cells | NK cells 0 | ctrl | 915 |
| | NK cells 0 | stim | 1047 |

Table S3: Overview of the cell types and subtypes in the Kang et al. (2017) PBMC dataset.

- $\gamma\delta$ T cells - IL17E

- CD4 T cells - TGF$\beta$

- CD8 T cells - TNF$\alpha$

- Conventional dendritic cells (cDC) 2 - INF$\alpha$1

- Langerhans - IFN$\gamma$

- Macrophages - M-CSF

- NK cells - IL15

### D.1.3 NAULT ET AL. LIVER DOSAGES RESPONSE DATASET

To assess scCBGM's capacity for modeling continuous responses and its utility in cellular control, we utilized the Nault et al. dataset Nault et al. (2023). This dataset comprises 131,613 cells across 11 granular cell types of liver responses to varying dosages of TCDD, a toxic dioxin compound, ranging from 0$\mu$g/kg up to 30$\mu$g/kg. Preprocessing followed the same procedure as the other datasets: median library size normalization, log-transformation of counts, and filtering to the top 3000 most highly expressed genes.

We conducted two primary experiments using this dataset:

**Continuous Dosage Prediction** For this experiment, concepts were defined to include broad cell types and TCDD dosage. The 11 original cell types were grouped into four broad categories: Hepatocyte, Immune cells, Cholangiocyte, and Stromal, which served as hard concepts. TCDD dosage was treated as a soft (continuous) concept and its true values were log-transformed and then scaled to be between 0 and 1. The counterfactual task involved predicting gene expression profiles for intermediate TCDD dosages, specifically holding out the 3 and $10\mu$g/kg dosed test cell subtypes during training. This setup enabled benchmarking scCBGM's ability to fill experimental gaps by inferring responses at unobserved dosage levels.

**Interpretable Cellular Control via Pathway Activation** This experiment demonstrated scCBGM's application in controlling cells towards a desired biological state, particularly enhancing treatment effects in specific cell types. We focused on hepatic stellate cells, which showed only a moderate response to TCDD in the original dataset (i.e., minimal effect at lower dosages, unlike hepatocytes, but responding at 10 and $30\mu$g/kg).

First, we computed gene-set loadings for PROGENy pathways Schubert et al. (2018) using the *decoupler* package Badia-i Mompel et al. (2022). By comparing responder hepatic stellate cells (dosed at 10 and $30\mu$g/kg) with non-responders (all other dosages), we identified differentially regulated pathways. Specifically, responder stellate cells exhibited high activity in TGF$\beta$, PI3K, and p53 pathways, and low activity in Trail and VEGF pathways.

For the editing task, all responder hepatic stellate cells were explicitly removed from the training data, meaning the model was unaware that these cells could respond to TCDD. We then performed in silico stimulation on control stellate cells by jointly modifying the soft concepts corresponding to the identified pathways (increasing TGF$\beta$, PI3K, p53, and decreasing Trail, VEGF), while simultaneously activating the TCDD dosage concept. The resulting edited cell population displayed markedly increased sensitivity to the compound, validating scCBGM's ability to identify and modulate mechanistic routes for treatment response.

### D.2 EVALUATION METRICS

We compare performance of different models using the following metrics: the Maximum Mean Discrepancy ratio (rMMD), the Frechet Inception Distance ratio (rFID), and the Sinkhorn Divergence ratio (rSD). We give a definition of these metrics below. Complete benchmarking results with these metrics are provided in Appendix D.9.

**Notations** We define $\hat{p}_{\mathbf{c}}^*$ as the true empirical distribution of an initial cell population of interest with concept $\mathbf{c}$, $\hat{p}_{\mathbf{c}}$ for the empirical distribution of the population of *all* cells with with concept $\mathbf{c}$, $\hat{p}_{\mathbf{c}\to\mathbf{c}'}^*$ the distribution of the target counterfactual cell population of interest with concept $\mathbf{c}'$. We write $\hat{p}_{\mathbf{c}\to\mathbf{c}'}^*$ for the *predicted* counterfactual distribution.

The ratio formulation normalizes the performance of the model's prediction against the best achievable (lowest) metric among all cell populations, providing a scale-independent measure of relative quality.

**rMMD** The Maximum Mean Discrepancy ratio is defined as

$$\text{rMMD} = \frac{\text{MMD}(\hat{p}_{\mathbf{c}\to\mathbf{c}'}^*, \hat{p}_{\mathbf{c}\to\mathbf{c}'}^*)}{\min_\gamma \text{MMD}(\hat{p}_\gamma, \hat{p}_{\mathbf{c}\to\mathbf{c}'}^*)}. \tag{51}$$

where rMMD is defined as the *squared* maximum mean discrepancy with kernel $k$:

$$\text{MMD}(p, q) = \mathbb{E}_p[k(X, X)] - 2\mathbb{E}_{p,q}[k(X, Y)] + \mathbb{E}_q[k(Y, Y)]. \tag{52}$$

In our experiments, we used the RBF kernel.

**rFID**  The Frechet Inception Distance ratio (rFID) is defined as

$$\text{rFID} = \frac{\text{FID}(\hat{p}^*_{\mathbf{c}\to\mathbf{c'}}, \hat{p}^*_{\mathbf{c}\to\mathbf{c'}})}{\min_\gamma \text{FID}(\hat{p}_\gamma, \hat{p}^*_{\mathbf{c}\to\mathbf{c'}})}. \tag{53}$$

The Fréchet Inception Distance (FID) measures the difference between two multivariate Gaussian distributions fitted to the input empirical distributions. For two distributions $p$ and $q$ with means $\mu_p, \mu_q$ and covariances $\Sigma_p, \Sigma_q$, it is given by

$$\text{FID}(p,q) = \|\mu_p - \mu_q\|_2^2 + \text{Tr}\left(\Sigma_p + \Sigma_q - 2(\Sigma_p\Sigma_q)^{1/2}\right). \tag{54}$$

**rSD**  The Sinkhorn Divergence ratio (rSD) is defined as

$$\text{rSD} = \frac{\text{SD}_\varepsilon(\hat{p}^*_{\mathbf{c}\to\mathbf{c'}}, \hat{p}^*_{\mathbf{c}\to\mathbf{c'}})}{\min_\gamma \text{SD}_\varepsilon(\hat{p}_\gamma, \hat{p}^*_{\mathbf{c}\to\mathbf{c'}})}. \tag{55}$$

The Sinkhorn Divergence $\text{SD}_\varepsilon(p,q)$ is an entropically regularized optimal transport divergence between two distributions $p$ and $q$:

$$\text{SD}_\varepsilon(p,q) = W^2_{2,\varepsilon}(p,q) - \tfrac{1}{2}\left(W^2_{2,\varepsilon}(p,p) + W^2_{2,\varepsilon}(q,q)\right), \tag{56}$$

where $W_{2,\varepsilon}$ denotes the entropically regularized Wasserstein-2 distance with regularization strength $\varepsilon$. The regularization smooths the optimal transport problem by adding an entropy term, which makes computation more tractable. As $\varepsilon \to 0$, $\text{SD}_\varepsilon$ converges to the squared Wasserstein distance $W_2^2$, while for large $\varepsilon$, it approaches the squared Maximum Mean Discrepancy (MMD) (Feydy et al., 2019).

### D.3  Challenges of evaluation on real data

The use of real scRNA-seq data is essential to assess how our model performs under realistic conditions — i.e., on data of the kind it is ultimately intended to be applied to. However, we emphasize that such datasets lack definitive ground truth labels. In our experiments, we use *cell type* and *stimulation* annotations both as concepts for intervention and as targets for evaluation in our ratio-based metrics. These labels, however, are not perfectly accurate: cell type annotations are inherently noisy and error-prone, and even for stimulation conditions, cells may have been exposed to the treatment but failed to exhibit a transcriptional response. To illustrate this, we analyze the Kang *et al.* dataset.

As shown in Figure S4, several annotated cell types exhibit strong overlap in the embedding space. A prominent example is the overlap between NK cells and CD8 T cells. To quantify this, we performed a 5-fold cross-validation experiment where we trained a linear classifier (logistic regression) to discriminate between pairs of cell types in PCA space and evaluated the held-out AUROC. Several pairs yield AUROC values close to 0.5, indicating that these classes are not linearly separable—either because the underlying cell states are biologically similar or because the annotations themselves are noisy or partially incorrect, see Table S4. This also explains why sometimes the ratio-based metrics sometimes exceed one.

### D.4  Neural Architectures and training parameters

This section details the neural network architectures, training hyper-parameters, and general experimental settings for scCBGM and the Flow Matching (FM) models used in our experiments. When possible for each model, gene expression data was reduced to 128 principal components (PCs), which were computed exclusively from the training data to prevent data leakage.

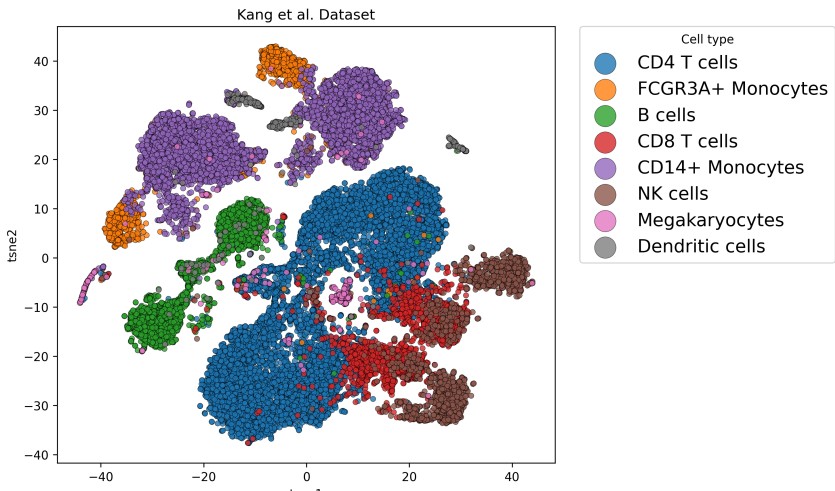

Figure S4: t-SNE of the Kang *et al.* dataset using the original cell type labels. Each point corresponds to a single cell, colored by its annotated cell type.

|  | B cells | CD4 T cells | CD8 T cells | CD14+ Monocytes | Dendritic cells | FCGR3A+ Monocytes | Megakaryocytes | NK cells |
|---|---|---|---|---|---|---|---|---|
| B cells | - | 0.992 | 0.989 | 0.996 | 0.947 | 0.984 | 0.846 | 0.987 |
| CD4 T cells | 0.992 | - | 0.560 | 0.999 | 0.992 | 0.998 | 0.773 | 0.590 |
| CD8 T cells | 0.989 | 0.560 | - | 0.999 | 0.988 | 0.996 | 0.702 | 0.624 |
| CD14+ Monocytes | 0.996 | 0.999 | 0.999 | - | 0.548 | 0.607 | 0.862 | 0.997 |
| Dendritic cells | 0.947 | 0.992 | 0.988 | 0.548 | - | 0.624 | 0.863 | 0.986 |
| FCGR3A+ Monocytes | 0.984 | 0.998 | 0.996 | 0.607 | 0.624 | - | 0.869 | 0.995 |
| Megakaryocytes | 0.846 | 0.773 | 0.702 | 0.862 | 0.863 | 0.869 | - | 0.754 |
| NK cells | 0.987 | 0.590 | 0.624 | 0.997 | 0.986 | 0.995 | 0.754 | - |

Table S4: Average AUROC from 5-fold cross-validation using a linear classifier trained to distinguish cell types based on their PCA representations.

**scCBGM Model**   The scCBGM encoder consisted of $4$ hidden layers, each with $1024$ neurons. These layers incorporated residual connections, layer normalization, and dropout ($p = 0.1$). The encoder mapped gene expression (represented by 128 PCs) to a latent space of $2 \times 128$ dimensions (128 for $\mu$ and for $\sigma$). From the sampled latent variable $z \sim \mathcal{N}(\mu, \sigma)$, the concept bottleneck layers produced the known concepts (whose dimensions varied by experiment) and a default of $128$ unknown concepts. The decoder mirrored the encoder's structure with 4 layers of $1024$ neurons, layer normalization, and dropout. It utilized residual connections for processing unknown concepts and direct skip connections for known concepts, as described in the main text.

scCBGM was trained for 200 epochs with a batch size of 128. Optimization was performed using Adam, starting with a learning rate of $5 \cdot 10^{-4}$ and an exponential learning rate scheduler with a decay rate of $0.997$. The loss function coefficients were set as follows: concept loss at $0.1$, orthogonality loss (for cross-covariance) at $0.5$, and the KL divergence for $\mu$ and $\sigma$ at $1 \cdot 10^{-5}$.

**Flow Matching (FM) Models**   Both scCBGM-FM and Vanilla FM models employed a 4-layer fully-connected network, each layer comprising 1024 neurons. Similar to scCBGM, these networks included residual connections, layer normalization, and dropout. Covariates, which were either scCBGM's known and unknown concepts (for scCBGM-FM) or the ground-truth known concepts (for Vanilla FM), were fed into the model via a learned embedding layer designed to match the hidden dimension of 1024. All FM models were trained for 200 epochs with a batch size of 128. Optimization used Adam, with an initial learning rate of $3 \cdot 10^{-4}$ and an exponential learning rate scheduler with a decay rate of $0.997$. Classifier-free guidance was not utilized, as no significant performance benefit was observed.

**Baseline Models**   We also implemented a conditional VAE (cVAE) for ablation purposes. To ensure a fair comparison, these models utilized the same backbone architecture and training hyperparameters as the CBGM model described above. All other external baseline models, including

biolord, CINEMTA-OT, and scGen, were trained using their respective default parameters as specified in their original implementations.

**Reconstruction Validation** We further validated the ability of these architectures to faithfully reconstruct single-cell data. As shown in Figure S5, scCBGM and scCBGM-FM reliably decode the dataset, preserving both global structure and local density. In contrast, Vanilla-FM exhibits poor reconstruction fidelity, demonstrating the limitations of conditioning solely on coarse labels compared to our granular latent approach. For Vanilla-FM, reconstruction was evaluated by generating a sample from random noise conditioned on the cell's observed concepts and comparing it to the original expression profile. As expected, since this model lacks access to cell-specific identity, it yields high MSE despite generating biologically plausible samples. This baseline highlights the necessity of the scCBGM latent representation for preserving individual cell identity during the encoding-decoding process.

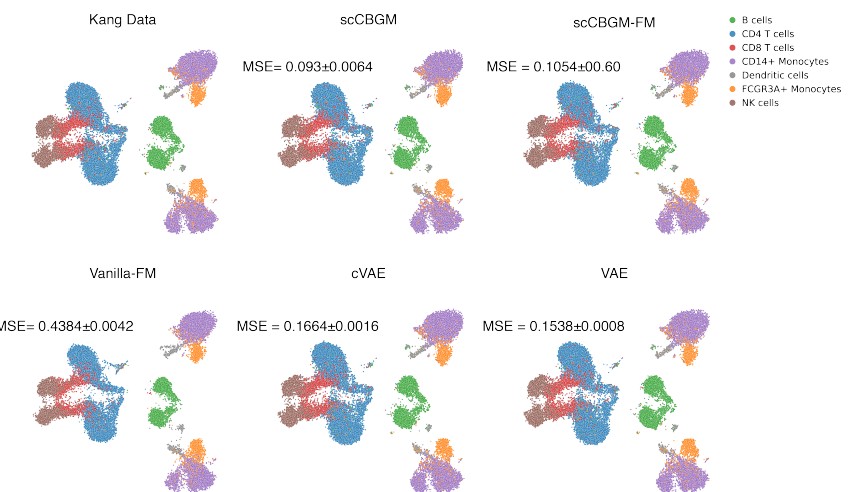

Figure S5: **Qualitative and quantitative comparison of data reconstruction.** UMAP visualizations of the Kang dataset (top-left) and reconstructions from five generative models. Points are colored by cell type. scCBGM and scCBGM-FM successfully reproduce the data distribution with low test-set MSE. Vanilla-FM shows poor reconstruction fidelity due to a lack of granular conditioning, while cVAE and VAE achieve moderate accuracy. MSE values are means and standard deviations over 5 random initialization seeds for each model.

## D.5 BENCHMARK ON SYNTHETIC DATA

For the synthetic data benchmark, we used the same architectural design and hyperparameters as described in Appendix D.4. We ran the experiments on the three datasets outlined in Appendix C.5, each with 5 interventions and 4 random seeds.

|  | Synthetic 1 | Synthetic 2 | Synthetic 3 |
|---|---|---|---|
| scCBGM | $0.0388\pm0.0053$ | $0.0318\pm0.0071$ | $0.0407\pm0.0027$ |
| scCBGM-FM (edit) | $\mathbf{0.0025\pm0.0016}$ | $\mathbf{0.0028\pm0.0014}$ | $\mathbf{0.0039\pm0.0023}$ |
| scCBGM-FM (decode) | $0.0539\pm0.0039$ | $0.0444\pm0.0082$ | $0.0594\pm0.0092$ |
| Vanilla-FM (edit) | $0.0031\pm0.0014$ | $0.0049\pm0.0016$ | $0.0049\pm0.0027$ |
| Vanilla-FM (decode) | $0.0756\pm0.0014$ | $0.0769\pm0.0014$ | $0.0773\pm0.0041$ |
| CINEMA-OT | $0.0390\pm0.0025$ | $0.0393\pm0.0012$ | $0.0400\pm0.0034$ |
| biolord | $0.0402\pm0.0014$ | $0.0411\pm0.0021$ | $0.0428\pm0.0021$ |

Table S5: Comparison of methods on synthetic data. We evaluate in PCA space to account for different normalization strategies, so scCBGM values differ from Table S8. Values are MSE (mean ± std), averaged over interventions (5) and seeds (4).

## D.6 PERFORMANCE AND NUMBER OF CONCEPTS

While only the concepts on which one intends to *intervene* need to be explicitly characterized in the *known concept layer*, the total number of concepts can increase rapidly when modeling complex biological systems. To further assess how scCBGM scales with the number of concepts, we generated three additional synthetic datasets using the same hyperparameters as *Synthetic 1* (see Appendix C.5), varying only the number of concepts. This setup enables a controlled evaluation of how concept dimensionality influences model performance.

As shown in Table S6, performance decreases only marginally—by less than 2%—as the number of concepts increases. This trend is consistent with prior observations in the literature Ismail et al. (2025), indicating that the model remains stable and effective even as the concept space expands.

| concepts | MSE |
|---|---|
| 5 | $0.19617 \pm 0.00195$ |
| 20 | $0.19597 \pm 0.00089$ |
| 100 | $0.19846 \pm 0.00081$ |
| 250 | $0.19809 \pm 0.00053$ |

Table S6: Performance under varying numbers of concepts. Values are reported as mean $\pm$ standard deviation, computed across three random seeds and five interventions. No noise was introduced in the concept annotations.

### D.7 CONCEPT LEAKAGE

The **cross-covariance loss** between the known and unknown concepts is introduced to prevent information leakage between the two layers. This effect is supported by our results in Section 5 and the theoretical justification in Appendix B.1. To examine this question more directly, we designed an experiment that mimics a realistic scenario in which relevant concepts are *not* included among the known ones and must instead be captured by the unknown layer, followed by an examination of how intervening on the known concepts impacted the state of the unknown ones.

In this experiment, we used the **Kang et al.** dataset (Appendix D.1.1). Only the *binary stimulation concept* was included in the model — i.e., cell-type information was *not* encoded in the known layer. The data was split into a 60/40 train/test partition. We trained scCBGM with the single stimulation concept, along with two separate classifiers: (i) a one-vs-all classifier predicting **cell type** from the learned expression representation, and (ii) a binary classifier predicting whether a cell was **stimulated** or **control**.

For evaluation, we took the test data and, within each cell type, *intervened* on all unstimulated (control) cells by setting their stimulation state to "on." We then applied the two classifiers—trained on the original training data—to predict cell type and stimulation status for each cell *before* and *after* the intervention.

| | f(cell type) | | f(stim) | |
|---|---|---|---|---|
| | Before | After | Before | After |
| B cells | 0.985 | 0.975 | 0.004 | 1.000 |
| CD14+ Monocytes | 0.974 | 0.955 | 0.002 | 1.000 |
| CD4 T cells | 0.984 | 0.979 | 0.009 | 1.000 |
| CD8 T cells | 0.714 | 0.630 | 0.014 | 1.000 |
| FCGR3A+ Monocytes | 0.743 | 0.794 | 0.000 | 1.000 |
| NK cells | 0.893 | 0.959 | 0.022 | 1.000 |

Table S7: Results from classifiers trained to predict cell type and stimuli applied to the test data before and after intervention. Values are reported as mean across all cells in the test data.

If concept leakage has been effectively prevented, the **stimulation predictions** should switch from 0 to 1, while the **cell-type predictions** should remain stable. This behavior is clearly observed in Figure S6, where the stimulation scores shift from low to high after intervention, whereas the cell-type scores remain largely unchanged. The corresponding quantitative results are summarized in Table S7.

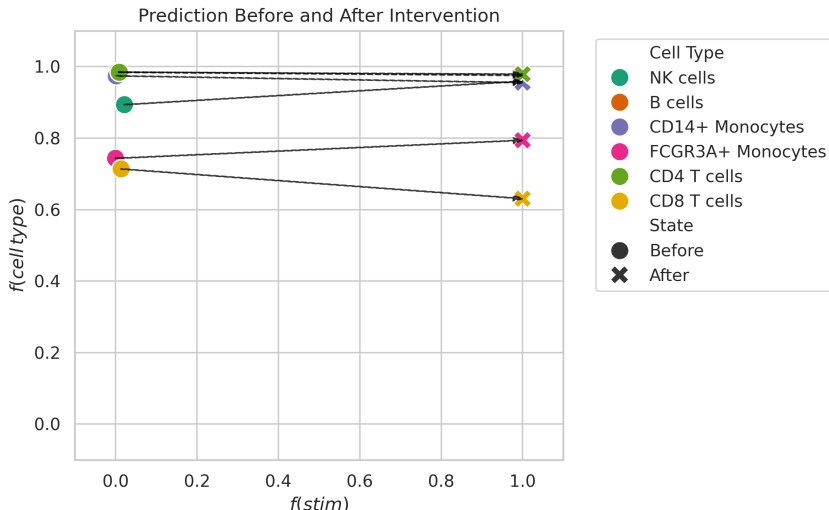

Figure S6: Results from classifiers trained to predict cell type and stimuli applied to the test data before and after intervention. Values are reported as mean across all cells in the test data. Each color represents a cell type. Good results are straight lines from left to right.

## D.8 ABLATIONS

### D.8.1 ARCHITECTURE ABLATIONS SYNTHETIC DATA

| Model | Bottleneck | Skip | $\mathcal{L}$ | MSE |
|---|---|---|---|---|
| | CBM | ✓ | $\mathcal{L}_{cosine}$ | 0.1989±0.00023 |
| | CEM | ✓ | $\mathcal{L}_{cc}$ | 0.19865±0.00023 |
| | CEM | ✓ | $\mathcal{L}_{cosine}$ | 0.19816±0.00022 |
| CBGM | CEM | ✗ | $\mathcal{L}_{cc}$ | 0.1981±0.0002 |
| | CEM | ✗ | $\mathcal{L}_{cosine}$ | 0.19804±0.00021 |
| | CBM | ✗ | $\mathcal{L}_{cosine}$ | 0.19791±0.00021 |
| | CBM | ✗ | $\mathcal{L}_{cc}$ | 0.19736±0.0002 |
| **scCBGM** | **CBM** | ✓ | $\mathcal{L}_{cc}$ | **0.19655±0.00019** |

Table S8: Ablation study over the skip connections and cross-covariance loss. MSE between true and predicted counterfactuals, averaged over datasets (3), interventions (3), noise levels (2), types of noise (5), and seeds (2). We report values as Mean ± SEM.

We compare the contribution of each individual component across a $2 \times 2 \times 2$ design space of Bottleneck (CEM vs. CBM), Skip Connection (yes or no), Loss (cross-covariance vs. cosine). On the synthetic data we are able to do cell-level evaluation, something that real data does not permit. We evaluate each model across: 3 different datasets, 5 different types of interventions (in each dataset), 2 random seeds, 5 different types of noise, and 2 different levels of noise. For comparable results, we used the exact same hyperparameters across all configurations except for the ablated components. However, we highlight that the models using a cosine loss are restricted to using the same number of known as unknown concepts, a limit not imposed on the cross-covariance loss.

### D.8.2 ARCHITECTURE ABLATIONS ON REAL DATA

In Table S9, we report the result of our model architecture ablations on the Kang et al. (2017) dataset. This experiment confirms the need for the disentanglement loss and the skip connection in scCBGM.

Table S9: rMMD over all subtypes on the Kang et al. (2017) dataset, for different ablations of scCBGM ordered mean in descending order. We evaluate different choices of bottleneck architecture, the presence of skip connection and different disentanglement losses (cosine similarity, cross-covariance, or no disentanglement loss (left blank)). We report values as Mean ± SEM, and report the average rMMD over all subtypes for each method (*Mean* column). Best model for each subtype is bolded. The scCBGM and CBGM configurations used in our benchmarks are highlighted in the *Method* column.

| Method | Bottleneck | Skip | $\mathcal{L}$ | B Cells subtype 0 | subtype 1 | Cd14 subtype 0 | subtype 1 | Cd4 subtype 0 | subtype 1 | subtype 2 | Cd8 subtype 0 | subtype 1 | Dendritic subtype 0 | subtype 1 | Fcgr3A subtype 0 | subtype 1 | Nk subtype 0 | Mean |
|---|---|---|---|---|---|---|---|---|---|---|---|---|---|---|---|---|---|---|
| | CEM | ✓ | $\mathcal{L}_{cc}$ | 2.859 ± 0.560 | 1.484 ± 0.348 | 11.505 ± 1.676 | 9.367 ± 0.917 | 4.294 ± 0.864 | 6.938 ± 2.221 | 2.264 ± 0.511 | 2.575 ± 0.174 | 5.936 ± 1.966 | 2.159 ± 0.244 | 1.779 ± 0.164 | 1.663 ± 0.073 | 24.590 ± 1.246 | 1.911 ± 0.094 | 5.666 |
| | CEM | ✓ | $\mathcal{L}_{cosine}$ | 1.456 ± 0.266 | 1.154 ± 0.143 | 8.753 ± 0.244 | 8.449 ± 0.728 | 3.103 ± 0.031 | 4.401 ± 0.609 | 1.178 ± 0.125 | 2.604 ± 0.169 | 2.676 ± 0.350 | 1.133 ± 0.163 | 1.603 ± 0.164 | 1.777 ± 0.087 | 20.765 ± 1.889 | 1.917 ± 0.517 | 4.355 |
| | CEM | ✓ | | 1.257 ± 0.063 | 0.930 ± 0.126 | 9.026 ± 0.429 | 7.466 ± 0.548 | 2.674 ± 0.441 | 3.970 ± 0.246 | 1.133 ± 0.032 | 2.271 ± 0.085 | 2.434 ± 0.275 | 1.394 ± 0.120 | 1.762 ± 0.244 | 1.159 ± 0.061 | 22.574 ± 3.052 | 1.391 ± 0.187 | 4.246 |
| | CBM | ✗ | | 1.078 ± 0.011 | 0.758 ± 0.007 | 9.634 ± 0.078 | 7.544 ± 0.100 | 2.081 ± 0.010 | 3.321 ± 0.040 | 0.809 ± 0.022 | 1.758 ± 0.007 | 1.674 ± 0.012 | 1.436 ± 0.010 | 1.449 ± 0.012 | 1.684 ± 0.015 | 22.232 ± 0.202 | 1.067 ± 0.008 | 4.038 |
| | CEM | ✓ | $\mathcal{L}_{cosine}$ | 1.111 ± 0.123 | 1.134 ± 0.247 | 4.007 ± 2.175 | 1.647 ± 0.134 | 1.664 ± 0.264 | 10.931 ± 3.377 | 1.276 ± 0.028 | 3.942 ± 1.281 | 2.650 ± 0.258 | 1.324 ± 0.221 | 1.527 ± 0.246 | 1.380 ± 0.296 | 14.633 ± 7.135 | 5.634 ± 0.890 | 3.776 |
| | CBM | ✓ | | 0.977 ± 0.012 | 0.639 ± 0.008 | 9.001 ± 0.045 | 6.776 ± 0.056 | 1.719 ± 0.072 | 2.797 ± 0.066 | 0.703 ± 0.003 | 1.578 ± 0.023 | 1.372 ± 0.022 | 1.336 ± 0.015 | 1.310 ± 0.022 | 1.467 ± 0.021 | 20.841 ± 0.071 | 0.953 ± 0.018 | 3.676 |
| | CEM | ✗ | $\mathcal{L}_{cc}$ | 1.060 ± 0.062 | 0.877 ± 0.048 | 7.820 ± 0.228 | 6.365 ± 0.218 | 2.036 ± 0.135 | 3.496 ± 0.164 | 1.058 ± 0.072 | 2.032 ± 0.136 | 1.896 ± 0.083 | 1.336 ± 0.028 | 1.741 ± 0.093 | 1.387 ± 0.119 | 17.643 ± 0.650 | 1.310 ± 0.065 | 3.575 |
| CBGM | CEM | ✗ | $\mathcal{L}_{cc}$ | 1.137 ± 0.132 | 0.823 ± 0.066 | 6.488 ± 0.284 | 5.767 ± 0.103 | 1.703 ± 0.071 | 3.088 ± 0.214 | 0.937 ± 0.093 | 1.999 ± 0.061 | 1.723 ± 0.051 | 1.196 ± 0.032 | 1.560 ± 0.153 | 1.400 ± 0.043 | 16.781 ± 0.306 | 1.793 ± 0.092 | 3.314 |
| | CBM | ✗ | $\mathcal{L}_{cosine}$ | 1.057 ± 0.052 | 0.866 ± 0.056 | 7.270 ± 0.435 | 5.371 ± 0.412 | 1.761 ± 0.084 | 3.268 ± 0.295 | 1.152 ± 0.029 | 1.829 ± 0.062 | 2.024 ± 0.185 | 1.225 ± 0.085 | 1.621 ± 0.074 | 1.305 ± 0.034 | 14.960 ± 0.971 | 1.165 ± 0.222 | 3.205 |
| | CBM | ✓ | $\mathcal{L}_{cosine}$ | 1.156 ± 0.339 | 1.148 ± 0.630 | 5.739 ± 2.045 | 1.599 ± 0.340 | 2.371 ± 1.264 | 1.647 ± 0.485 | 0.609 ± 0.030 | 0.928 ± 0.121 | 1.260 ± 0.196 | 1.118 ± 0.298 | 1.943 ± 0.534 | 1.437 ± 0.189 | 9.970 ± 2.096 | 0.791 ± 0.114 | 2.265 |
| scCBGM | CBM | ✓ | $\mathcal{L}_{cc}$ | **0.153 ± 0.004** | **0.103 ± 0.005** | 1.195 ± 0.058 | **1.069 ± 0.061** | **0.155 ± 0.008** | **0.279 ± 0.015** | **0.076 ± 0.004** | **0.182 ± 0.010** | **0.165 ± 0.014** | **0.390 ± 0.011** | 0.359 ± 0.039 | **0.166 ± 0.007** | **3.151 ± 0.318** | 0.921 ± 0.244 | 0.598 |
| | CBM | ✗ | $\mathcal{L}_{cc}$ | 0.175 ± 0.004 | **0.103 ± 0.007** | **0.751 ± 0.082** | 1.142 ± 0.030 | 0.201 ± 0.012 | 0.297 ± 0.023 | 0.096 ± 0.011 | 0.221 ± 0.010 | 0.198 ± 0.011 | 0.430 ± 0.018 | **0.343 ± 0.059** | 0.202 ± 0.012 | 3.474 ± 0.129 | **0.166 ± 0.009** | **0.557** |

### D.8.3 VARIATIONAL AUTO-ENCODER VS. AUTO-ENCODER

In figure S7, we present a performance comparison between scCBGM and an ablated version of scCBGM where the variational auto-encoder backbone has been replaced with a simple auto-encoder (scCBGM (ae)). we found that the rMMD values on different cell populations of the Kang et al. (2017) dataset were typically lower (better) with a variational auto-encoder backbone than without.

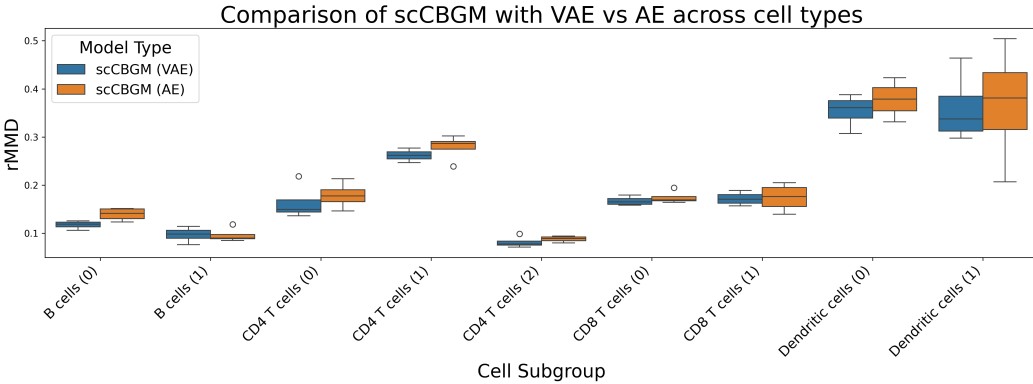

Figure S7: Comparison of rMMD performance between two versions of the auto-encoder backbone of scCBGM: a variational auto-encoder (VAE) and a regular auto-encoder (AE). The boxplots are computed over 4 seeds for each cell population from the Kang et al. (2017) dataset. Lower rMMD is better.

### D.9 ADDITIONAL BENCHMARKING RESULTS

### D.9.1 COUNTER-FACTUAL MODELING PREDICTS CELLULAR RESPONSE TO PERTURBATION

Tables S10 and S11 show results on the Kang et al. (2017) dataset on the rFID and rSD metrics. Similarily to our rMMD results in the main text, we find that our model outperforms other baselines in 5 out of 7 experiments.

### D.9.2 COMPOSITIONAL GENERALIZATION WITH MANY STIMULI

Tables S12 and S13 show results on rFID and rSD on the Cui et al. (2024) dataset. Similarly to our rMMD results in the main text, we find that our model outperforms other baselines in 4 out of 7 experiments.

### D.9.3 COMPLETE NAULT ET AL. (2023) RESULTS

In Tables S14, S15, and S16 we report the results of our experiment on the Nault et al. (2023) dataset on all available cell types for the rMMD, rFID, and rSD metrics respectively.

Table S10: rFID (Mean ± SEM) per cell group for different models (best, second-best, and **third-best** bolded) in the Kang et al. (2017) dataset.

| Model | B cells | T cells (CD4) | T cells (CD8) | Monocytes (FCGR3A) | Monocytes (CD14) | Dendritic Cells | NK cells |
|---|---|---|---|---|---|---|---|
| scCBGM | **0.265 ± 0.003** | **0.277 ± 0.003** | **0.515 ± 0.004** | **1.135 ± 0.044** | **1.719 ± 0.032** | **0.702 ± 0.010** | 1.230 ± 0.285 |
| scCBGM-FM (decode) | 0.247 ± 0.002 | 0.253 ± 0.001 | 0.457 ± 0.003 | 0.996 ± 0.040 | 1.769 ± 0.051 | 0.665 ± 0.009 | 0.215 ± 0.006 |
| scCBGM-FM (edit) | 0.248 ± 0.002 | 0.241 ± 0.003 | 0.472 ± 0.009 | 0.971 ± 0.018 | 1.723 ± 0.038 | 0.670 ± 0.005 | 0.202 ± 0.001 |
| CBGM | 1.086 ± 0.053 | 2.031 ± 0.059 | 2.142 ± 0.041 | 5.244 ± 0.112 | 5.353 ± 0.123 | 1.634 ± 0.031 | 1.664 ± 0.174 |
| Vanilla-FM (decode) | 0.937 ± 0.013 | 0.953 ± 0.002 | 0.999 ± 0.013 | 1.995 ± 0.009 | 1.589 ± 0.043 | 2.017 ± 0.026 | **0.215 ± 0.001** |
| Vanilla-FM (edit) | 0.551 ± 0.012 | 0.452 ± 0.007 | 0.584 ± 0.007 | 1.233 ± 0.039 | 1.341 ± 0.011 | 1.300 ± 0.033 | 0.177 ± 0.003 |
| biolord | 1.730 ± 0.001 | 3.219 ± 0.001 | 3.039 ± 0.006 | 3.978 ± 0.009 | 1.855 ± 0.003 | 1.979 ± 0.002 | 1.949 ± 0.001 |
| biolord-FM | 1.165 ± 0.003 | 1.782 ± 0.002 | 1.613 ± 0.004 | 5.446 ± 0.011 | 7.439 ± 0.020 | 1.986 ± 0.004 | 1.049 ± 0.004 |
| Cinema-OT | 1.500 ± 0.001 | 3.563 ± 0.001 | 3.077 ± 0.003 | 3.550 ± 0.004 | 4.637 ± 0.002 | 0.928 ± 0.000 | 2.565 ± 0.003 |
| scGen | 1.275 ± 0.003 | 2.732 ± 0.010 | 2.833 ± 0.013 | 2.567 ± 0.014 | 2.801 ± 0.081 | 0.883 ± 0.010 | 1.966 ± 0.022 |
| CVAE | 0.653 ± 0.006 | 1.002 ± 0.007 | 1.068 ± 0.019 | 4.075 ± 0.016 | 4.553 ± 0.034 | 1.216 ± 0.013 | 0.732 ± 0.007 |
| CVAE-FM (decode) | 0.568 ± 0.005 | 0.812 ± 0.006 | 0.948 ± 0.006 | 3.608 ± 0.023 | 4.257 ± 0.029 | 1.109 ± 0.010 | 0.623 ± 0.006 |
| CVAE-FM (edit) | 0.563 ± 0.002 | 0.803 ± 0.004 | 0.933 ± 0.003 | 3.723 ± 0.014 | 4.213 ± 0.035 | 1.123 ± 0.007 | 0.600 ± 0.003 |

Table S11: rSD (Mean ± SEM) per cell group for different models (best, second-best, and **third-best** bolded) in the Kang et al. (2017) dataset.

| Model | B cells | T cells (CD4) | T cells (CD8) | Monocytes (FCGR3A) | Monocytes (CD14) | Dendritic Cells | NK cells |
|---|---|---|---|---|---|---|---|
| scCBGM | **0.393 ± 0.004** | 0.507 ± 0.004 | 0.971 ± 0.006 | **0.968 ± 0.028** | 1.246 ± 0.033 | 0.744 ± 0.010 | 1.135 ± 0.254 |
| scCBGM-FM (decode) | 0.387 ± 0.014 | **0.515 ± 0.005** | 0.853 ± 0.015 | 0.868 ± 0.023 | 1.312 ± 0.046 | **0.742 ± 0.013** | 0.290 ± 0.023 |
| scCBGM-FM (edit) | 0.387 ± 0.002 | 0.500 ± 0.002 | 0.960 ± 0.013 | 0.838 ± 0.012 | 1.261 ± 0.040 | 0.726 ± 0.003 | 0.293 ± 0.003 |
| CBGM | 1.046 ± 0.057 | 1.824 ± 0.048 | 1.865 ± 0.010 | 3.432 ± 0.024 | 4.401 ± 0.124 | 1.677 ± 0.008 | 1.446 ± 0.130 |
| Vanilla-FM (decode) | 1.089 ± 0.029 | 1.042 ± 0.003 | 1.336 ± 0.044 | 1.484 ± 0.020 | **1.165 ± 0.038** | 2.062 ± 0.038 | 0.436 ± 0.031 |
| Vanilla-FM (edit) | 0.662 ± 0.011 | 0.750 ± 0.011 | 1.124 ± 0.018 | 1.025 ± 0.019 | 0.916 ± 0.010 | 1.431 ± 0.032 | **0.318 ± 0.012** |
| biolord | 0.888 ± 0.002 | 0.897 ± 0.002 | 0.878 ± 0.001 | 1.405 ± 0.004 | 0.371 ± 0.001 | 1.570 ± 0.001 | 0.444 ± 0.002 |
| biolord-FM | 1.283 ± 0.016 | 1.694 ± 0.005 | 1.785 ± 0.021 | 3.765 ± 0.012 | 6.414 ± 0.030 | 1.937 ± 0.009 | 1.114 ± 0.022 |
| Cinema-OT | 0.840 ± 0.001 | 1.593 ± 0.004 | 1.154 ± 0.003 | 1.591 ± 0.002 | 3.241 ± 0.007 | 0.700 ± 0.001 | 1.195 ± 0.007 |
| scGen | 0.967 ± 0.005 | 1.876 ± 0.012 | 1.872 ± 0.007 | 1.454 ± 0.019 | 1.979 ± 0.075 | 0.796 ± 0.010 | 1.366 ± 0.014 |
| CVAE | 0.730 ± 0.005 | 1.192 ± 0.018 | 1.422 ± 0.030 | 2.914 ± 0.030 | 3.846 ± 0.029 | 1.265 ± 0.010 | 0.767 ± 0.009 |
| CVAE-FM (decode) | 0.670 ± 0.005 | 0.959 ± 0.011 | 1.213 ± 0.036 | 2.527 ± 0.032 | 3.601 ± 0.022 | 1.206 ± 0.019 | 0.666 ± 0.018 |
| CVAE-FM (edit) | 0.643 ± 0.004 | 1.010 ± 0.008 | 1.309 ± 0.012 | 2.665 ± 0.015 | 3.544 ± 0.036 | 1.181 ± 0.009 | 0.646 ± 0.002 |

Table S12: rFID (Mean ± SEM) per cell group for different models (best, second-best, and **third-best** bolded) in the Cui et al. (2024) dataset.

| Model | T cells (Gamma-delta) | T cells (CD4) | T cells (CD8) | Dendritic (cDC2) | Dendritic (Langerhans) | Myeloid (Macrophages) | Lymphoid (NK cells) |
|---|---|---|---|---|---|---|---|
| scCBGM | 0.688 ± 0.006 | 0.523 ± 0.003 | 0.474 ± 0.007 | 0.851 ± 0.007 | 0.162 ± 0.002 | 1.263 ± 0.019 | 1.941 ± 0.021 |
| scCBGM-FM (decode) | **0.731 ± 0.007** | **0.555 ± 0.003** | 0.509 ± 0.005 | 0.875 ± 0.008 | 0.170 ± 0.002 | 1.276 ± 0.004 | **1.956 ± 0.008** |
| scCBGM-FM (edit) | 0.751 ± 0.005 | 0.570 ± 0.003 | 0.508 ± 0.003 | 0.869 ± 0.008 | 0.169 ± 0.001 | 1.319 ± 0.008 | 1.985 ± 0.015 |
| CBGM | 0.884 ± 0.091 | 0.588 ± 0.025 | **0.488 ± 0.015** | **0.840 ± 0.056** | **0.167 ± 0.004** | 1.555 ± 0.160 | 2.335 ± 0.070 |
| Vanilla-FM (decode) | 1.527 ± 0.040 | 1.474 ± 0.090 | 1.470 ± 0.057 | 1.223 ± 0.055 | 1.472 ± 0.044 | 4.001 ± 0.253 | 2.124 ± 0.110 |
| Vanilla-FM (edit) | 0.875 ± 0.020 | 0.644 ± 0.009 | 0.510 ± 0.016 | 0.646 ± 0.049 | 0.478 ± 0.045 | 3.518 ± 0.169 | 1.014 ± 0.044 |
| biolord | 2.164 ± 0.003 | 1.412 ± 0.001 | 1.265 ± 0.001 | 1.223 ± 0.000 | 0.999 ± 0.000 | 4.400 ± 0.013 | 2.612 ± 0.006 |
| biolord-FM | 1.357 ± 0.007 | 1.117 ± 0.005 | 1.056 ± 0.009 | 1.577 ± 0.003 | 1.125 ± 0.010 | 1.859 ± 0.009 | 2.634 ± 0.019 |
| Cinema-OT | 1.655 ± 0.002 | 1.056 ± 0.001 | 1.127 ± 0.001 | 0.976 ± 0.001 | 0.816 ± 0.000 | 4.187 ± 0.015 | 5.729 ± 0.007 |
| scGen | 1.601 ± 0.011 | 1.405 ± 0.009 | 1.070 ± 0.013 | 0.830 ± 0.004 | 0.291 ± 0.002 | 2.053 ± 0.005 | 2.767 ± 0.008 |
| CVAE | 0.730 ± 0.004 | 0.525 ± 0.002 | 0.470 ± 0.002 | 0.930 ± 0.003 | 0.167 ± 0.001 | 1.312 ± 0.006 | 2.047 ± 0.016 |
| CVAE-FM (decode) | 0.754 ± 0.005 | 0.559 ± 0.002 | 0.511 ± 0.004 | 0.929 ± 0.003 | 0.176 ± 0.001 | **1.288 ± 0.008** | 2.048 ± 0.006 |
| CVAE-FM (edit) | 0.778 ± 0.002 | 0.571 ± 0.004 | 0.507 ± 0.005 | 0.938 ± 0.001 | 0.177 ± 0.000 | 1.351 ± 0.002 | 2.093 ± 0.005 |

Table S13: rSD (Mean ± SEM) per cell group for different models (best, second-best, and **third-best** bolded) in the Cui et al. (2024) dataset.

| Model | T cells (Gamma-delta) | T cells (CD4) | T cells (CD8) | Dendritic (cDC2) | Dendritic (Langerhans) | Myeloid (Macrophages) | Lymphoid (NK cells) |
|---|---|---|---|---|---|---|---|
| scCBGM | 0.781 ± 0.018 | 0.684 ± 0.004 | 0.704 ± 0.011 | 0.852 ± 0.012 | 0.194 ± 0.005 | 1.301 ± 0.024 | **1.898 ± 0.015** |
| scCBGM-FM (decode) | 0.822 ± 0.010 | 0.757 ± 0.011 | 0.808 ± 0.013 | 0.866 ± 0.010 | 0.204 ± 0.002 | 1.326 ± 0.015 | 1.915 ± 0.014 |
| scCBGM-FM (edit) | 0.858 ± 0.003 | 0.756 ± 0.001 | 0.747 ± 0.003 | 0.872 ± 0.007 | 0.208 ± 0.002 | 1.343 ± 0.008 | 1.984 ± 0.013 |
| CBGM | 0.936 ± 0.122 | 0.767 ± 0.033 | 0.680 ± 0.029 | **0.836 ± 0.056** | 0.196 ± 0.002 | 1.520 ± 0.101 | 2.272 ± 0.049 |
| Vanilla-FM (decode) | 1.503 ± 0.024 | 1.678 ± 0.038 | 2.127 ± 0.069 | 1.212 ± 0.049 | 1.454 ± 0.035 | 3.455 ± 0.155 | 2.040 ± 0.121 |
| Vanilla-FM (edit) | 1.005 ± 0.011 | 0.873 ± 0.020 | 0.819 ± 0.029 | 0.636 ± 0.045 | 0.511 ± 0.050 | 3.105 ± 0.108 | 0.964 ± 0.038 |
| biolord | 1.099 ± 0.001 | 0.778 ± 0.001 | 0.785 ± 0.002 | 0.897 ± 0.001 | 0.795 ± 0.000 | 2.287 ± 0.007 | 1.537 ± 0.005 |
| biolord-FM | 1.431 ± 0.019 | 1.359 ± 0.022 | 1.526 ± 0.033 | 1.530 ± 0.008 | 1.132 ± 0.010 | 2.019 ± 0.016 | 2.525 ± 0.030 |
| Cinema-OT | 1.240 ± 0.003 | 0.869 ± 0.002 | 1.105 ± 0.002 | 0.918 ± 0.001 | 0.792 ± 0.001 | 3.138 ± 0.010 | 4.971 ± 0.028 |
| scGen | 0.999 ± 0.010 | 1.044 ± 0.009 | 0.821 ± 0.017 | 0.689 ± 0.002 | 0.205 ± 0.002 | 1.496 ± 0.007 | 2.119 ± 0.009 |
| CVAE | 0.852 ± 0.005 | 0.717 ± 0.001 | 0.688 ± 0.005 | 0.928 ± 0.003 | 0.204 ± 0.003 | 1.356 ± 0.012 | 2.035 ± 0.015 |
| CVAE-FM (decode) | 0.878 ± 0.016 | 0.733 ± 0.008 | 0.750 ± 0.017 | 0.919 ± 0.004 | 0.214 ± 0.006 | 1.383 ± 0.019 | 2.016 ± 0.028 |
| CVAE-FM (edit) | 0.895 ± 0.004 | 0.752 ± 0.003 | 0.754 ± 0.009 | 0.937 ± 0.000 | 0.216 ± 0.001 | 1.385 ± 0.003 | 2.076 ± 0.011 |

Table S14: rMMD (Mean ± SEM) per model and target group (best, second-best, and **third-best** bolded) in Nault et al. (2023) dataset

| Model | B cells | T cells | Hepatocytes Centrilobular | Endothelial cells | Stellate cell | Macrophages | Hepatocytes (Periportal) |
|---|---|---|---|---|---|---|---|
| scCBGM | 0.9309 ± 0.3402 | 0.8832 ± 0.3996 | 0.6236 ± 0.0114 | 1.1352 ± 0.3946 | 0.8952 ± 0.5483 | 0.6171 ± 0.4101 | 0.7521 ± 0.0457 |
| scCBGM-FM (decode) | **0.9497 ± 0.2785** | 0.8368 ± 0.3466 | 0.6084 ± 0.0286 | **1.0465 ± 0.3216** | **0.8609 ± 0.4399** | 0.5960 ± 0.3432 | 0.7186 ± 0.0564 |
| scCBGM-FM (edit) | 0.9499 ± 0.3040 | 0.8507 ± 0.3719 | **0.6172 ± 0.0012** | 1.0589 ± 0.3438 | 0.8437 ± 0.4515 | **0.6286 ± 0.4083** | 0.7079 ± 0.0565 |
| Vanilla-FM (decode) | 3.2153 ± 1.6126 | 3.7338 ± 2.0041 | 1.4582 ± 0.6899 | 8.1729 ± 2.6690 | 10.4811 ± 7.0579 | 2.3361 ± 1.3756 | 1.1853 ± 0.4838 |
| Vanilla-FM (edit) | 1.2832 ± 0.4411 | 1.4381 ± 0.2921 | 0.4424 ± 0.0877 | 6.8274 ± 0.9141 | 7.2351 ± 3.2392 | 1.2286 ± 0.4893 | 1.0301 ± 0.3172 |
| Biolord | 22.2091 ± 9.1562 | 27.6154 ± 12.7611 | / | 41.3361 ± 17.3826 | 44.3398 ± 32.5313 | 21.1511 ± 10.7040 | 4.7065 ± 2.0315 |
| Cinema-OT | 23.1503 ± 11.3323 | 29.3735 ± 15.0724 | 4.6668 ± 1.5976 | 40.3636 ± 20.2509 | 45.5700 ± 33.2892 | 11.7172 ± 4.8449 | 5.2949 ± 1.3632 |
| scGen | 11.5939 ± 5.2396 | 13.4637 ± 6.5632 | 2.2144 ± 0.6419 | 16.7187 ± 7.2767 | 12.0668 ± 9.0206 | 6.1012 ± 3.3398 | 2.3892 ± 0.7981 |
| CVAE | 1.263 ± 0.072 | 1.328 ± 0.060 | 1.842 ± 0.809 | 1.625 ± 0.121 | 1.349 ± 0.281 | 1.104 ± 0.229 | 1.554 ± 0.433 |
| CVAE-FM (decode) | 1.005 ± 0.102 | **0.870 ± 0.091** | 1.333 ± 0.585 | 1.018 ± 0.110 | 0.882 ± 0.117 | 0.852 ± 0.015 | 1.143 ± 0.263 |
| CVAE-FM (edit) | 0.938 ± 0.101 | 0.887 ± 0.161 | 1.337 ± 0.569 | 0.963 ± 0.102 | 0.853 ± 0.171 | 0.909 ± 0.122 | 1.131 ± 0.253 |

Table S15: rFID (Mean ± SEM) per cell group for different models (best, second-best, and **third-best** bolded) in the Nault et al. (2023) dataset.

| Model | B cells | T cells | Hepatocytes (Centrilobular) | Endothelial cells | Stellate cell | Macrophages | Hepatocytes (Periportal) |
|---|---|---|---|---|---|---|---|
| scCBGM | 1.028 ± 0.158 | 0.964 ± 0.241 | 0.755 ± 0.034 | **1.121 ± 0.255** | 1.042 ± 0.353 | **0.993 ± 0.756** | **0.842 ± 0.050** |
| scCBGM-FM (decode) | **1.015 ± 0.140** | 0.918 ± 0.210 | 0.717 ± 0.007 | 1.132 ± 0.230 | **0.964 ± 0.324** | 0.911 ± 0.674 | 0.807 ± 0.026 |
| scCBGM-FM (edit) | 1.022 ± 0.151 | 0.942 ± 0.224 | **0.727 ± 0.006** | 1.156 ± 0.250 | **0.964 ± 0.332** | 0.957 ± 0.723 | 0.810 ± 0.036 |
| Vanilla-FM (decode) | 3.425 ± 0.001 | 3.022 ± 0.147 | 1.552 ± 0.793 | 21.293 ± 6.744 | 21.950 ± 7.596 | 3.885 ± 1.869 | 1.232 ± 0.482 |
| Vanilla-FM (edit) | 1.499 ± 0.185 | 1.333 ± 0.198 | 0.436 ± 0.151 | 17.998 ± 3.897 | 14.030 ± 2.147 | 1.766 ± 0.489 | 1.028 ± 0.283 |
| biolord | 5.103 ± 0.560 | 5.510 ± 0.702 | / | 26.211 ± 9.435 | 23.090 ± 9.283 | 7.530 ± 4.283 | 4.353 ± 1.775 |
| Cinema-OT | 5.414 ± 0.918 | 5.841 ± 0.928 | 4.035 ± 1.900 | 18.180 ± 8.224 | 24.920 ± 9.252 | 4.808 ± 2.321 | 4.335 ± 1.276 |
| scGen | 6.018 ± 0.665 | 6.498 ± 1.000 | 3.246 ± 1.460 | 9.912 ± 3.031 | 6.016 ± 2.582 | 6.495 ± 5.042 | 3.457 ± 1.269 |
| CVAE | 1.064 ± 0.020 | 1.069 ± 0.020 | 1.778 ± 0.314 | 1.276 ± 0.134 | 1.126 ± 0.126 | 1.215 ± 0.303 | 1.623 ± 0.308 |
| CVAE-FM (decode) | 0.985 ± 0.087 | 0.909 ± 0.108 | 1.235 ± 0.417 | 1.014 ± 0.123 | 0.924 ± 0.106 | 1.032 ± 0.294 | 1.190 ± 0.324 |
| CVAE-FM (edit) | 0.961 ± 0.070 | **0.926 ± 0.113** | 1.250 ± 0.421 | 0.974 ± 0.082 | 0.898 ± 0.136 | 1.056 ± 0.326 | 1.192 ± 0.321 |

Table S16: rSD (Mean ± SEM) per cell group for different models (best, second-best, and **third-best** bolded) in the Nault et al. (2023) dataset.

| Model | B cells | T cells | Hepatocytes (Centrilobular) | Endothelial cells | Stellate cell | Macrophages | Hepatocytes (Periportal) |
|---|---|---|---|---|---|---|---|
| scCBGM | 0.986 ± 0.103 | 0.974 ± 0.182 | 0.705 ± 0.090 | 1.073 ± 0.232 | 1.014 ± 0.358 | **0.961 ± 0.794** | **0.742 ± 0.001** |
| scCBGM-FM (decode) | 0.999 ± 0.089 | 0.902 ± 0.165 | 0.683 ± 0.061 | 1.108 ± 0.209 | 0.943 ± 0.326 | 0.927 ± 0.722 | 0.713 ± 0.027 |
| scCBGM-FM (edit) | 1.021 ± 0.084 | 0.968 ± 0.170 | **0.704 ± 0.089** | 1.154 ± 0.230 | 0.914 ± 0.342 | 0.959 ± 0.763 | 0.712 ± 0.016 |
| Vanilla-FM (decode) | 3.051 ± 0.276 | 2.686 ± 0.012 | 1.394 ± 0.572 | 20.301 ± 5.140 | 25.426 ± 10.854 | 3.173 ± 1.247 | 1.330 ± 0.635 |
| Vanilla-FM (edit) | 1.703 ± 0.123 | 1.290 ± 0.287 | 0.504 ± 0.218 | 17.380 ± 2.853 | 16.764 ± 4.256 | 1.416 ± 0.136 | 1.037 ± 0.319 |
| biolord | 2.261 ± 0.000 | 2.277 ± 0.137 | / | 20.420 ± 6.117 | 22.210 ± 10.198 | 3.418 ± 1.693 | 1.444 ± 0.641 |
| Cinema-OT | 2.784 ± 0.189 | 2.839 ± 0.320 | 1.414 ± 0.290 | 13.078 ± 5.379 | 24.800 ± 10.770 | 1.440 ± 0.128 | 1.813 ± 0.307 |
| scGen | 3.544 ± 0.109 | 3.778 ± 0.372 | 0.981 ± 0.118 | 5.742 ± 1.091 | 3.615 ± 1.817 | 3.859 ± 3.268 | 1.215 ± 0.312 |
| CVAE | 0.922 ± 0.028 | 0.911 ± 0.015 | 1.398 ± 0.616 | **1.046 ± 0.054** | 0.926 ± 0.082 | 1.102 ± 0.266 | 1.303 ± 0.429 |
| CVAE-FM (decode) | 0.944 ± 0.031 | 0.816 ± 0.121 | 1.235 ± 0.449 | 1.015 ± 0.039 | 0.921 ± 0.104 | 0.989 ± 0.248 | 1.154 ± 0.307 |
| CVAE-FM (edit) | **0.942 ± 0.070** | 0.933 ± 0.100 | 1.268 ± 0.435 | 0.990 ± 0.076 | 0.892 ± 0.140 | 1.032 ± 0.311 | 1.166 ± 0.294 |

### D.10 CASE STUDY: CONTROLLED SINGLE-CELL EDITING FOR ENHANCED DRUG RESPONSE

Figures S8 & S9 show the cell population after editing appeared similar to the populations which responded to the treatment, in both rMMD and gene-expression changes

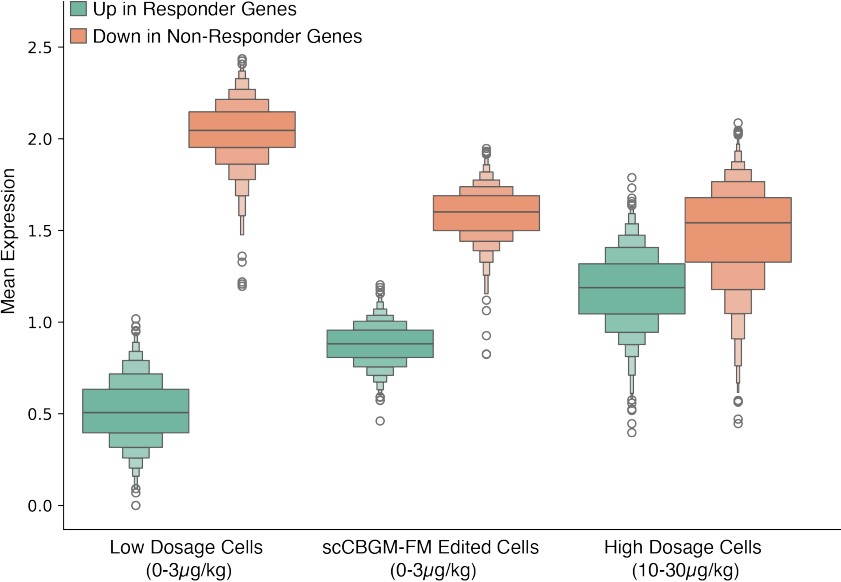

Figure S8: **Gene expression trends under in-silico perturbation in the Nault et al. (2023) dataset.** The distribution of mean expression values (averaged across cells in each group) for the top 100 upregulated (green) and downregulated (orange) marker genes. The edited cells (center) successfully reproduce the target gene signatures, shifting the expression of responder and non-responder genes towards the levels observed in the true high-dosage cells, complementing the global results in Figure 5. We note that only 40 of these marker genes overlap with the total set of top 100 genes defining the manipulated pathway concepts (500 total). The successful recovery of the remaining 80% suggests that the model captures downstream regulatory effects beyond the direct inputs.

### D.11 CELL SUBTYPE ACCURACY

To complement our benchmark, we evaluate whether edited cells from a given model preserve their subtype $s$. For each edited cell in the Kang et al. (2017) dataset, we predict its subtype using a kNN classifier with $k = 15$ neighbors. In Table S17, we report the predicted subtype accuracy (mean $\pm$ SEM) for different models used to edit the cells. scCBGM (-FM) achieves again the best performance over most cell types. As expected, models that achieved an rMMD $\leq 1$ show high accuracy (rMMD $\leq 1$ 1 means that the predicted counterfactual distribution is closer to the target than any other cell-type distribution in the data). However, we note that high subtype accuracy does not necessarily lead to successful counterfactual predictions. Indeed, the predicted distribution may be closest to the correct subtype distribution but not necessarily overlap.

Table S17: Subtype Prediction Accuracy (Mean ± SEM) per subtype for different models on the Kang et al. (2017) dataset (best, second-best, and **third-best** models bolded).

| | B-cells | | CD14 | | CD4 T cells | | | CD8 T cells | | Dendritic | | FCGR3A+ | | NK cells |
|---|---|---|---|---|---|---|---|---|---|---|---|---|---|---|
| Model | Type 0 | Type 1 | Type 0 | Type 1 | Type 0 | Type 1 | Type 2 | Type 0 | Type 1 | Type 0 | Type 1 | Type 0 | Type 1 | Type 0 |
| scCBGM | **0.996 ± 0.001** | **0.822 ± 0.008** | 0.990 ± 0.000 | **0.222 ± 0.045** | **0.983 ± 0.002** | 0.863 ± 0.007 | **0.734 ± 0.011** | 0.980 ± 0.001 | **0.546 ± 0.007** | 0.973 ± 0.003 | 0.710 ± 0.006 | 0.769 ± 0.043 | **0.106 ± 0.009** | 0.992 ± 0.001 |
| scCBGM-FM (decode) | 0.996 ± 0.000 | 0.817 ± 0.014 | **0.988 ± 0.003** | 0.139 ± 0.028 | 0.985 ± 0.002 | 0.837 ± 0.011 | 0.730 ± 0.007 | **0.972 ± 0.004** | 0.552 ± 0.005 | 0.958 ± 0.004 | 0.722 ± 0.006 | **0.786 ± 0.037** | 0.121 ± 0.005 | **0.980 ± 0.002** |
| scCBGM-FM (edit) | 0.996 ± 0.001 | 0.833 ± 0.006 | 0.988 ± 0.001 | 0.306 ± 0.053 | 0.983 ± 0.001 | 0.881 ± 0.005 | 0.745 ± 0.006 | 0.975 ± 0.001 | **0.551 ± 0.008** | **0.948 ± 0.004** | **0.682 ± 0.000** | 0.876 ± 0.018 | 0.093 ± 0.007 | 0.960 ± 0.002 |
| Vanilla-FM (decode) | 0.338 ± 0.014 | 0.020 ± 0.002 | 0.802 ± 0.014 | 0.028 ± 0.028 | 0.474 ± 0.009 | 0.125 ± 0.007 | 0.008 ± 0.002 | 0.412 ± 0.011 | 0.065 ± 0.005 | 0.033 ± 0.005 | 0.028 ± 0.006 | 0.108 ± 0.004 | 0.011 ± 0.003 | 0.990 ± 0.001 |
| Vanilla-FM (edit) | 0.425 ± 0.026 | 0.450 ± 0.014 | 0.812 ± 0.015 | 0.306 ± 0.053 | 0.858 ± 0.005 | 0.468 ± 0.006 | 0.281 ± 0.022 | 0.825 ± 0.012 | 0.362 ± 0.009 | 0.044 ± 0.021 | 0.131 ± 0.025 | 0.164 ± 0.025 | 0.056 ± 0.004 | 0.967 ± 0.004 |
| CVAE | 0.988 ± 0.001 | 0.833 ± 0.006 | 0.990 ± 0.001 | 0.111 ± 0.000 | 0.970 ± 0.002 | **0.844 ± 0.003** | 0.738 ± 0.008 | 0.945 ± 0.006 | 0.444 ± 0.010 | 0.904 ± 0.008 | 0.648 ± 0.007 | 0.983 ± 0.001 | 0.142 ± 0.005 | 0.922 ± 0.001 |

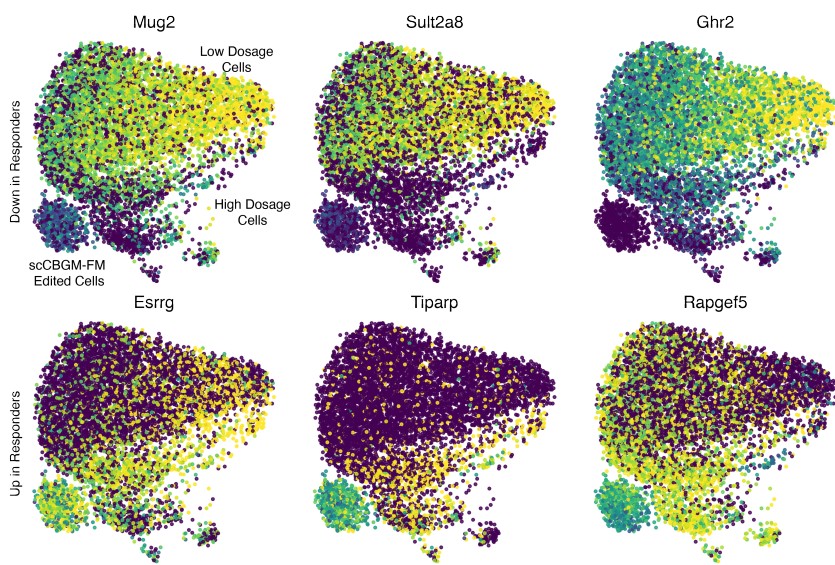

Figure S9: **Predicted gene expression trends match experimental ground truth.** UMAP visualizations of representative genes from the top 100 differentially expressed set analyzed in Figure S8. **Top row:** Genes downregulated in responders exhibit high expression (yellow) in low-dosage cells but are correctly suppressed in the edited population. **Bottom row:** Genes upregulated in responders are activated in the edited cells, shifting from the low-dosage state (purple) to the high-dosage phenotype (yellow). These patterns confirm that scCBGM-FM drives granular gene-specific shifts consistent with the aggregate trends.

