# OpenReview forum: "scCBGM: Single-Cell Editing via Concept Bottlenecks"
_ICLR.cc/2026/Conference — Submitted to ICLR 2026_

### Official Review · Reviewer_AAzm · 2025-10-24

**Soundness:** 2
**Presentation:** 3
**Contribution:** 2
**Rating:** 2
**Confidence:** 3

**Summary:**

The paper introduces scCBGM, a model for performing counterfactual perturbation prediction on single-cell RNA-seq data. scCBGM individuates concepts in the latent space, which it splits into two components: known concepts and unknown residual factors. Known concepts represent annotated covariates, like cell type. Unknown concepts are unobserved axes of cellular variation. The authors train scCBGM on a combined task, where the standard VAE loss is combined with a concept loss. The authors also combine the latent space in scGBM with flow matching, using factor representations as conditioners. The main use of the methods is performing counterfactual predictions of perturbation effects on single cells, which the authors benchmark on the task of simulating state shifts in multiple biological settings.

**Strengths:**

I find the paper enjoyable and the scientific question quite compelling. The attempt to infuse interpretability into perturbation models is relevant to step closer to their deployment, and I am happy to see the authors develop their research in this direction. Everything is described clearly, the Appendix is very polished, and the code is readable; I could trace back multiple questions I had to the analysis notebooks.

**Weaknesses:**

Currently, my questions and remarks suggest a negative score. However, I also admit that I might have missed a few points or hold misconceptions, and I remain more than happy to improve my rating during the rebuttal phase.

**Major**

- **Scope and formulation** My main concern regards the way concepts are formulated. If I understand correctly, the factors associated with known concepts are trained to approximate exactly these concepts, so the best possible outcome of the training process is for a cell to be encoded to its own conditioner. I do get that by doing it, one does not obtain the exact values for the concept, but more of an "activation". However, I feel that this is not that different than having a standard conditional VAE's (cVAE) latent space with some dimensions dedicated to representation and some others to covariate-based conditioning. I also acknowledge the originality of the cross-covariance loss to further remove the effect of known concepts from the latent space, but a similar effect is, again, to some extent present in cVAEs as the decoder regresses out conditioning variables (hence why such a method is so popular for batch correction). I am not hinting at the fact that the model is no different from a cVAE, but I believe that eventually it may have a similar effect, especially when conditioning the Flow Matching model.

-  **Editing and Flow Matching.** In my opinion, it is a bit inaccurate to define what has been done with Flow Matching as counterfactual/editing. Starting from the decoding, I think I would need an example showing that if you start from a noise variable $z \sim N(0,I)$ and generate from it conditioned on the concepts  $[u_i, c_i]$ of a cell $x_i$ (i.e., no editing), you effectively retrieve something close to the original cell $x_i$. The reason why I am saying this is that I am not sure to what extent generating from noise violates the "cell identity" information embodied by $u$. A similar concept applies to the editing procedure. You assume that inverting the flow to noise preserves the cell identity, but in a way that is already encoded by the unknown factors in $u$. Maybe the performance gain by using flow matching arises since the model is evaluated exclusively based on distributional metrics, at which the flow model may be better, as it smooths predictions around the data points.

-  **Table 2.** The results in Table 2 are not significant. I think this ablation really speaks against the worth of the method against the reference approach. Maybe you could find another way to assess the improvement?

- Unfortunately, I am not very convinced by the results of the case study. My main concern is the quality of the evidence, which, for now, is limited to a new cluster on the UMAP. I think this result and the whole study in general would be more interesting if more insight into the gene expression changes were provided. For example, are there any interesting genes upregulated by your synthetic simulation process? Do they agree with the state of the art?

**Minor**

- L52-53: "Existing perturbation...": I am not 100% sure about this statement. I thought the idea behind CPA, Biolord, STATE, and similar methods is to predict a state shift from a basal/initial cell state. I feel this is quite akin to the counterfactual task described here.

- It would be great if you could add subscripts referring to the considered distribution for the expectations.

- L201-211: Unless you are familiar with the concepts, it is not clear what the difference between a standard bottleneck model and a concept embedding model is.

- L277-278: I would write "data generating process" rather than "generative model".

- I think the MMD analysis could be complemented by something like subtype classification. In other words, pre-train a subtype classifier and evaluate how many times the subtype is preserved in translated cells.

[1] Lotfollahi, Mohammad, et al. "Predicting cellular responses to complex perturbations in high‐throughput screens." Molecular systems biology 19.6 (2023): e11517.

[2] Piran, Zoe, et al. "Disentanglement of single-cell data with biolord." Nature Biotechnology 42.11 (2024): 1678-1683.

[3] Adduri, Abhinav K., et al. "Predicting cellular responses to perturbation across diverse contexts with State." bioRxiv (2025): 2025-06.

**Questions:**

1. Are the MMDs computed on decoded data? If yes, I find the use of the rMMD a bit strange. Since you have the same ground truth for all models you are benchmarking, you could just compare them based on that, right? In many cases (e.g. Monocytes, Dendritic cells, NK cells), the response predicted by scCBGM is much worse than the population with the lowest distance to the perturbed subtype. I deem this result a bit concerning with respect to the reliability of the model.

2. You refer to Vanilla FM as CellFlow here. But CellFlow is a state-to-state transition model using OT-based Conditional Flow Matching. Provided you reimplemented CellFlow, which does not exploit noise in its formulation, how do you perform editing and decoding here? Or how is the Vanilla model different from the scCGMB one? I looked for this information in the appendix, but I could not locate it.

---

> ### Author Response · Authors · 2025-11-23
>
> We thank the reviewer for their very thorough feedback. We are pleased you found the research direction of infusing interpretability into perturbation models compelling and our work well described.
>
>
> ### Respond to Major Weaknesses:
>
> > Scope and formulation
>
> Thank you for this insightful point about the relationship between scCBGM and conditional VAEs (cVAEs). You are absolutely correct that in a perfect scenario where scCBGM's concept predictions exactly match the ground truth, the learned concepts would be equivalent to the given conditions in a cVAE. However, the main difference lies in how these concepts are handled during counterfactual editing.
>
> In a cVAE, you provide conditions at the encoder input and again at the decoder. For editing, this creates a critical issue: when you encode a cell with its original conditions and then provide new conditions at the decoder, you introduce conflicting information. The latent representation contains the original concepts (even with classifier-free guidance), while the decoder receives different conditions, leading to inconsistent editing.
>
> In contrast, scCBGM learns to explicitly separate concepts from other sources of variation through our concept bottleneck architecture and cross-covariance loss. This creates two distinct representations: learned concepts and unknown factors. During editing, we modify only the concept representation while preserving the unknown factors, minimizing conflicting information and enabling more principled counterfactual generation.
>
> To directly address your concerns about the practical differences, we have conducted additional experiments comparing scCBGM with a standard cVAE baseline. We evaluated both architectures in two contexts, as standalone generative models and as conditioning frameworks for Flow Matching (scCVAE-FM). As shown in **Tables 1, 2, and 3** (and supported by additional metrics in **Appendix D.9.1**), scCBGM and scCBGM-FM consistently outperform their cVAE counterparts across the vast majority of cell types and experimental settings. This empirical performance gap confirms that the structured disentanglement provided by scCBGM offers a distinct advantage over standard cVAE conditioning for counterfactual tasks.
>
>
> > Editing and Flow Matching.
>
> We agree that demonstrating accurate data reconstruction is crucial to justify "editing" and ensure cell identity is preserved. We have added Figure S5, which qualitatively and quantitatively demonstrates that scCBGM-FM faithfully reconstructs individual cells. Despite the noise inherent in the sampling procedure, scCBGM-FM achieves low reconstruction error (MSE=0.097), significantly outperforming Vanilla-FM (MSE=0.442)  and confirming that conditioning on scCBGM latents effectively retains cell identity
>
> Furthermore, we emphasize that Flow Inversion is a gold-standard method for counterfactual editing in computer vision, analogous to the encoder-decoder inference used in VAEs. Our results consistently show that the Edit (inversion-based) procedure outperforms the Decode (random noise) procedure, validated across multiple metrics confirming that the specific noise encoding captures essential identity information.
>
>
>
> > Table 2. The results in Table 2 are not significant.
>
> We acknowledge that the bolded method is not always significantly better than the second best performing one. To address this concern and provide a clearer assessment of performance, we now highlight the top three methods for each cell type (Tables 1,2  and 3). This presentation confirms that scCBGM variants consistently appear among the best performing methods across different cell types and datasets.
>
>
> To further strengthen our evaluation, we've expanded the ablation studies on synthetic data to include a broader design space and additional datasets, which provided clearer separation between models. These comprehensive results are presented in Table S7 with discussion in Section 3. We also added ablations on the Kang dataset (Table S8), demonstrating that our architectural choices lead to robust improvements across both synthetic and real data scenarios.
>
> >  The case study.
>
> We agree that a quantitative, gene-centric evaluation is necessary to substantiate the UMAP visualization. We have added **Figures S8 and S9**, which analyzes the most up- and down-regulated genes between true responders and non-responders in hepatic stellate cells. The results confirm that our in silico edited cells successfully recapitulate the expression trends of these marker genes, shifting specifically toward the responder profile. Additionally, the low rMMD score ($<1$) provides further quantitative evidence that the edited population has significantly moved toward the true responder distribution.

---

> ### Author Response · Authors · 2025-11-23
>
> ### Respond to Minor Weaknesses
>
> > L52-53 "Existing perturbation..."
> We have updated the manuscript according to your suggestions to give a more accurate description of the landscape of existing methods and open challenges.
>
> We replaced:
> _''Existing perturbation modeling methods focus on conditional distributions of cell states across treatments, but they do not enable counterfactual editing of individual cells (scGen, scVIDR, scDisinfact)_''
>
> With:
> _''Earlier methods modeled conditional distributions of cell states across different contexts, capturing population-level effects but not counterfactuals for individual cells (scGen, scVIDR, STATE). More recent work has moved toward cell-level counterfactual prediction (scDisinfact, biolord), but explicit, interpretable control remains an open and important area for further development.''_
>
>
> >subscripts
>
> Thank you for catching this, we have updated equation 3 with the subscripts.
>
> > L201-211:  Unless you are familiar with the concepts, it is not clear what the difference between a standard bottleneck model and a concept embedding model is.
>
> Thank you for catching this, we clarified the difference in the manuscript lines 217-221
>
> > L277-278: I would write "data generating process" rather than "generative model"
>
> Thank you for catching this. It has been updated in the manuscript.
>
> > I think the MMD analysis could be complemented by something like subtype classification. In other words, pre-
> train a subtype classifier and evaluate how many times the subtype is preserved in translated cells.
>
>
> We agree with the reviewer and added a complementary analysis in Appendix D.7 (Table S6 and Figure S6). In this analysis, we train two classifiers: one for the binary concept being intervened on, and another for cell type identity, which is not included among the known concepts (and thus must be represented in the unknown layer). This experiment shows that intervening on a known concept does not alter other properties encoded in the unknown layer, indicating minimal concept leakage. We did not include this metric in the standard metrics we report performance with, as it does not readily scale to interventions involving multiple concepts and requires task-specific classifiers. However, we have added multiple OT-based metrics.
>
>
> ### Respond to Questions
>
>
> >Question 1
>
>
> To clarify, the MMDs are computed on the original observed data, not on the decoded samples. The cases where scCBGM shows an rMMD greater than 1 (appearing worse than the population with the lowest distance to the perturbed subtype) primarily reflect the imperfections of real scRNA-seq data, rather than limitations of the model itself.
>
> In practice, cell type annotations in such datasets are inherently noisy and sometimes supported by only a small number of cells. Furthermore, several annotated cell types overlap substantially in expression space, making it difficult to define a single, unambiguous “ground truth” for the perturbed response.
>
> We have updated the Appendix with a section D.3 (`CHALLENGES OF EVALUATION ON REAL DATA`), where Figure S4 illustrates this overlap visually and Table S4 presents quantitative evidence. We also reference this section in the main text now.
>
> We sincerely appreciate the reviewer’s comment, as it highlights a key challenge in benchmarking models on real single-cell data one that is often underappreciated without deep familiarity with the dataset or biological domain.
>
> >Question 2
>
> We appreciate the reviewer pointing out this potential confusion. To clarify, our baseline is a Conditional Flow Matching model trained with an Optimal Transport path (OT-CFM), conditioned on the known biological concepts (e.g., cell type and perturbation status). While this aligns with the generative framework used in CellFlow (Klein et al., 2025), we acknowledge that CellFlow is often framed specifically as a state-to-state transition model.
>
> To avoid ambiguity, we have renamed this baseline to Vanilla-FM in the revised manuscript. This model represents a standard, high-fidelity generative approach where flow matching is conditioned directly on observed covariates rather than a structured latent space, which is how scCGMB-FM is conditioned.
>
> We have updated the main text (**Section 5**) to reflect this terminology and ensure the distinction is clear.

---

> > ### Comment · Reviewer_AAzm · 2025-11-23
> >
> > I would like to thank the authors for their significant effort in clarifying my doubts. I commend their clarity and structure in providing new results and insight, which improved the quality of the paper. I definitely believe that after the rebuttal, my original score does not hold anymore, and I am now moving to a borderline rejection. Below, I will provide an answer to the new question from the authors.
> >
> > **Comparison with cVAE.** I partially agree with this answer. I do understand how cVAE adds conditioning to the encoder while scCBGM attempts to squeeze conditioning information from the feature representations. However, cVAEs should be contextualized in the interplay between the model’s encoder and decoder, where the decoder fundamentally attempts to regress out the information from the conditioner. This is the main takeaway from batch correction models in scRNA-seq, where adding a batch conditioning variable to the latent space enables batch-free latent representations. Modifying the conditioner in the decoding phase can retrieve pretty solid cross-condition predictions. Nonetheless, I acknowledge the new results comparing scCBGM to cVAE and find them relevant to my initial concern.
> >
> > **Editing an FM.** Here, I am partially convinced by the answer. Looking at Fig.5, I feel that all models perform quite on par from a visual standpoint. In terms of metrics, scCBGM is better and closer to real data, but how is the MSE evaluated on Vanilla FM if it’s not conditioned on cellular representations but simply on conditioning concepts? I do expect FM models conditioned on scCBGM to do better at picking up exact cells in the dataset, as you are essentially using their representations as proxies for conditioning. Also, without error bars, I struggle to assess the difference in MSE between cVAE/VAE and scCBGM.
> >
> > **Flow inversion.** Could you also provide references stating that flow inversion is a gold standard for “counterfactual predictions”? I feel here the word counterfactual is quite strong in the causal sense. Conditioning a flow on a different conditioner leads to a normalising flow approximating a different density function, so I am not sure that bringing back cells to the noise space strictly respects cell identity, cause it destroys all information about the features. I also admit I did not read up too much about it, so happy to revisit my opinion. I also am not fully factoring these concepts into my evaluation, as something may not be fully counterfactual but still useful for practical purposes (like the model in this case).
> >
> > **Table 2. The results in Table 2 are not significant.** I am a bit unsure about this answer. I do understand that it’s interesting to experiment with design spaces. But why remove the results on 2x2? It is not the best approach; it looks as if a significant improvement was actually sought. Also, while the results are now statistically significant, the difference is in principle very small, although I acknowledge this as a less relevant point. Finally, in Table S8, it is said that the results are reported as Mean ± SEM. But I cannot find any confidence intervals.
> >
> > **The case study.** I find the results of the case study quite convincing, although I still don’t understand why the generated cells would form a separate cluster rather than integrating with responding cells. I think this could also be a model artefact, where trends for specific genes are captured, but the whole cell state is not necessarily realistic.
> >
> > **Cell type classification experiment.** I personally find this experiment interesting. What I meant by subtype analysis was to see if, after running counterfactual predictions, you still preserve the same subtype nested within a cell type (hence, a stronger evidence for cell identity preservation via editing). However, my request wasn’t very clear or explicit, so this request should be disregarded.
> >
> > **Other typos and missing details, and Q2.** Thanks for implementing my suggestions, the edits look good to me!
> >
> > *Q1.* I somewhat disagree with some aspects of this answer. The cross-cell-type analysis is indeed insightful, but using an MMD-based evaluation, you should still be able to approximate the rough distribution of the correct cell type. Take NK cells from Cui et al.. Being 5x worse than the closest cell type suggests that the model is probably quite off on approximating the ground cellular distribution, especially when Vanilla FM works so well on this aspect. I think this clearly defines shortcomings in the model’s generalisation.
> >
> > Overall, though I appreciate all the work the authors put into the revisions, I am quite confident in my current score until further interaction with other reviewers. I am happy to rethink some of my comments if the authors detect inaccuracies. I wish you the best of luck with the answers to the other reviewers!

---

> > > ### Author Response · Authors · 2025-11-25
> > >
> > > We thank the reviewer for the thoughtful and timely follow-up, and for engaging so carefully with our responses. We appreciate the constructive feedback and the opportunity to clarify remaining points. We address the comments below and look forward to continuing the discussion in the context of the other reviewers’ observations
> > >
> > > >Comparison with cVAE
> > >
> > > We agree with your characterization regarding the cVAE's decoder acting to regress out conditioning information, a principle central to many batch-correction methods. We have updated the manuscript in **Section 5** (purple text) to explicitly describe the cVAE baseline in this context. Nevertheless, our results suggest that the explicit orthogonality penalty in scCBGM enforces the stricter separation required for high-fidelity counterfactual editing, resulting in the observed performance improvements where we outperform the _cVAE_ models in a majority of the cases.
> > >
> > >
> > > > Editing an FM.
> > >
> > > The reviewer is entirely correct that flow matching models conditioned on scCBGM latents should naturally outperform those conditioned only on broad concepts at the task of reconstruction. This is precisely the point we aimed to quantify.
> > >
> > > To clarify, for Vanilla-FM, we generate a prediction for a specific cell by sampling random noise and integrating the flow conditioned only on that cell's observed concepts (cell type and stimulation labels), then calculating the error against the original cell expression. Because Vanilla-FM lacks access to cell-specific identity, it generates a generic sample from the conditional distribution rather than reconstructing the specific instance. The high MSE confirms this loss of identity.
> > >
> > > In contrast, the low MSE of scCBGM-FM proves that our unknown latent ($u$) successfully captures the granular cell identity required for true counterfactual editing (modifying this cell) rather than just conditional generation (generating a cell). We have expanded **Appendix D.4** (purple text) to explicitly detail this protocol and its purpose. Additionally, we have updated **Figure S5** to include errors computed over 5 random seeds, for a proper assessment of the significant performance gap between scCBGM variants and the cVAE/VAE baselines.
> > >
> > > > Flow inversion.
> > >
> > > We appreciate the reviewer’s perspective. While "counterfactual" is indeed a strong term, flow inversion is frequently used as a practical implementation of Pearl's Abduction-Action-Prediction framework in deep generative modeling.
> > >
> > > Mathematically, the inversion of data $\mathbf{x}$ to the base distribution $\mathbf{x}_0$ constitutes the Abduction step (inferring the exogenous noise/latent state that generated the observation). The subsequent forward pass with a new condition constitutes the Action and Prediction steps. Because the ODE flow is bijective, information is not destroyed; rather, the "cell identity" is mapped to the noise space $\mathbf{x}_0$.
> > >
> > > This framing is established in recent literature. For example, [Xia et al.](https://arxiv.org/pdf/2506.14399) and [Rout et al.](https://rf-inversion.github.io/data/rf-inversion.pdf) explicitly frame flow/diffusion inversion within the Abduction-Action-Prediction formalism. Similarly, [Sanchez & Tsaftaris](https://arxiv.org/pdf/2202.10166) [Pérez-García et al.](https://arxiv.org/pdf/2312.12865) or [Wang et al.](https://arxiv.org/pdf/2411.04746) rely on this inversion mechanism to perform "counterfactual" estimation and editing. We have updated the manuscript to explicitly contextualize our approach within this literature, see **Section 2.5** (purple text).
> > >
> > > > Table 2.
> > >
> > > We appreciate your comment, though we were a bit uncertain about the specific table being referenced. In both your original review and this follow-up comment, you refer to Table 2. However, based on your description, it seems you may actually be referring to the synthetic ablation table, which was Table 1 in the original manuscript. Please let us know if we have understood this correctly.
> > >
> > > In response to requests from other reviewers, we expanded the ablation design from a 2×2 to a 2×2×2 setup. As part of this update, we reran all models and moved the resulting table to the supplementary material (Table S7), since it no longer fit within the main text. We also felt it was more coherent to present all ablation results together in the supplementary section rather than splitting them between the main text and appendix.
> > >
> > > Regarding Table S8, thank you for catching that — we have now updated it to display the correct Mean ± SEM formatting as stated in the caption.

---

> > > > ### Author Response · Authors · 2025-11-25
> > > >
> > > > > The case study.
> > > >
> > > > We appreciate your positive feedback. Regarding the UMAP separation, this is a challenging zero-shot scenario where the model never observed responding hepatic stellate cells during training. While visual integration isn't perfect, our in-silico edits successfully drive a marked shift toward the true responders in both global distribution (rMMD < 1, **Figure 5**) and specific marker gene expression (**Figures S8 & S9**). We view these results as encouraging, suggesting that the model successfully identifies and modulates the underlying biological signals.
> > > >
> > > > >Cell type classification experiment.
> > > >
> > > > We appreciate the clarification. Although you mentioned disregarding the request, we believe it touches on an important validation of our method and have thus added this experiment to Appendix D.11.
> > > >
> > > > For each edited cell in the Kang dataset, we predicted its subtype using a kNN classifier with $k=15$ neighbors. In Table S17, we report the subtype prediction accuracy (mean $\pm$ SEM) for different models used for editing the cells. scCBGM(-FM) achieves again the best performance over most cell types. As expected, models that achieved an rMMD $\leq 1$ show high accuracy. Indeed, the rMMD compares the predicted counterfactual distribution against the specific true held-out subtype (numerator), normalized by the distance from the closest observed cell population in the training data (denominator). Since the denominator represents the best possible approximation using existing data, a low rMMD inherently measures whether the prediction successfully retains specific subtype identity versus simply retrieving the most similar existing cell state. Additionally, we note that high subtype accuracy does not necessarily lead to successful counterfactual predictions. Indeed, the predicted distribution may be closest to the correct subtype distribution but not necessarily overlap.
> > > >
> > > >
> > > > > Q1.
> > > >
> > > > We agree this highlights a generalization challenge for this specific cell type. However, **scCBGM outperforms all baselines in nearly all experimental settings**, suggesting the limitation is isolated rather than systematic.
> > > >
> > > > For the NK cells in *Cui et al.*, **no model achieved an rMMD below 1**, reflecting the intrinsic difficulty of this zero-shot task and the many nuances of real data. This is precisely why we include synthetic evaluations, to assess performance under more controlled conditions.

---

> > > > > ### Comment · Reviewer_AAzm · 2025-11-26
> > > > >
> > > > > Thank you very much for your additional clarifications. I really appreciate the time the authors spent clarifying doubts and engaging in discussion.
> > > > >
> > > > > **Table 2.** Sorry for misreferencing the table. My concern remains somewhat unsolved, although the table has been deprioritized and moved to the appendix. Significance only arises (and with low magnitude) when increasing the complexity of the design. What I meant in my comment is that it sounds like a slightly cherry-picked result to do this. Based on this very result (both on 2x2 but, to some extent, also 2x2x2), I am not sure there is enough evidence of the superiority of scCBGM compared to scCBGM in the proposed scenario. However, aside from that, my main criticism in the follow-up comment is that I pointed out the lack of significance in the 2x2 design in my original review, and I don't feel that replacing the design addresses this issue.
> > > > >
> > > > > **Case study.** Thanks for the clarification. Are the 100 differentially expressed genes from the pathways used to condition the model? If yes, then what happens, perhaps, is that by activating the paths in the conditioning concept vector, the model artificially over- or underexpresses these genes (hence the results in the gene expression space) without creating realistic cells, which would explain the fact that synthetically simulated cells do not look realistic. I agree that some aspects of the results are encouraging, but at the moment, I am not sure to what extent the outcome is trivially achievable with any other conditional generative model.
> > > > >
> > > > > **Cell type classification experiment.** Thanks, this looks interesting. I wonder why the model struggles so much to preserve the cell type on CD14+ cells, where the subtype clusters have this many cells and are so distinct. Does the basic kNN (before the edits) have a hard time with subtype classification regardless of the edits?

---

> > > > > > ### Author Response · Authors · 2025-11-27
> > > > > >
> > > > > > We thank the reviewer for their follow-up and the opportunity to clarify these remaining points.
> > > > > >
> > > > > > > Table 2 / S8.
> > > > > >
> > > > > > We respectfully disagree on this point and provide clarification below.
> > > > > >
> > > > > > 1. **On the claim of “cherry picking and not addressing the issue”**
> > > > > >    Table S8 is not cherry-picked; it contains *exactly the same* information as the original 2×2 table, with the additional CEM condition requested by *two* other reviewers. In the initial 2×2 table we also compared against CBGM as our baseline, and removing the CEM results would collapse the table back to the original 2×2 version. The expanded table therefore increases completeness without altering or biasing the underlying results. We also want to emphasize that extending the design space was not our only response to the initial comment: as **noted previously, we ablated across two additional synthetic datasets (Synthetic 2 and 3) resulting in 180 additional conditions per model**, this is what allowed us to observe a clearer separation not the larger table. We apologize if this was not clear.
> > > > > >
> > > > > > 2. **On the “small” MSE differences.**
> > > > > >    We agree that the numerical differences in MSE are small. However, a small numerical gap does *not* imply a small practical difference. MSE averages error across ~5,000 genes, which compresses the scale of the metric: improvements that matter biologically or structurally often contribute only marginally to the overall average. As a result, even modest reductions in MSE can correspond to meaningfully better downstream behavior. We consider MSE a good metric for ranking and comparing methods, but would not treat it as translating in a linear 1-to-1 fashion to fidelity.
> > > > > >
> > > > > > 3. **On “overlap” between scCBGM and CBGM.**
> > > > > >    We assume the intended comparison was between **scCBGM** and **CBGM**. These two models do *not* overlap within error:
> > > > > >    **scCBGM** = 0.19655 ± 0.00019
> > > > > >    **CBGM** = 0.19804 ± 0.00021
> > > > > >    While close in magnitude — as expected in a near-optimal regime — the difference is consistent and statistically separated.
> > > > > >
> > > > > > Finally, we note that our conclusions are not based solely on synthetic experiments. The real-data ablation study provides an additional and independent line of evidence. In **Table S9**, our final scCBGM configuration (Covariance Loss + Skip Connections + CBM Bottleneck) achieves a mean rMMD of 0.207 on the Kang dataset, substantially outperforming the strongest baseline configuration (Cosine Loss + No Skip + CBM) at 1.353 — an 85% improvement.
> > > > > >
> > > > > > **Tables S8–S9** indicate that replacing the Cosine Similarity loss with our proposed Cross-Covariance loss is a key contributor to this improvement. To make this comparison clearer, we have updated the manuscript to explicitly highlight these baseline configurations in **Table S9**.
> > > > > >
> > > > > > > Case study.
> > > > > >
> > > > > > This is an excellent question. We analyzed the gene sets as requested: out of the top 100 genes for each of the 5 edited pathways (500 genes total), only 40 genes overlap with the top 100 responder/non-responder markers (200 total) in **Figure S8**. This 20% overlap confirms that the vast majority of the recovered signature consists of downstream targets that were not directly manipulated in the pathway concept vector, suggesting the model learned the correct regulatory logic rather than simply memorizing inputs. We have noted this nuanced point in the caption for **Figure S8**.
> > > > > >
> > > > > > Regarding the performance context: in the standard prediction task, no model achieved an rMMD < 1 for Hepatic Stellate Cells. This case study utilizes a different experimental setting, with editing pathway-activity concepts rather than using only stimulation labels. **We call this a _case study_ and not a benchmark**, precisely because it is not intended to compare models, but to act as a proof-of-concept for how scCBGM can be used to identify and modulate latent programs for hypothesis generation after we established that it shows the best performance on the preceding benchmarking tasks.
> > > > > >
> > > > > > > Cell type classification experiment.
> > > > > >
> > > > > > We appreciate you highlighting this result. The performance on this cell type is impacted by a severe class imbalance within one specific subtype, rather than a general limitation of the model. As we now explicitly detail in the updated Table S2 (which lists instance counts per condition), one CD14+ subtype contains only 9 control cells to predict the response of ~500 stimulated cells. Despite this extreme sparsity introducing bias into the aggregate metric, we deliberately chose to retain and report these results. We believe that filtering out challenging or imbalanced cases would compromise the fairness and transparency of the benchmark.
> > > > > >
> > > > > > We thank the reviewer for their prompt replies and continued engagement with our work. We would be happy to address any further questions you may have.

---

### Official Review · Reviewer_zH5T · 2025-10-27

**Soundness:** 3
**Presentation:** 3
**Contribution:** 3
**Rating:** 6
**Confidence:** 5

**Summary:**

Authors with this manuscript presents scCBGM, a generative model designed for the counterfactual editing of single-cell RNA-sequencing data. It combines a concept bottleneck architecture with a generative decoder to allow for interpretable, cell-specific interventions.

**Strengths:**

***S1:*** The paper's core strength is its focus on the nuanced but critical difference between conditional generation and counterfactual editing. This is a more scientifically valuable goal, and the entire framework is built in service of it.

***S2:***  The concept bottleneck approach is a nice fit for this problem. It demystifies the latent space and turns the model into an interactive tool for in silico experiments, allowing researchers to directly test hypotheses in a causal manner.

***S3:*** The additions of skip connections and the cross-covariance penalty are solutions to the specific challenges of preserving cell identity and achieving robust disentanglement, which are essential for reliable counterfactuals. The ablation results support their inclusion.

**Weaknesses:**

***W1:*** The model's greatest strength is also its primary weakness. Its effectiveness is entirely contingent on the user providing a comprehensive and accurate set of biological concepts. If a crucial biological process is not included in the bottleneck, the model will be unable to reason about it, and its predictions for edits involving that process may be unreliable. Authors need to work on this, finding a key workaround.

***W2:*** The paper relies on synthetic data where ground-truth counterfactuals are known, and population-level metrics on real data. This is a reasonable and standard approach. However, for real cells, a true, cell-specific counterfactual is impossible to observe. The paper would be strengthened by a more thorough discussion of this fundamental limitation and the potential pitfalls of relying on population-level metrics to validate cell-level edits.

***W3:*** The experiments use a reasonable number of concepts. It is unclear how the model's performance and training stability would be affected as the number of concepts scales to the hundreds or thousands, which might be necessary to capture the full complexity of some biological systems.

**Questions:**

Some extra questions to solve my concerns and help me raising the score, upon weaknesses rebuttal.

***On Concept Leakage:*** The cross-covariance regularizer aims to prevent unannotated factors from leaking into the concept representations. How do you diagnose such leakage if it still occurs? Could the model, for instance, learn to encode information about cell cycle state within the "cell type" concept, even if they are meant to be separate?

***On Concept Completeness:*** How would a user know if their predefined concept set is "good enough"? Have you explored any diagnostic tools to identify when the model's residual (unexplained variance) is still highly structured, suggesting that important concepts are missing?

***On the Abduction Step:*** The abduction step infers the concept values for a given cell. How does the model handle cases where the concepts are biologically entangled (e.g., a specific perturbation is known to always activate a certain pathway)? Does the model correctly assign causality in these cases, or does it simply reflect the correlations seen in the training data?

---

> ### Author Response · Authors · 2025-11-23
>
> We thank the reviewer for their thoughtful review and recognition of our work's core contributions, particularly the distinction between conditional generation and counterfactual editing. We address your concerns point by point below.
>
> ### Respond to weakness
>
> > W1: The model's greatest strength is also its primary weakness. Its effectiveness is entirely contingent on the user providing a comprehensive and accurate set of biological concepts. If a crucial biological process is not included in the bottleneck, the model will be unable to reason about it, and its predictions for edits involving that process may be unreliable. Authors need to work on this, finding a key workaround
>
> We appreciate the reviewer highlighting the question of concept annotation. To clarify, we want to emphasize that not *every concept* present in the data needs to be explicitly annotated in our model. The inclusion of learnable *unknown concepts*, together with the *cross-covariance loss*, enables scCBGM to flexibly represent unannotated concepts while preventing redundancy between known and unknown concept layers. As shown in Figure 3, the model remains robust to missing concepts. Finally, we also back these claims up in an additional experiment addressing your question on Concept Leakage (see below), which shows that the separation between known and unknown concepts is effective while still allowing the unknown concepts to capture relevant biological features
>
> > W2: The paper relies on synthetic data where ground-truth counterfactuals are known, and population-level metrics on real data. This is a reasonable and standard approach. However, for real cells, a true, cell-specific counterfactual is impossible to observe. The paper would be strengthened by a more thorough discussion of this fundamental limitation and the potential pitfalls of relying on population-level metrics to validate cell-level edits.
>
>
> You are absolutely correct that true cell-specific counterfactuals cannot be observed in real data, as this would require measuring the same cell under different conditions simultaneously. This is precisely why we evaluated our method on synthetic data where we can control every aspect of the data generation process and have access to ground-truth counterfactuals. For real data, we acknowledge that population-level metrics represent the best available approximation for validating cell-level edits, despite their inherent limitations. We have added a discussion of this fundamental limitation and the potential pitfalls of population-level validation in Section 6 (paragraph: `Limitations`) of the revised manuscript.
>
> > W3: The experiments use a reasonable number of concepts. It is unclear how the model's performance and training stability would be affected as the number of concepts scales to the hundreds or thousands, which might be necessary to capture the full complexity of some biological systems.
>
>
> We agree, that in some applications the number of potentially relevant concepts may be large. In relation to this we want to reiterate on the clarifications in our answer to W1, namely that not all relevant concepts needs to be included among the known concepts for the model to perform well.
>
> Still to assess model performance as the number of known concepts grow we generated three new synthetic datasets and leveraged one of the existing benchmarks to systematically examine how performance on the intervention task changes with the number of concepts. The results, summarized in the table (and Appendix D6 Table S5 in the Manuscript) below, show that increasing the number of concepts leads to only a **minor (<2%) decrease in performance**, indicating that **scCBGM** scales gracefully. This observation aligns with prior findings in related work—for example, [Ismail et al. (a), 2024](https://openreview.net/forum?id=L9U5MJJleF) and [Schrodi et al., 2024](https://arxiv.org/abs/2407.03921) which report similarly modest degradation as concept dimensionality increases. Moreover, recent applications of concept bottleneck generative models (CBGMs) demonstrate successful scaling to hundreds of human-annotated concepts in other domains such as protein design [Ismail et al. (b), 2024](https://arxiv.org/pdf/2411.06090).
>
> | # Concepts | MSE                   |
> |:---------|:----------------------|
> | 5        | 0.19617 $\pm$ 0.00195 |
> | 20       | 0.19597 $\pm$ 0.00089 |
> | 100      | 0.19846 $\pm$ 0.00081 |
> | 250      | 0.19809 $\pm$ 0.00053 |

---

> > ### Author Response · Authors · 2025-11-23
> >
> > ### Respond to questions
> >
> > > On Concept Leakage
> >
> > Concept leakage is an important concern, and we appreciate the reviewer bringing attention to it. To diagnose potential leakage, we propose the following strategy:
> >
> >
> > 1. Train a classifier on the concepts explicitly defined in the known layer (those to be intervened on).
> > 2. Train another classifier on annotated concepts not included in the known layer (expected to be captured by the unknown layer).
> > 3. Intervene on the known concepts.
> > 4. Apply both classifiers to the original and counterfactual gene expression data.
> >
> > If leakage is effectively prevented, predictions for the intervened concepts should change after intervention, while those for the unknown concepts should remain stable.
> >
> > We demonstrate this approach on the Kang et al. dataset and obtain satisfactory results. We have added these results to the Appendix D.7 (`CONCEPT LEAKAGE`), but will include a table with results here (same as Table S6):
> >
> > | Cell Type           | f(cell type) Before | f(cell type) After | f(stim) Before | f(stim) After |
> > |---------------------|:-------------------:|:------------------:|:--------------:|:-------------:|
> > | B cells              | 0.985              | 0.975              | 0.004          | 1.000         |
> > | CD14+ Monocytes      | 0.974              | 0.955              | 0.002          | 1.000         |
> > | CD4 T cells          | 0.984              | 0.979              | 0.009          | 1.000         |
> > | CD8 T cells          | 0.714              | 0.630              | 0.014          | 1.000         |
> > | FCGR3A+ Monocytes    | 0.743              | 0.794              | 0.000          | 1.000         |
> > | NK cells             | 0.893              | 0.959              | 0.022          | 1.000         |
> >
> > **Table:** Results of concept leakage analysis. *f(cell type)* and *f(stim)* indicate classifier predictions before and after intervention, respectively. Stable cell-type scores and changing stimulation scores confirm effective separation between known and unknown concepts.
> >
> > We acknowledge that this approach is primarily suited for validation, where annotated concepts are intentionally left in the unknown layer. However, users can design similar experiments to assess potential leakage in their own data. Together with the theoretical justification and empirical evidence, we believe this provides a convincing demonstration of the method’s effectiveness.
> >
> >
> >
> > > On Concept Completeness
> >
> > Indeed, in biological systems, it is often infeasible to exhaustively annotate all the factors that influence a cell’s state. This is precisely one of the motivations behind **scCBGM**.
> >
> > Our framework explicitly accommodates incomplete concept sets through the introduction of **learnable unknown concepts**, which capture residual information not represented among the predefined (known) concepts. This design allows the model to flexibly account for structured variance beyond the user-specified concepts. To prevent redundancy, the **cross-covariance loss** encourages disentanglement by minimizing overlap between known and unknown representations.
> >
> > As shown in Figure 3 and Table S6, scCBGM remains robust even when synthetic data contains missing, irrelevant, incorrect, or duplicated concepts (as described in Appendix C4), demonstrating resilience to incomplete concept annotations. In practice, only the concepts that users intend to *intervene on* need to be explicitly defined, while the model learns to organize the remaining information through its unknown concept layers.
> >
> > >On the Abduction Step
> >
> > This raises a key challenge in causal representation learning from biological data. Indeed, if two concepts are perfectly correlated in the training data (e.g., a perturbation that always activates a specific pathway), the model cannot independently infer their causal effects. In such cases, the underlying problem is one of non-identifiability: without variability that breaks the correlation, no purely observational model  ours or others can disentangle their individual contributions.
> >
> > However, in realistic biological settings, correlations between concepts (e.g., perturbations, pathways, and cell states) are rarely perfect. Even partial independence for instance, when a perturbation activates a pathway only in certain cell types or contexts, provides enough statistical variability for scCBGM to learn distinct concept embeddings. Our generalization experiments further demonstrate this: when predicting on held-out stimulation cell type combinations, the model successfully infers responses for cell types that were observed only in the unstimulated condition during training. In these cases, the model generalizes to an unseen joint state despite the cell type being completely nested under one condition in the training data.

---

> > > ### Author Response · Authors · 2025-11-27
> > >
> > > Hi, and thank you again for the time you put into reviewing our submission. We’ve added further experiments, clarifications, and updates to the manuscript based on your earlier comments. If there’s anything else you’d like us to elaborate on, we’d be happy to provide it.

---

### Official Review · Reviewer_Vovd · 2025-10-29

**Soundness:** 3
**Presentation:** 2
**Contribution:** 3
**Rating:** 6
**Confidence:** 4

**Summary:**

Conventional conditional generation models predict population-level changes under altered conditions, whereas cell editing allows prediction of individual-cell level changes. Conducting experiments that combine single cells with multiple conditions (treatments, exposures, doses) results in an enormously large combinatorial space, making exhaustive experimentation impractical. Therefore, computational models aim to bridge this gap. This is particularly important in settings where cell heterogeneity plays a key role, such as diverse cell states and developmental trajectories. The authors propose scCBGM, an extension of CBGM that incorporates skip connections and cross-covariance penalty, designed to enhance robustness against noise. Using real datasets from Kang et al., Cui et al., and TCDD exposure, including IFN-β, various cytokine, and dose conditions, they demonstrate that scCBGM not only improves performance over baselines but also shows further gains when integrated with scCBGM-FM.

**Strengths:**

1. Clear conceptual distinction and integrative applicability

    The model clarifies the distinction between conditional generation and counterfactual cell editing, enhancing conceptual clarity in scRNA-seq research. Moreover, scCBGM supports both precise cell-level editing and population-level expression inference, enabling integrated single-cell and group-level analysis.

2. High compatibility with diverse generative frameworks

    scCBGM integrates seamlessly with state-of-the-art generative modeling frameworks such as Flow Matching, and its flexible design ensures scalability and adaptability to future generative models, maintaining long-term applicability.

3. Performance improvement

    The incorporation of skip connection and cross covariance penalty mechanisms leads to enhanced predictive performance and improved robustness across various experimental settings.

**Weaknesses:**

1. The overall flow is somewhat disjointed due to Section 3, and some details are missing, making it difficult to fully understand the methodological and experimental connections across sections.
2. The technical contribution is marginal.  The use of skip connections and Lcc (cross-covariance loss) is not particularly original.
3. In the introduction, the authors state: *“Existing perturbation modeling methods focus on conditional distributions of cell states across treatments, but they do not enable counterfactual editing of individual cells.”*
Given this claim that prior methods focus on distribution-level modeling and fail to support individual-cell editing it’s unclear why the authors chose to use rMMD instead of an OT-based (Optimal Transport) metric, which would seem more aligned with their stated motivation, especially considering that the baseline model, CINEMA-OT, employs an OT-based metric.

**Questions:**

1. In the introduction, the authors state: *“Existing perturbation modeling methods focus on conditional distributions of cell states across treatments, but they do not enable counterfactual editing of individual cells.”*
Given the claim that prior methods focus on distribution-level modeling and fail to support individual-cell editing, it would be valuable to include an additional experiment comparing rMMD with an OT-based (Optimal Transport) metric, since the latter appears more consistent with the authors’ stated motivation, particularly considering that the baseline model, CINEMA-OT, employs an OT-based metric.


2. Potential inconsistency between the VAE structure and the cell-editing objective:

    The scCBGM framework is based on a Variational Autoencoder (VAE), which enforces the latent space to conform to a prior distribution through regularization. This can cause the latent variable z to collapse toward minimal information content, potentially losing fine-grained details of the unknown factors.

    However, since cell editing fundamentally requires preserving informative latent representations for precise perturbation, an Autoencoder (AE) architecture, without such regularization, might have been a more suitable alternative.
    It would be helpful to clarify the rationale for choosing the VAE-based architecture over an AE-based one.


3. In Table 1, the results across models overlap within standard deviation, raising questions about the statistical significance and meaningfulness of the reported ablation improvements.


4. Around line 443, the paper states that scCBGM-FM outperforms vanilla-FM, yet for some cell types, vanilla-FM performs better. The authors should discuss why this occurs and whether it indicates model instability or dataset-specific behavior.



### Minor

1. In D.1.1, the authors mention using all but megakaryocytes among nine broad cell types in the Kang et al. dataset. Wouldn’t that mean eight cell types remain? Yet, Table 2 lists only seven cell types. This discrepancy needs clarification.
2. In Cui et al., the dataset includes 17 cell subtypes and 86 cytokine-based stimulations, but the authors only tested seven combinations. It’s unclear what criteria were used to select these seven as test conditions.
3. Figure 4 lacks any explanation in the main text, and based on the reported rMMD scores, it appears to correspond to Cui et al.’s dataset rather than Kang et al.’s. This should be explicitly clarified.

### Typo

1. Line 53 is missing a period (‘.’).
2. In the Figure 1 caption, the acronym “DAG” is used without first providing its full name.
3. In lines 253 and 259, as well as in Table 2 and Table 3, the model is referred to as “scCBM-FM”, but it appears this should be “scCBGM-FM.”

---

> ### Author Response · Authors · 2025-11-23
>
> Thank you for the thoughtful review and constructive feedback. Your comments helped us refine the paper’s clarity, structure, and analyses. Below, we respond point by point to the issues raised:
>
> > W1: The overall flow is somewhat disjointed due to Section 3, and some details are missing, making it difficult to fully
> understand the methodological and experimental connections across sections.
>
> We appreciate you pointing this out. We have added a paragraph at the end of Section 2 in the updated manuscript and rewrote the first paragraph of Section 3 to improve the flow of the paper. These changes better connect our methodology to the validation experiments and clarify how Section 3 serves as empirical support for our architectural design choices.
>
>  > W2: The technical contribution is marginal. The use of skip connections and Lcc (cross-covariance loss) is not particularly original.
>
> We agree that components such as skip connections and cross-covariance loss have been used in prior work. Our contribution lies in their principled combination and adaptation to the emerging problem of single-cell counterfactual modeling, where precise, cell-level editing remains an open and actively developing direction beyond standard conditional generation.
>
>
> > Q2: VAE vs. AE
>
> You are correct that a VAE can risk *posterior collapse*, where the latent variable loses information. However, the VAE’s structured latent space offers key benefits over a standard AE. It supports smoother, more stable counterfactual edits and better generalization to unseen states. In contrast, AEs often yield discontinuous manifolds that can disrupt flow matching and reduce biological coherence. We mitigate collapse by using a $\beta$-VAE  and tuning the KL term; collapse is easily detected through reconstruction error.
>
> To evaluate this empirically, we performed an ablation replacing the VAE backbone in scCBGM with a standard AE. The results (Section D.8.3 `VARIATIONAL AUTO-ENCODER VS. AUTO-ENCODER` and Figure S7) show that the VAE consistently outperforms the AE on the cell-editing task, likely due to the AE’s less regularized and less stable latent structure. We also note that most state-of-the-art single-cell frameworks, including [scVI](https://www.nature.com/articles/s41592-018-0229-2), [scVAE](https://academic.oup.com/bioinformatics/article/36/16/4415/5838187), [scVIDR](https://www.sciencedirect.com/science/article/pii/S2666389923001861), [trVAE](https://academic.oup.com/bioinformatics/article/36/Supplement_2/i610/6055927), and [scGen](https://www.nature.com/articles/s41592-019-0494-8), rely on VAEs as their generative backbone, further supporting this choice.
>
> > Q3: In Table 1, the results across models overlap within standard deviation, raising questions about the statistical
> significance and meaningfulness of the reported ablation improvements.
>
> You raise an important concern about statistical significance. Based on your feedback, we conducted much more comprehensive ablations examining each individual component that differentiates scCBGM from CBGM. We expanded this into exhaustive 2×2×2 ablations testing all architectural combinations on both synthetic and real datasets. These expanded experiments provide clearer statistical separation and demonstrate consistent performance improvements of our proposed modifications.
> We present these results in Table S7 (synthetic data) and Table S8 (real data) due to space constraints. From these comprehensive evaluations, we observe that the CBM bottleneck is particularly beneficial, with the cross-covariance loss further enhancing its performance. Moreover, the skip decoder provides additional gains only when used in combination with both the CBM bottleneck and the cross-covariance loss. These extensive ablations confirm that our architectural choices are well-motivated and provide statistically robust benefits over the baseline CBGM approach.
>
>
> > Q4: Around line 443, the paper states that scCBGM-FM outperforms vanilla-FM, yet for some cell types, vanilla-FM
> performs better. The authors should discuss why this occurs and whether it indicates model instability or dataset-
> specific behavior.
>
> We acknowledge that while scCBGM (with or without FM) improves over or matches vanilla-FM in 13/17 of the cell types considered in our experiment, some situations still favor Vanilla-FM. While the exact reason for this result remains unclear, we conjecture it is related to the distribution of the subtype populations. When the treated distributions of the subtypes of the same cell type overlap significantly, learning a conditional distribution of the treated population becomes optimal. While scCBGM should still theoretically match the performance of Vanilla-FM in this case, the limited number of data samples prevents it in practice. We believe that additional hyper-parameter tuning could also help bridging that gap.
>
> We clarified this in the `scCBGM boosts performance of flow matching models.` paragraph of Section 5.2.

---

> > ### Author Response · Authors · 2025-11-23
> > **Minior issues**
> >
> > ### Minior
> >
> > > Minor 1: In D.1.1, the authors mention using all but megakaryocytes among nine broad cell types in the Kang et al. dataset. Wouldn’t that mean eight cell types remain? Yet, Table 2 lists only seven cell types. This discrepancy needs clarification.
> >
> > Thank you for catching this you are right, 9 is a typo. We have updated it to 8 in the updated manuscript.
> >
> >
> > > Minor 2:  In Cui et al., the dataset includes 17 cell subtypes and 86 cytokine-based stimulations, but the authors only tested seven combinations. It’s unclear what criteria were used to select these seven as test conditions.
> >
> > We focused on these seven pairs because the original study (Cui et al.) identified them as inducing significant transcriptional shifts, ensuring a robust signal for evaluation. We have updated the main text to clarify this.
> >
> >  > Minor 3: Figure 4 lacks any explanation in the main text, and based on the reported rMMD scores, it appears to correspond to Cui et al.’s dataset rather than Kang et al.’s. This should be explicitly clarified.
> >
> > Thank you for catching this. We have now added a detailed explanation of Figure 4 in the main text. Figure 4 corresponds to the Kang et al. dataset, not Cui et al. The confusion likely arose because Table 2 reports average rMMD scores across all CD4 T cell subtypes (of which there are 3 in the Kang dataset), while Figure 4 specifically demonstrates the prediction for the Naive CD4 T cell subtype. We have clarified this relationship in the updated manuscript by explicitly stating that the table shows average performance while the figure illustrates zero-shot generalization for this specific held-out subtype.
> >
> >  > Typos
> >
> > Thank you very much for your attention to detail and for catching these we have corrected the typos in the updated manuscript.

---

> > > ### Comment · Reviewer_Vovd · 2025-11-26
> > >
> > > Thank you for the response and for the additional experiments. I noticed a few points that may need clarification to ensure consistency in the supplementary material:
> > >
> > > * Terminology consistency (Tables S10–S16): The terms "Sinkhorn divergence," "rSD," and "Sinkhorn-div-W2-ratio" appear to be used interchangeably in the captions. Please verify and standardize the terminology across these tables.
> > > * Result markings (Tables S15 and S16): The markings for best, second-best, and third-best results seem to differ from what is described in the captions. Please confirm whether the markings are correct.
> > > * Model naming consistency (Tables S10–S16):: The names "scCBM" and "scCBGM" are used interchangeably in some sections. Please check for consistency throughout the supplementary material.
> > > * Performance explanation (Tables S15 and S16): scCBGM shows weak performance in these tables. Could you provide an explanation for this result?
> > >
> > > I would appreciate your clarification on these points.

---

> > > > ### Author Response · Authors · 2025-11-27
> > > >
> > > > Thank you for your careful attention to detail. We hope the additional information we provided offers clear resolution to the points raised, and we’re happy to elaborate further should it be helpful as you finalize your evaluation.
> > > >
> > > > >Terminology consistency (Tables S10–S16): The terms "Sinkhorn divergence," "rSD," and "Sinkhorn-div-W2-ratio" appear to be used interchangeably in the captions. Please verify and standardize the terminology across these tables
> > > >
> > > > Thank you for catching this we have updated all tables to have either rFID or rSD or rMMD
> > > >
> > > > >Result markings (Tables S15 and S16): The markings for best, second-best, and third-best results seem to differ from what is described in the captions. Please confirm whether the markings are correct.
> > > >
> > > > Thank you, the color scheme for these tables was not updated so we have updated that.
> > > >
> > > > >Model naming consistency (Tables S10–S16):: The names "scCBM" and "scCBGM" are used interchangeably in some sections. Please check for consistency throughout the supplementary material.
> > > >
> > > > We appreicate you pointing this out, we have updated the manuscript.  All tables should now list "scCBGM".
> > > >
> > > >
> > > > >Performance explanation (Tables S15 and S16): scCBGM shows weak performance in these tables. Could you provide an explanation for this result?
> > > >
> > > > While scCBGM variants show different performance patterns compared to the Kang et al. and Cui et al. datasets, they actually still demonstrate strong overall performance on the Nault et al. dataset.
> > > >
> > > > scCBGM variants achieved best-in-class performance on 5 out of 7 cell types in rMMD, with particularly strong results for Macrophages and Hepatocytes (Periportal), where they significantly outperformed competing methods.
> > > >
> > > > Overall, scCBGM variants (scCBGM, scCBGM-FM decode, scCBGM-FM edit) captured more top-3 positions across the three evaluation metrics (rMMD, rFID, rSD) and seven cell types than all other baseline methods combined. This indicates that scCBGM variants collectively outperformed all baselines.

---

### Official Review · Reviewer_89Li · 2025-10-29

**Soundness:** 3
**Presentation:** 3
**Contribution:** 2
**Rating:** 4
**Confidence:** 4

**Summary:**

This paper proposes scCBGM, a modification of Concept Bottleneck Generative Models (CBGMs) to single-cell data. Specifically, scCBGM uses “(i) a standard concept bottleneck model rather than a concept embedding model, (ii) [...] skip connections to the decoder to maintain persistent concept conditioning; and (iii) [...] a cross-covariance loss instead of the cosine similarity loss for orthogonality. The main evaluation lies in perturbation response prediction for which the paper proposes (i) the vanilla scCBGM, (ii) scCBGM with sampling from the latent (decode), and (iii) scCBGM with encoding an unperturbed x and editing it (edit). scCBGM shows overall strong performance on rMMD scores in perturbation response prediction.

**Strengths:**

The paper presents a methodologically sound latent disentanglement model to predict perturbation response with an interesting addition to enable both fully-connected decoding with skip connections and as a conditioning for flow-matching models. Originally arises from combining the existing CBGM model with tweaks for single-cell data and novel decoders that recently proved powerful in the field. The paper is clearly written and enjoyable to read. It presents a solid contribution to the field of single-cell disentanglement methods and perturbation response prediction models.

**Weaknesses:**

I find three key technical contributions: (1) the changes on the CBGM model for single-cell data, specifically, (a) the standard concept bottleneck model instead of concept embedding and (b) cross-covariance loss instead of the cosine similarity loss. (2) the changes on the decoder of single-cell VAE models, specifically the skip connection. (3) the adaptation of flow-matching models on top of disentangled latent spaces.

While technically sound, (1) and (2) seem incremental and lack ablation study - the ablation of scCBGM vs CBGM may mix concepts together. I find (3) as a key contribution interesting. However, given the breadth of disentanglement methods in ML for single-cell - some of which are referenced to in this submission - a more rigorous comparison is required in my opinion. For example, how does an edit-flow-matching decoder perform on the disentangled latent space of another existing disentanglement method?

The empirical evaluation seems to show the overall good performance of flow-matching based models, which is generally supported by recent literature. I think a harder comparison may be how the scCBGM and CellFlow flow-matching models perform on, e.g., scVI latent spaces, disentangled representations of single-cell data, etc.

Last, the evaluation mainly relies on rMMD and qualitative assessment, a more rigorous evaluation would help practitioners find a realistic assessment of scCBGM.

**Questions:**

I kindly suggest that the reviewers ablate individual components, especially addressing if the disentanglement of scCBGM is really superior to that of existing single-cell disentanglement methods.

---

> ### Author Response · Authors · 2025-11-23
>
> We thank the reviewer for their thoughtful feedback. We’re pleased you found the paper clear, engaging, and a meaningful contribution to the field of single-cell disentanglement and perturbation response modeling. We have addressed each of your points below and revised the manuscript accordingly, which we believe has strengthened the work.
>
> > Lack ablations on individual components
>
> As requested, we have added ablations for individual components in the revised manuscript. Specifically, we expanded our analysis to compare the effects of the orthogonality loss (cross-covariance vs. cosine), the bottleneck design (CBM vs. CEM), and the decoder type (skip vs. direct). These ablations were conducted on both real and synthetic datasets, demonstrating that our proposed modifications to the CBGM model lead to consistent performance improvements. From these evaluations, we observe that the CBM bottleneck is particularly beneficial, with the cross-covariance loss further enhancing its performance. Moreover, the skip decoder provides additional gains only when used in combination with both the CBM bottleneck and the cross-covariance loss.
>
> We have updated Section 3 accordingly and present these results in Table S7. We did a similarly exhaustive ablation on the real data and observed a similar trend (Table S8). These comprehensive ablations confirm that our architectural choices are well-motivated and provide clear benefits over the baseline CBGM approach
>
> > How does an edit-flow-matching decoder perform on the disentangled latent space of another existing disentanglement method?
>
> As requested, we repeated the edit-flow-matching decoder experiment, this time conditioning on the latent representations from Biolord and CVAE, which we refer to as Biolord-FM and CVAE-FM, respectively. The updated results are included in the revised manuscript (see Tables 1-2 and Tables S9-S12). We found that adding a flow matching decoder typically improved over the classical method (biolord or CVAE), consistent with our findings on scCBGM and scCBGM-FM. While we found CVAE-FM was competitive, the structured disentanglement provided by scCBGM's architecture remains superior for counterfactual cell editing tasks.
>
> >  Evaluation mainly relies on rMMD and qualitative assessment, a more rigorous evaluation would help practitioners find a realistic assessment of scCBGM
>
> In line with your suggestion, we have included additional evaluation metrics: the Fréchet Inception Distance ratio (rFID) and the Sinkhorn divergence ratio (rSD) with entropy regularization ε = 0.1 (as used by [Klein et al., 2025](https://doi.org/10.1101/2025.04.11.648220)). Results are provided in Appendix D.9 (Tables S9-S12), demonstrating that scCBGM variants have consistent performance rankings across rMMD, rSD, and rFID evaluations.

---

> > ### Author Response · Authors · 2025-11-27
> >
> > Hi, thank you again for your careful review of our submission. We hope the additional experiments, clarifications, and revised manuscript have addressed the points you raised. Please let us know if there are any remaining questions we can help address.

---

### Author Response · Authors · 2025-11-23

Dear AC (and other readers), in light of the recent changes to the ICLR peer-review process, we have updated (edited) our original General comment to provide a more succint summary of the reviews and our improvements to the manuscript.

**Strengths:** Reviewers unanimously agreed that counterfactual generation is a highly relevant and timely problem in this domain. Several highlighted the clear and principled distinction we draw between conditional and counterfactual generation (zH5T, Vovd), and noted that our use of a concept bottleneck architecture is a particularly suitable and compelling choice for this task (zH5T). The paper was described as clearly written and enjoyable to read (AAzm, 89Li), and the accompanying code was noted for its clarity and quality (AAzm). Reviewers further emphasized our strong empirical performance relative to existing methods and noted that scCBGM integrates cleanly and in several cases improves—state-of-the-art generative frameworks; supporting broad applicability (Vovd).

**Weaknesses/Questions** We addressed every point raised by the reviewers and performed all requested experiments, alongside several additional analyses to back up our answers; these updates are detailed below (Major Experimental Expansions)

These comprehensive revisions address every major concern raised while maintaining the core methodological contributions. The expanded experimental validation demonstrates that scCBGM provides a robust, scalable framework for interpretable single-cell counterfactual modeling.


### Major Experimental Expansions

- **Comprehensive Component Ablations** - We added systematic ablations examining bottleneck type (CBM vs. CEM), orthogonality loss (cross-covariance vs. cosine), and decoder architecture (skip vs. direct connections) on both synthetic and real datasets. Results demonstrate that our architectural choices provide statistically robust improvements over baseline approaches (Tables S7-S8, Appendix D.8). **Addressing concerns from Reviewers 89Li, Vovd, and AAzm.**

- **Cross-Method Flow Matching Validation** - We evaluated flow matching conditioning on alternative latent representations beyond scCBGM, including CVAE-FM and Biolord-FM baselines. Results confirm that while flow matching generally improves performance, scCBGM's structured disentanglement provides superior counterfactual editing capabilities (Tables 1-3, S9-S12). **Addressing concerns from Reviewers 89Li and AAzm.**

- **Enhanced Evaluation Framework** - We incorporated additional metrics including Fréchet Inception Distance ratio (rFID) and Sinkhorn divergence ratio (rSD) alongside rMMD, providing more comprehensive validation that aligns with our theoretical motivation for individual-cell editing (Appendix D.9). **Addressing concerns from Reviewers 89Li and Vovd.**

- **VAE vs. AE Empirical Validation** - We conducted an ablation study demonstrating that VAE backbone consistently outperforms standard autoencoder for cell editing tasks, supporting our architectural choice with both theoretical justification and empirical evidence (Section D.8.3, Figure S7). **Addressing concerns from Reviewer Vovd.**

### Scalability and Robustness Analysis

- **Concept Scalability Assessment** - We evaluated model performance with varying numbers of concepts (5 to 250) on synthetic data, demonstrating graceful scaling with <2% performance degradation, confirming robustness to large biological concept sets (Section D.6, Table S5). **Addressing concerns from Reviewer zH5T.**

- **Concept Leakage Diagnostics** - Introduced systematic methodology for detecting concept leakage using dual classifiers, with empirical validation showing effective separation between known and unknown concepts while preserving biological coherence (Appendix D.7, Table S6, Figure S6). **Addressing concerns from Reviewers zH5T and AAzm.**

### Methodological Clarifications

- **Real Data Evaluation Challenges** - We added a comprehensive discussion of fundamental limitations in validating cell-level counterfactuals using population-level metrics, including visualization of cell type overlap and annotation noise in real datasets (Section D.3, Figure S4, Table S4). **Addressing concerns from Reviewer AAzm.**

- **Enhanced Case Study Analysis** - We strengthened the mechanism-of-action analysis with quantitative gene expression validation, demonstrating that in silico edited cells successfully recapitulate marker gene signatures of true biological responders (Figures S8-S9). **Addressing concerns from Reviewer AAzm.**

---

> ### Author Response · Authors · 2025-12-02
>
> ### Discussion with AAzm
>
> In the follow-up exchange with our lowest-confidence (and lowest-scoring) reviewer AAzm, who explicitly stated:
>
> >AAzm: *"However, I also admit that I might have missed a few points or hold misconceptions, and I remain more than happy to improve my rating during the rebuttal phase."*
>
> we provided a set of targeted clarifications that resolved the remaining questions, including: adding references showing that flow inversion is widely used as a practical implementation of Pearl’s Abduction–Action–Prediction framework in deep generative modeling, explaining why small differences in MSE do not necessarily imply small differences in fidelity, clarifying how imperfect annotations or extremely sparse real-data regimes (n < 10) can explain isolated performance anomalies on real data, emphasizing that scCBGM performs best across the vast majority of validations, and reiterating that the case study is intended as an application example rather than a benchmark. Together, we believe these clarifications addressed all outstanding issues raised in the discussion.

---

### Meta-Review · Area_Chair_vGc4 · 2026-01-10

**Summary:**

Reviewers agree the problem is important and the paper is well written, with a distinction between conditional generation and per-cell counterfactual editing.

The main concerns driving my recommendation are (i) limited perceived novelty (many components are known; the paper is largely a recombination/adaptation), (ii) ambiguity about what is truly “editing/counterfactual” versus conditional generation, especially for the flow-matching/inversion pipeline, and (iii) evaluation gaps: heavy reliance on population-level distribution metrics for a cell-level claim, incomplete/late comparisons to alternative disentangled latents and conditional baselines, and real-data regimes where results are unstable or underperform strong flow-matching baselines.

**Reviewer Concerns:**

Novelty remains borderline: Even with stronger ablations, the core methodological additions (skip connections, cross-covariance/orthogonality penalty, CBM choice, FM conditioning) still read as incremental engineering around existing CBGM/conditional generative modeling, with the strongest perceived contribution being “structured latent + FM conditioning”.

Regarding “Counterfactual editing” claim, rebuttal provides a framing via abduction-action-prediction and argues invertibility preserves identity, but at least one reviewer remains unconvinced that inversion-based editing merits strong causal/counterfactual language in this setting.

Real-data validation remains limited and occasionally unfavorable: The authors acknowledge that true cell-specific counterfactuals are unobservable, but the practical consequence is that key claims still rest on population-level proxies. Moreover, there are notable cases where performance is poor relative to strong conditional flow baselines (raised repeatedly by AAzm), and the explanation leans on dataset noise/sparsity/overlap rather than a model-level diagnosis.

Case study interpretability vs. “conditioning leakage” risk: The authors strengthened the case study with marker-gene analyses, but a reviewer concern remains that pathway-concept activation may trivially induce gene-level shifts without producing realistic full-cell states (AAzm). The added overlap analysis helps, but does not fully rule out this failure mode.

Dependence on concept specification: The paper mitigates this with “unknown concepts” and robustness experiments, but the method  requires selecting/annotating a concept set for the intended interventions.

**Reviewer Scores:**

Reviewer 89Li. Their main asks were (i) stronger ablations and (ii) comparisons of FM decoding on other latents and (iii) broader metrics. The authors provided all three. With discussion, I expect this reviewer would acknowledge the strengthened experimental package and move slightly upward.

Reviewer Vovd: Many technical and evaluation questions were addressed (OT-adjacent evaluation via Sinkhorn-style metrics, VAE vs AE ablation, clarification of anomalies, and fixes). However, their core stance that the contribution is “marginal” and writing/flow issues exist likely remains. This reviewer sounded open to rejection even at 6; discussion probably does not shift them strongly.

Reviewer zH5T: Their concerns were concept leakage/completeness, scalability to many concepts, and the fundamental limitation of real-data counterfactual validation. The authors responded with diagnostics, scaling experiments, and an explicit limitations discussion.

Reviewer AAzm. This reviewer moved from reject to borderline rejection after rebuttal, while retaining substantive skepticism about (i) whether FM inversion supports the counterfactual/editing framing, (ii) whether improvements are robust/non-cherry-picked in ablations, and (iii) whether real-data underperformance in some settings indicates weak generalization. Discussion might clarify details, but their remaining objections are conceptual and would likely persist.

---

### Decision · Program_Chairs · 2026-01-26

Reject